# Fractional Activation Functions for Off-Policy Reinforcement Learning: A Systematic Empirical Study

## Abstract

Activation functions play a key role in deep reinforcement learning by shaping how neural networks approximate policies and value functions. The Rectified Linear Unit (ReLU) remains the dominant choice due to its simplicity and computational efficiency. However, ReLU and its common variants rely on piecewise-linear transformations, which may limit the nonlinear transformations available for modeling complex dynamics in continuous-control tasks. Smooth alternatives such as the Gaussian Error Linear Unit (GELU) and Swish provide differentiable nonlinear transformations with smoother gradient behavior, offering a contrasting design philosophy to rectifier-based activations. A complementary direction is offered by fractional-order formulations, which introduce non-integer exponents that can continuously adjust the shape of the activation function. In this work, we investigate fractional-order nonlinearities that extend both rectifier-style and smooth activations while preserving their computational simplicity. We study three fractional rectifier variants, Fractional ReLU (FReLU), Fractional Leaky ReLU (FLReLU), and Fractional Parametric ReLU (FPReLU), together with two fractional smooth variants, Fractional GELU (FGELU) and Fractional Swish (FSwish). This comparison allows us to evaluate the effect of fractionalization on both rectifier-style activations and already strong smooth activations in off-policy RL.

We conduct a systematic empirical study of these activations in off-policy reinforcement learning by integrating them into two widely used algorithms, TD3 and SAC. Experiments are performed on continuous-control benchmarks from MuJoCo and the DeepMind Control Suite. Across tasks, architectures, and algorithms, fractional activations frequently outperform their corresponding non-fractional baselines. The best fractional configurations yield an average improvement of approximately 26% in normalized area under the learning curve (AUC) relative to the ReLU baseline across the evaluated algorithm–architecture settings, with seed-level statistical validation supporting the consistency of these gains. Fractional extensions of GELU and Swish further improve performance over their smooth-baseline counterparts across several settings. These results suggest that fractional activation functions provide a simple architectural modification that can improve function approximation and learning efficiency in off-policy deep reinforcement learning.

## 1 Introduction

Reinforcement learning (RL) has advanced rapidly in recent years, moving from relatively simple control problems to complex domains with high-dimensional observations, continuous action spaces, and dynamic environments (Terven, 2025; Tang et al., 2025). A key driver of this progress is the integration of deep neural networks, which provide flexible function approximators for learning policies and value functions (Arulkumaran et al., 2017). As a result, the performance of deep RL systems depends not only on the learning algorithm itself but also on the architectural choices in the underlying neural networks. These choices can influence optimization dynamics, learning stability, convergence speed, and sample efficiency (Henderson et al., 2018; Goodfellow et al., 2016). Among these architectural components, activation functions play a

particularly important role because they determine how nonlinear transformations are formed inside the actor and critic networks.

Activation functions introduce the nonlinearity required for neural networks to approximate complex mappings across diverse tasks (Dubey et al., 2022). Among them, the Rectified Linear Unit (ReLU) remains one of the most widely used choices in deep RL because of its simplicity, computational efficiency, and favorable gradient properties (Nair & Hinton, 2010; Agarap, 2018). However, ReLU also has known limitations, including inactive neurons and limited nonlinear flexibility due to its piecewise linear structure (Xu et al., 2015; Lu et al., 2020). Several rectifier variants, such as Leaky ReLU (LReLU) (Maas et al., 2013) and Parametric ReLU (PReLU) (He et al., 2015), address part of this limitation by modifying the negative region of the activation. In parallel, smoother activation functions such as the Gaussian Error Linear Unit (GELU) (Hendrycks & Gimpel, 2017) and Swish (Ramachandran et al., 2017) provide differentiable nonlinear transformations and smoother gradient behavior, making them important baselines for evaluating whether new activation designs offer benefits beyond standard rectifier families.

Fractional-order formulations provide another way to increase the flexibility of neural nonlinearities. Fractional calculus extends differentiation and integration to non-integer orders, enabling richer functional representations (Ortigueira & Coito, 2004). Fractional operators have been widely used in control theory and signal processing, where they support more flexible modeling of complex dynamical systems (Kumar et al., 2024a; Jian et al., 2025). More recently, fractional-order activation functions have been explored in deep learning and have shown promising improvements in representation quality and classification performance (Kumar et al., 2024b; Job et al., 2022; Molek & Alijani, 2025). These findings suggest that fractional transformations may provide a simple mechanism for introducing controllable nonlinear behavior into activation functions. However, despite the central role of activation functions in deep RL, fractional activation design has not been systematically evaluated in off-policy actor–critic methods.

This paper addresses this gap by studying fractional activation functions in off-policy deep RL. We consider five fractional activations spanning both rectifier-based and smooth families, namely Fractional ReLU (FReLU), Fractional Leaky ReLU (FLReLU), Fractional Parametric ReLU (FPReLU), Fractional GELU (FGELU), and Fractional Swish (FSwish). This design allows us to examine whether fractional transformations are useful only for ReLU-style nonlinearities or also for stronger smooth baselines such as GELU and Swish. Importantly, the underlying RL algorithms are left unchanged. Only the hidden-layer activation functions in the actor and critic networks are replaced, enabling a controlled evaluation of how activation design affects learning performance.

To evaluate this idea empirically, we integrate these activation functions into two widely used actor–critic algorithms, Twin Delayed Deep Deterministic Policy Gradient (TD3) (Fujimoto et al., 2018) and Soft Actor–Critic (SAC) (Haarnoja et al., 2018). We focus on these algorithms because they rely heavily on neural approximators for both policy learning and value estimation, allowing us to examine the effect of activation design on both components while keeping the learning rules unchanged. Experiments are conducted on continuous-control benchmarks from MuJoCo (Todorov et al., 2012) and the DeepMind Control Suite (Tassa et al., 2018). All activation functions are evaluated under matched training budgets, architectures, random seeds, optimizer settings, and evaluation protocols. We consider both single-hidden-layer and two-hidden-layer actor–critic architectures with fixed width. In the two-layer setting, we further examine how the placement of activations within the actor and critic networks affects performance.

Our experimental results show that fractional activations frequently improve learning performance across tasks, algorithms, and architectural settings. The rectifier-based fractional variants often outperform ReLU, LReLU, and PReLU, while the smooth fractional variants improve over GELU and Swish in several settings. These results indicate that the benefit of fractional activation design is not limited to rectifier-based nonlinearities. The strongest gains appear in several locomotion environments, where fractional activations often accelerate early learning and achieve higher final returns. Additional statistical validation using bootstrap confidence intervals, paired Wilcoxon signed-rank tests, Cliff's $\delta$, and sign-test summaries further supports the consistency of the observed improvements. The main contributions of this paper are summarized as follows.

- A systematic empirical study of activation-function design in off-policy actor–critic RL, modifying only the hidden-layer activations while keeping TD3 and SAC unchanged.

- A controlled comparison of fractional activation functions across rectifier-based and smooth activation families, including ReLU, LReLU, PReLU, GELU, Swish, FReLU, FLReLU, FPReLU, FGELU, and FSwish.

- An evaluation across MuJoCo and DeepMind Control Suite benchmarks using matched architectures, training budgets, random seeds, optimizer settings, and activation-placement strategies.

- Statistical and sensitivity analyses show that fractional activations frequently improve normalized AUC, that the best fractional order depends on the activation function, task, and algorithm, and that placement influences performance in deeper actor–critic networks.

The implementation code will be made publicly available on GitHub after the anonymous review process.

## 2 Related Work

This section reviews activation-function design in deep learning and actor–critic RL. We first discuss rectifier-based and smooth activation families, then examine neural network design choices in RL. We next review fractional activation functions and position this work as a systematic off-policy actor–critic evaluation of fractional nonlinearities.

### 2.1 Activation Functions in Deep Learning

Activation functions play a critical role in deep learning by introducing the nonlinear transformations required for neural networks to approximate complex functions (Dubey et al., 2022). Early activation functions such as the sigmoid and hyperbolic tangent were widely used, but their saturating behavior often resulted in vanishing gradients and slow optimization in deep networks (Glorot & Bengio, 2010). The introduction of ReLU significantly improved training stability by providing non-saturating gradients in the positive region, and it quickly became a dominant activation function in modern deep learning (Nair & Hinton, 2010; Agarap, 2018).

Despite its success, ReLU has several known limitations. First, its zero gradient for negative inputs can lead to neuron inactivation, commonly referred to as the dying ReLU problem, where some units become inactive for most inputs and receive little or no gradient during training (Lu et al., 2020). Second, its piecewise-linear form provides only a hard threshold at zero followed by a linear positive branch, which may limit the smooth nonlinear transformations available within each layer. Third, the non-differentiable transition at zero introduces an abrupt change in gradient, in contrast to smoother activation functions. To mitigate the zero-gradient issue in the negative region, several rectifier variants have been proposed. For example, Leaky ReLU introduces a small fixed slope for negative inputs (Maas et al., 2013), while Parametric ReLU allows this negative slope to be learned during training (He et al., 2015). Although these variants improve gradient propagation and reduce neuron inactivation, they retain the piecewise-linear structure of the original rectifier function.

A complementary direction is to use smooth nonlinearities that replace the hard thresholding behavior of ReLU with differentiable transitions. The Gaussian Error Linear Unit (GELU) weights inputs according to a Gaussian cumulative distribution, producing a smooth probabilistic gating effect that has been widely adopted in modern neural architectures (Hendrycks & Gimpel, 2017). Swish follows a related smooth-gating principle by multiplying the input by a sigmoid gate, and is closely related to the Sigmoid Linear Unit (SiLU) when the gating parameter is fixed to one (Ramachandran et al., 2017). Unlike hard rectifiers, GELU and Swish provide smoother gradient behavior and richer nonlinear shaping. For this reason, they serve as important non-fractional baselines for evaluating whether activation-function design improves performance beyond standard ReLU-family comparisons.

## 2.2   Neural Network Design in Reinforcement Learning

Deep RL relies on neural networks as function approximators for both policies and value functions. Consequently, architectural choices that influence optimization dynamics and function approximation can directly affect agent performance. However, many RL algorithms adopt neural network components originally developed for supervised learning, with relatively limited investigation of their impact in RL settings. In practice, ReLU is commonly used in deep RL algorithms such as Deep Q-Networks (DQN) (Mnih et al., 2015), Deep Deterministic Policy Gradient (DDPG) (Lillicrap et al., 2015), TD3 (Fujimoto et al., 2018), and SAC (Haarnoja et al., 2018).

A large body of RL research improves performance by modifying the agent's algorithmic or architectural components. One prominent direction focuses on improving value estimation through critic ensembles. For example, Randomized Ensembled Double Q-learning (REDQ) improves sample efficiency by maintaining a large ensemble of critic networks with randomized updates (Chen et al., 2021). Similarly, Truncated Quantile Critics (TQC) reduce overestimation in distributional RL by truncating the upper tail of the learned value distribution (Kuznetsov et al., 2020). These approaches demonstrate that architectural choices within value-function approximation can substantially affect the stability and performance of deep RL algorithms.

Other work improves RL performance through regularization and normalization mechanisms. DroQ introduces dropout into critic networks to approximate ensemble-style regularization at a lower computational cost (Hiraoka et al., 2022). Similarly, CrossQ incorporates batch normalization within Q-networks to stabilize training and improve sample efficiency (Bhatt et al., 2024). These studies further show that neural network design choices can influence optimization behavior in deep RL.

Despite these advances, comparatively little attention has been given to activation functions themselves. As fundamental elements of neural architectures, activation functions define the nonlinear transformations used in policy and value approximation and therefore influence gradient propagation and optimization behavior (Goodfellow et al., 2016; Glorot & Bengio, 2010). This suggests that activation-function design remains an underexplored architectural factor in RL.

## 2.3   Fractional Activation Functions

One promising direction for designing more flexible nonlinearities arises from fractional calculus. Fractional calculus extends classical differentiation and integration to non-integer orders, enabling more flexible mathematical representations of dynamic systems (Ortigueira & Coito, 2004). Fractional-order models have been widely applied in areas such as control systems and signal processing, where they can capture complex system behaviors more effectively than integer-order formulations (Jian et al., 2025).

More recently, fractional-order concepts have been incorporated into neural network activation functions. Prior studies have shown that fractional-order parameterizations can produce flexible nonlinear transformations and improve performance in supervised learning settings across different architectures and activation families (Job et al., 2022; Kumar et al., 2024b; Molek & Alijani, 2025). Recent work has also investigated fractional-order derivatives as tunable components in activation design, analyzing their effects on accuracy, computational cost, and memory usage (Molek & Alijani, 2025). These studies suggest that fractional activations form a broader research direction rather than an isolated modification of a single rectifier variant.

However, existing studies on fractional activation functions have primarily focused on supervised learning tasks such as classification and regression, while their impact on RL remains largely unexplored (Job et al., 2022; Kumar et al., 2024b; Molek & Alijani, 2025). In this work, we address this gap by systematically evaluating fractional activation functions in off-policy actor–critic RL. The rectifier-based fractional formulations, FReLU, FLReLU, and FPReLU, are adapted from prior fractional activation studies and evaluated in the RL setting. We further extend the comparison to smooth fractional variants, FGELU and FSwish, to examine whether fractional transformations remain useful beyond rectifier-based activations and when applied to already smooth nonlinearities.

### 2.4 Motivation for Fractional Activations in Actor–Critic RL

Activation functions are an important architectural component in actor–critic RL because they define the nonlinear transformations and gradient pathways used by policy and value networks (Goodfellow et al., 2016; Glorot & Bengio, 2010). Unlike standard supervised learning, off-policy RL trains these networks with bootstrapped targets, replayed experience, evolving data distributions, and correlated samples (Sutton & Barto, 2018; Mnih et al., 2015; Lillicrap et al., 2015). These properties make deep RL sensitive to implementation details, hyperparameter choices, optimization settings, and architectural design decisions (Henderson et al., 2018; Agarwal et al., 2021). Therefore, changing the activation function can affect optimization dynamics and function approximation even when the underlying RL algorithm remains unchanged.

This is particularly relevant in TD3 and SAC. In TD3, critic approximation errors directly affect deterministic policy updates, motivating mechanisms such as clipped double-Q learning and delayed policy updates (Fujimoto et al., 2018). In SAC, the critic provides the value signal used for entropy-regularized policy optimization (Haarnoja et al., 2018). In both algorithms, the actor and critic are coupled through gradients and bootstrapped targets. Activation-function design can therefore influence value estimation, policy gradients, and the interaction between actor and critic.

Fractional activations provide a controlled way to modify activation shape without changing the RL algorithm, network width, target updates, or policy objective. Prior work shows that fractional-order activation formulations can modify nonlinear responses and affect training behavior in deep networks (Job et al., 2022; Kumar et al., 2024b; Molek & Alijani, 2025). For rectifier-based functions, fractionalization introduces curvature while preserving the basic gating structure of ReLU-style activations. For smooth functions such as GELU and Swish, fractional variants provide an additional mechanism for adjusting the nonlinear response while retaining smooth gradient behavior (Hendrycks & Gimpel, 2017; Ramachandran et al., 2017). This motivates the evaluation of fractional activation functions as lightweight architectural modifications for policy learning and value approximation in off-policy actor–critic RL.

## 3 Methodology

This section describes the methodology used to evaluate activation-function design in off-policy actor–critic RL. We first define the non-fractional baselines and the fractional activation functions evaluated in this work. We then describe the fractional-order parameter, implementation details, network architectures, activation placement strategies, and their integration into TD3 and SAC.

### 3.1 Baseline Activation Functions

We first define the non-fractional activation functions used as baselines. The study includes two baseline groups. The first group consists of rectifier-based activations, ReLU, LReLU, and PReLU, which provide direct counterparts for the fractional rectifier variants. The second group consists of smooth activations, GELU and Swish, which provide stronger modern baselines beyond the ReLU family.

#### 3.1.1 Rectifier-Based Baselines

The standard ReLU activation is defined as

$$\text{ReLU}(x) = \max(0, x). \tag{1}$$

ReLU preserves positive inputs and suppresses negative inputs, providing a simple non-saturating positive branch.

Leaky ReLU introduces a fixed non-zero slope in the negative region, defined as

$$\text{LReLU}_k(x) = \begin{cases} x, & x \geq 0, \\ kx, & x < 0, \end{cases} \tag{2}$$

where $k$ controls the negative-side slope. In this study, we set $k = 0.1$. This activation preserves the rectifier structure while allowing gradient flow for negative inputs.

Parametric ReLU replaces the fixed negative-side coefficient with a learnable parameter, defined as

$$\text{PReLU}_p(x) = \begin{cases} x, & x \geq 0, \\ px, & x < 0, \end{cases} \tag{3}$$

where $p$ is optimized during training. This allows the negative-side response to adapt during learning.

Together, ReLU, LReLU, and PReLU define the rectifier-family baselines used to evaluate whether fractional rectifier variants improve over their corresponding non-fractional forms.

### 3.1.2 Smooth Baselines

GELU is a smooth activation that weights inputs according to the standard Gaussian cumulative distribution function, defined as

$$\text{GELU}(x) = x\Phi(x), \tag{4}$$

where $\Phi(x)$ denotes the cumulative distribution function of the standard Gaussian distribution. In implementation, we use the common tanh approximation:

$$\text{GELU}(x) \approx \frac{1}{2}x\left(1 + \tanh\left[\sqrt{\frac{2}{\pi}}\left(x + 0.044715x^3\right)\right]\right). \tag{5}$$

Swish is a smooth self-gated activation function defined as

$$\text{Swish}(x) = x\sigma(x), \tag{6}$$

where

$$\sigma(x) = \frac{1}{1 + \exp(-x)}. \tag{7}$$

GELU and Swish are included to test whether fractional activations remain useful when compared against smooth nonlinearities rather than only against ReLU-family baselines.

## 3.2 Fractional Activation Functions

Fractional activation functions introduce an additional fractional-order parameter that modifies activation shape while preserving compatibility with standard actor–critic networks. In this study, we evaluate five fractional activations grouped into two families. The rectifier-based variants, FReLU, FLReLU, and FPReLU, are adapted from prior fractional-activation studies and implemented here for off-policy actor–critic RL (Molek & Alijani, 2025; Job et al., 2022; Kumar et al., 2024b). The smooth variants, FGELU and FSwish, are introduced in this work as RL-oriented smooth fractional activations that extend the design of fractional activations beyond ReLU-style nonlinearities.

### 3.2.1 Fractional Calculus Basis

Fractional activation functions are motivated by fractional calculus, which extends classical differentiation and integration to non-integer orders (Oldham & Spanier, 1974; Podlubny, 1998). Unlike integer-order derivatives, fractional derivatives define a continuous family of operators, allowing the order of the transformation to be adjusted through a fractional parameter. This provides a mathematical basis for activation functions whose nonlinear shape can be continuously controlled.

Several definitions of fractional derivatives exist, including the Riemann–Liouville, Caputo, and Grünwald–Letnikov formulations (Podlubny, 1998; Samko, 1993). Although these definitions differ in their construction,

they yield a useful closed-form expression for power functions. For a power function $f(x) = x^q$, the fractional derivative of order $\alpha$ can be written as

$$\frac{d^\alpha}{dx^\alpha} x^q = \frac{\Gamma(q+1)}{\Gamma(q+1-\alpha)} x^{q-\alpha}, \tag{8}$$

where $\Gamma(\cdot)$ denotes the Gamma function, which extends the factorial to non-integer values.

Equation equation 8 shows that the fractional order changes the exponent of the power function continuously. For activation design, this motivates transformations based on $x^{1-\alpha}$, where $\alpha$ controls the activation's curvature. Smaller values of $\alpha$ keep the activation closer to its base form, while larger values introduce stronger fractional curvature.

### 3.2.2 Fractional Rectifier Variants

The rectifier-based fractional activations, FReLU, FLReLU, and FPReLU, extend standard rectifier functions by replacing linear rectifier branches with fractional-power transformations. The fractional order $\alpha$ controls the curvature of the activation. When $\alpha$ is close to zero, the activation remains close to its corresponding classical rectifier. As $\alpha$ increases, the exponent $1 - \alpha$ decreases, producing a more curved sublinear response in the positive region.

**Fractional ReLU.**

Fractional ReLU modifies the positive region of the standard ReLU by introducing the following fractional-power transformation.

$$\text{FReLU}_\alpha(x) = \begin{cases} x^{1-\alpha}, & x > 0, \\ 0, & x \leq 0. \end{cases} \tag{9}$$

The exponent $1 - \alpha$ controls the curvature of the positive branch. When $\alpha = 0$, FReLU reduces to the standard ReLU. For $\alpha > 0$, the positive branch becomes nonlinear while the negative branch remains zero. Thus, FReLU preserves the gating behavior of ReLU while introducing curvature control through $\alpha$ without adding learnable parameters.

**Fractional Leaky ReLU.**

Fractional Leaky ReLU extends LReLU by applying fractional-power transformations to both positive and negative regions.

$$\text{FLReLU}_{\alpha,k}(x) = \begin{cases} \dfrac{1}{\Gamma(2-\alpha)} x^{1-\alpha}, & x > 0, \\ -\dfrac{k}{\Gamma(2-\alpha)} |x|^{1-\alpha}, & x < 0, \end{cases} \tag{10}$$

where $k > 0$ is a fixed negative-side slope. In this study, we use the same fixed value as the LReLU baseline, $k = 0.1$. Therefore, FLReLU is the fractional counterpart of LReLU. It preserves a fixed non-zero negative response, but replaces the linear positive and negative branches with fractional-power branches. The parameter $\alpha$ controls the curvature of both branches, while $k$ controls the fixed magnitude of the negative-side response.

**Fractional Parametric ReLU.**

Fractional Parametric ReLU extends FLReLU by replacing the fixed negative-side slope with a learnable coefficient.

$$\text{FPReLU}_{\alpha,p}(x) = \begin{cases} \dfrac{1}{\Gamma(2-\alpha)} x^{1-\alpha}, & x > 0, \\ -\dfrac{p}{\Gamma(2-\alpha)} |x|^{1-\alpha}, & x < 0, \end{cases} \tag{11}$$

where $p$ is optimized during training. In this formulation, $\alpha$ controls the curvature of both branches, while $p$ adaptively controls the negative-side response. Thus, FPReLU combines fractional curvature control with a learnable negative branch, making it the fractional counterpart of PReLU.

In the RL implementation, we adapt these mathematical definitions to improve stability during repeated off-policy actor–critic updates. This adaptation introduces three practical changes. First, fractional powers are computed using safeguarded positive and negative branches,

$$x_\epsilon^+ = \max(x, \epsilon), \qquad x_\epsilon^- = \max(-x, \epsilon), \tag{12}$$

where $\epsilon$ is a small numerical constant. This avoids numerical instability when fractional powers are evaluated near zero.

Second, the two-sided fractional rectifier variants, FLReLU and FPReLU, use Gamma-based scale control through the factor $1/\Gamma(2 - \alpha)$. This helps keep activation magnitudes comparable across different fractional orders and ensures that the fractional form remains close to the corresponding classical rectifier when $\alpha = 0$.

Third, the negative-side coefficient differs between FLReLU and FPReLU. For FLReLU, the coefficient is fixed, matching the LReLU baseline with $k = 0.1$. For FPReLU, the coefficient is learned during training but bounded for stability.

$$p_c = \mathrm{clip}\left(\mathrm{softplus}(p_{\mathrm{raw}}), 0.01, 0.3\right). \tag{13}$$

The softplus operation keeps the learned coefficient positive, while clipping prevents the negative-side response from becoming too small or excessively large during training. With these safeguards, the implemented FPReLU uses $p_c$ in place of $p$, and the implemented fractional branches use $x_\epsilon^+$ and $x_\epsilon^-$ in place of $x$ and $|x|$.

### 3.2.3 Fractional Smooth Variants

To evaluate fractional activation design beyond rectifier-style nonlinearities, we introduce smooth fractional variants of GELU and Swish. These activations are designed as RL-oriented smooth fractional nonlinearities that combine fractional modulation with adaptive smooth gating. In both cases, we use a bounded learnable gating parameter $\beta_c$:

$$\beta_c = \mathrm{clip}\left(\mathrm{softplus}(\beta_{\mathrm{raw}}), 0.1, 10.0\right). \tag{14}$$

The role of $\beta_c$ is to control the transition sharpness of the smooth gate. This is useful for GELU and Swish because both activations are gated nonlinearities. Swish uses a sigmoid gate, while GELU uses a Gaussian-inspired smooth gate (Ramachandran et al., 2017; Hendrycks & Gimpel, 2017). In contrast, applying a similar scaling parameter to a fractional rectifier branch mainly gives

$$(\beta x)^{1-\alpha} = \beta^{1-\alpha} x^{1-\alpha}, \tag{15}$$

which acts primarily as magnitude rescaling rather than introducing a new gating mechanism. For this reason, the bounded gating parameter is used only in the proposed smooth fractional variants, while FReLU, FLReLU, and FPReLU are treated as fractional rectifier variants without an additional gate-sharpness parameter.

**Fractional GELU.**

FGELU is introduced in this work as a smooth fractional activation for off-policy actor–critic RL. It applies a finite Grünwald–Letnikov-style fractional approximation to an adaptive GELU core. We first define the adaptive GELU core as

$$\psi_\beta(z) = z \left(1 + \tanh\left[\sqrt{\frac{2}{\pi}}\left(\beta_c z + 0.044715(\beta_c z)^3\right)\right]\right). \tag{16}$$

The implemented fractional GELU activation is then defined as

$$\mathrm{FGELU}_{\alpha,\beta}(x) = \frac{1}{2h^\alpha} \sum_{i=0}^{N-1} (-1)^i \frac{\Gamma(1+\alpha)}{\Gamma(i+1)\Gamma(1-i+\alpha)} \psi_\beta(x - ih). \tag{17}$$

In our implementation, we use $h = 0.5$ and $N = 3$. This finite approximation keeps the activation computationally practical during repeated actor and critic updates. Undefined or infinite outputs are replaced

with bounded finite values during training. The bounded parameter $\beta_c$ allows the smooth gating response to adapt during learning, while $\alpha$ controls the fractional modulation.

**Fractional Swish.**

FSwish is introduced in this work as a smooth fractional activation based on an adaptive Swish core. We first define the adaptive Swish core as

$$\text{Swish}_\beta(x) = x\sigma(\beta_c x). \tag{18}$$

The implemented fractional Swish activation is then defined as

$$\text{FSwish}_{\alpha,\beta}(x) = \text{Swish}_\beta(x) + \alpha\sigma(\beta_c x)\left(1 - \text{Swish}_\beta(x)\right). \tag{19}$$

Here, $\alpha$ controls the fractional modulation and $\beta_c$ controls the smooth gating response. When $\alpha = 0$, the fractional modulation term is removed, and the activation reduces to the adaptive Swish core. Standard Swish is recovered as the special case $\beta_c = 1$ (Ramachandran et al., 2017).

Together, FGELU and FSwish allow us to separate two effects. The first is the benefit of using smooth activation functions rather than rectifier-based ones. The second is the additional effect of fractional modulation on top of smooth gated nonlinearities. This distinction is important because the revised evaluation includes GELU and Swish as non-fractional smooth baselines, rather than comparing fractional activations only against ReLU-family functions.

Figure 1 illustrates the activation functions evaluated in this study for different values of the fractional order $\alpha$. The figure shows how $\alpha$ modifies the shape of each activation family. For the rectifier-based variants, increasing $\alpha$ introduces progressively stronger curvature relative to the corresponding ReLU-style baseline. For FGELU and FSwish, $\alpha$ controls the strength of the fractional modulation applied to the adaptive smooth core. Overall, smaller values of $\alpha$ keep the activation closer to its base form, while larger values produce stronger deviations.

A structural summary of all baseline and fractional activations, including their learnable parameters and roles in the comparison, is provided in Appendix A.

### 3.3 Fractional-Order Search Space

The fractional-order parameter $\alpha$ controls how strongly a fractional activation deviates from its non-fractional baseline. Smaller values keep the activation closer to the original function, while larger values produce stronger shape changes that can affect activation magnitudes and gradients. Thus, $\alpha$ is treated as an activation-shape parameter rather than a parameter for which larger values are always expected to improve performance.

In this study, we evaluate

$$\alpha \in \{0.1, 0.2, 0.3, 0.4, 0.5\}. \tag{20}$$

For all fractional activations, $\alpha$ is fixed within each experiment rather than learned. This enables a controlled sensitivity analysis and avoids introducing activation-specific tuning through a learned fractional order.

### 3.4 Network Architectures

To isolate the influence of activation functions from changes in model capacity, all activations are evaluated using matched multilayer perceptron actor and critic networks. The hidden-layer width is fixed to 256 units across all configurations.

**Single-hidden-layer architecture** In the first configuration, both actor and critic networks contain one hidden layer. This setting provides a controlled architecture in which differences in performance can be attributed primarily to the choice of activation function.

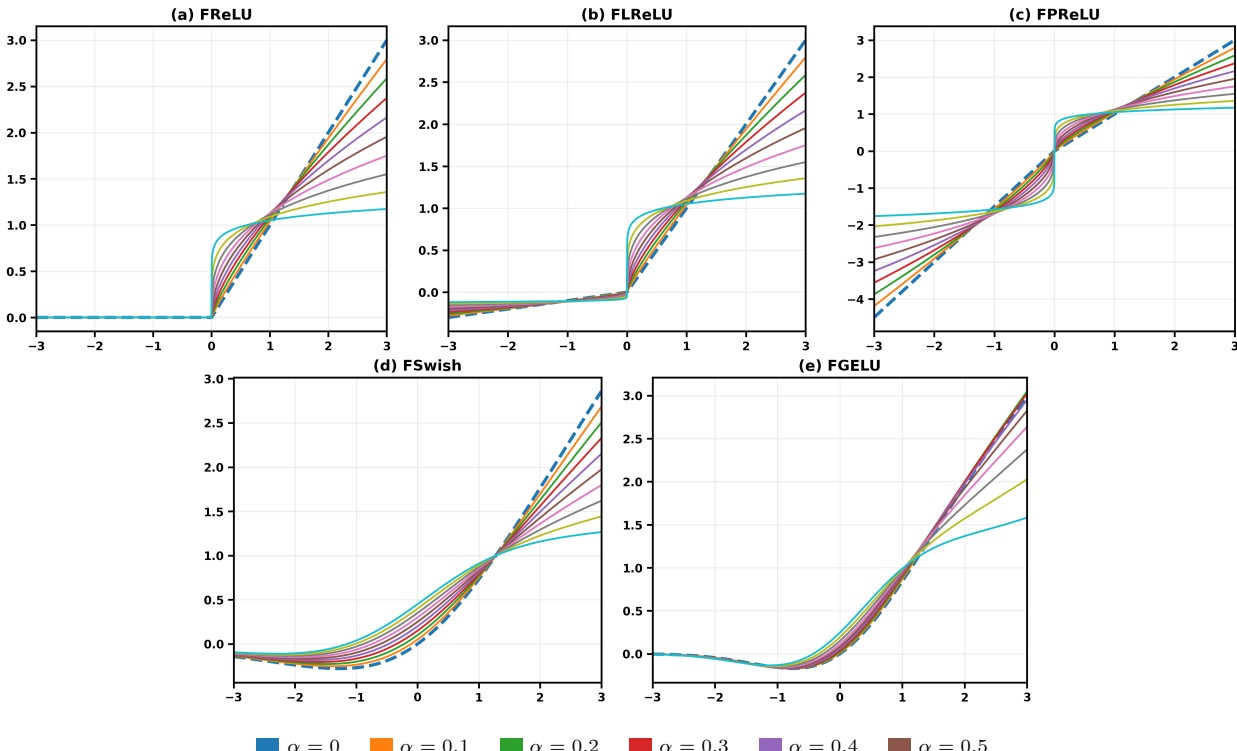

Figure 1: Fractional activation functions evaluated in this study for different values of the fractional order $\alpha$. The rectifier-based variants introduce fractional curvature into ReLU-style activations, while the smooth variants introduce fractional modulation into GELU- and Swish-based gated activations. For the rectifier-based variants, $\alpha = 0$ recovers the corresponding non-fractional baseline, up to the numerical safeguards used in the implementation. For the smooth variants, $\alpha = 0$ removes the fractional modulation and leaves the corresponding adaptive smooth core.

**Two-hidden-layer architecture**   In the second configuration, both actor and critic networks contain two hidden layers. This architecture matches the standard multilayer perceptron structure commonly used in TD3 and SAC. The two-layer setting allows us to examine whether fractional activations interact with deeper representations and whether their effects depend on their placement in the actor–critic architecture.

### 3.5   Activation Placement

Activation placement is evaluated for the two-hidden-layer architecture, where nonlinearities can be applied to different parts of the actor–critic networks. This analysis examines whether activation-function effects are mainly associated with policy representation, value estimation, or their joint interaction.

We evaluate the following placement strategies:

- `all-actor` (AA): the selected activation is applied to all hidden layers of the actor network,

- `all-critic` (AC): the selected activation is applied to all hidden layers of the critic network,

- `all-both` (AB): the selected activation is applied to all hidden layers of both actor and critic networks,

- `first-actor` (FA): the selected activation is applied only to the first hidden layer of the actor network,

- `first-both` (FB): the selected activation is applied only to the first hidden layer of both actor and critic networks.

The actor-only configurations, AA and FA, isolate the effect of activation design on policy representation. The critic-only configuration, AC, focuses on value-function approximation and the gradients used to guide policy updates. The AB configuration evaluates the effect of applying the same activation throughout the actor–critic system. The first-layer configurations, FA and FB, test whether modifying the initial transformation of state or state-action inputs is sufficient to affect learning. Full placement-specific results are reported in the appendix, while the main text summarizes the dominant trends.

### 3.6 Integration into TD3 and SAC

We integrate the evaluated activation functions into two standard off-policy continuous-control algorithms, TD3 (Fujimoto et al., 2018) and SAC (Haarnoja et al., 2018). In both algorithms, the actor and critic are implemented as multilayer perceptrons. Let $H = 256$ denote the hidden-layer width used in all actor and critic networks.

The hidden-layer activation is the only architectural component varied in this study. For a single-hidden-layer actor or critic, the hidden representation is

$$\texttt{Linear}(d_{\text{in}} \rightarrow H) \rightarrow \phi,$$

where $\phi$ denotes the selected activation function. For the actor, $d_{\text{in}}$ is the state dimension. For the critic, $d_{\text{in}}$ is the state-action dimension. The output layers are unchanged and follow the standard TD3 or SAC design.

For the two-hidden-layer architecture, the hidden representation is

$$\texttt{Linear}(d_{\text{in}} \rightarrow H) \rightarrow \phi_1 \rightarrow \texttt{Linear}(H \rightarrow H) \rightarrow \phi_2,$$

where $\phi_1$ and $\phi_2$ are determined by the activation-placement strategy defined in Section 3.5.

For fractional activations, $\phi$ denotes one of FReLU, FLReLU, FPReLU, FGELU, or FSwish with a fixed fractional order $\alpha$. For non-fractional baselines, $\phi$ denotes ReLU, LReLU, PReLU, GELU, or Swish. All non-activation components of TD3 and SAC, including replay, target-network updates, policy updates, optimization settings, and training budgets, are kept fixed. This isolates activation-function design as the experimental variable of interest.

## 4 Experiments

This section evaluates the effect of activation-function design in off-policy continuous-control RL. We compare conventional rectifier activations, modern smooth activations, and their fractional variants within matched TD3 (Fujimoto et al., 2018) and SAC (Haarnoja et al., 2018) architectures. The evaluated baselines include ReLU, LReLU, PReLU, Swish, and GELU. The evaluated fractional activations include the rectifier-based variants FReLU, FLReLU, and FPReLU, as well as the smooth variants FGELU and FSwish. This design allows us to examine whether the observed gains are specific to fractional rectifier functions or whether fractional transformations remain useful when compared against stronger smooth activation baselines.

All experiments are designed to isolate the effect of the activation function. The underlying RL algorithms, network widths, optimizer settings, replay mechanisms, target-network updates, policy update rules, training budgets, and evaluation schedules are kept unchanged across activation configurations. This controlled setup allows performance differences to be attributed primarily to the choice of activation function rather than to algorithmic or hyperparameter changes.

### 4.1 Benchmarks

We evaluate the activation functions on continuous-control tasks drawn from two widely used RL benchmark suites, MuJoCo environments accessed through OpenAI Gym (Brockman et al., 2016) and the DeepMind Control Suite (Tassa et al., 2018). These benchmarks include high-dimensional state spaces, nonlinear

dynamics, and diverse control objectives, providing a broad testbed for analyzing the effect of activation design in actor–critic learning.

The evaluation includes the following tasks.

**MuJoCo** *Ant-v4*, *HalfCheetah-v4*, *Hopper-v4*, and *Humanoid-v4.* These environments focus on locomotion control with articulated agents. *HalfCheetah-v4* and *Hopper-v4* emphasise forward locomotion efficiency, while *Ant-v4* and *Humanoid-v4* involve higher-dimensional control and coordinated whole-body motion.

**DeepMind Control Suite** *Walker-Walk*, *Cheetah-Run*, *Finger-Spin*, and *Cartpole-Swingup.* These tasks include locomotion, manipulation, and balance challenges. *Walker-Walk* and *Cheetah-Run* evaluate locomotion and stability, while *Finger-Spin* and *Cartpole-Swingup* require precise control and dynamic balancing.

Together, these tasks cover a range of continuous-control challenges, including high-dimensional locomotion, underactuated balance, coordinated whole-body movement, and dynamic control requiring precise timing. The benchmark illustration is moved to the appendix to reduce redundancy in the main text.

## 4.2 Experimental Setup

All experiments follow standard TD3 and SAC training configurations for continuous-control benchmarks. The actor and critic networks are implemented as multilayer perceptrons with hidden layers of width 256, following the architectures described in Section 3.4. We evaluate both single-hidden-layer and two-hidden-layer architectures.

Agents are trained for one million environment steps. During training, policy evaluations are performed periodically using deterministic evaluation policies without exploration noise. Each evaluation consists of 10 episodes, and the average return is recorded to construct the learning curves.

The evaluated activation functions and placement strategies follow the definitions in Sections 3.1–3.5. The hidden-layer activation is the main architectural component varied across configurations. For the two-hidden-layer architecture, we additionally vary activation placement to examine whether activation effects differ across the actor, critic, and first hidden layers.

The fractional-order settings follow Eq. equation 20. This range is chosen to examine mild-to-moderate fractional transformations while remaining close to the corresponding non-fractional activation functions. We do not extend the main grid to larger values such as $\alpha > 0.5$ because, for power-based fractional rectifiers, the exponent $1 - \alpha$ becomes smaller than 0.5. This produces stronger activation compression and sharper behavior near zero, which can increase numerical and optimization sensitivity during repeated actor and critic updates. Therefore, the main experiments focus on a range that introduces meaningful fractional curvature while remaining close to the corresponding base activations.

Learnable internal activation parameters, including the PReLU slope, FLReLU negative coefficient, FPReLU negative coefficient, and the smooth gating parameter $\beta_c$ in FGELU and FSwish, are optimized jointly with the actor and critic network parameters. Numerical safeguards, bounded learnable parameters, and finite-fractional approximations are used to support stable off-policy actor–critic training.

Unless otherwise specified, all hyperparameters remain identical across activation functions. No activation-specific learning-rate tuning, initialization-specific tuning, or optimizer tuning is performed. This avoids confounding activation effects with differences in tuning effort.

## 4.3 Evaluation Protocol

During training, agents are evaluated periodically, and the returns obtained at each evaluation point are averaged following standard RL evaluation practice (Henderson et al., 2018; Agarwal et al., 2021). The resulting learning curves represent the mean episode return as training progresses.

To quantify overall learning performance, we compute the area under the learning curve over the one-million-step training horizon. For each run, the evaluation returns are integrated using the trapezoidal rule and normalized by the total training budget. Let $R_j$ denote the average evaluation return at step $t_j$. The normalized AUC is computed as

$$\text{AUC} = \frac{1}{T} \sum_{j=1}^{m-1} \frac{R_j + R_{j+1}}{2} \left( t_{j+1} - t_j \right), \tag{21}$$

where $T$ is the total training horizon and $m$ is the number of evaluation points. This metric captures both learning speed and final performance, making it more informative than terminal return alone.

Because reward scales differ across environments, the main results are reported as relative AUC improvements over the matched ReLU baseline. For a given configuration, the relative improvement is

$$\Delta_{\text{ReLU}}\% = \frac{\text{AUC}_{\text{config}} - \text{AUC}_{\text{ReLU}}}{|\text{AUC}_{\text{ReLU}}|} \times 100. \tag{22}$$

For smooth fractional activations, we additionally report comparisons against their corresponding smooth baselines when appropriate. This matched-baseline comparison is computed as

$$\Delta_{\text{matched}}\% = \frac{\text{AUC}_{\text{frac}} - \text{AUC}_{\text{base}}}{|\text{AUC}_{\text{base}}|} \times 100, \tag{23}$$

where base denotes the corresponding non-fractional activation. For example, FSwish is compared against Swish, and FGELU is compared against GELU. This distinction allows us to test whether fractional variants improve over their direct non-fractional counterparts, rather than only over ReLU.

Each configuration is trained with five independent random seeds. All statistical comparisons are performed using seed-matched AUC values whenever the same seed is available for both configurations.

## 5 Results

This section reports the empirical evaluation of fractional activation functions in continuous-control RL. We evaluate TD3 and SAC under single-layer and two-layer actor–critic architectures, varying both the activation function and its placement within the network. The results are presented in four stages, covering overall performance across tasks, sensitivity to the fractional order $\alpha$, activation-family and placement effects, and statistical validation. Unless stated otherwise, performance is reported relative to the ReLU baseline using the $\Delta\%$ metric defined in Section 4.3.

### 5.1 Overall Performance Across Tasks

This subsection provides an overall comparison of the evaluated activation functions across tasks, algorithms, and network depths. We first use learning curves to examine training behavior over time, then use heatmaps to summarize AUC improvements across activation–task pairs, and finally report the best-performing configuration for each task in Table 1. Together, these results provide complementary views of learning dynamics, aggregate performance, and task-level activation preference.

Figures 2, 3, 4, and 5 show representative learning curves over one million environment steps. Each curve corresponds to the best-performing configuration within an activation family. Solid lines denote non-fractional activations, while dashed lines denote their fractional counterparts, allowing direct comparison between each base activation and its fractional variant.

The learning curves indicate that fractional activations often improve learning speed, final return, or both. These improvements are particularly visible in locomotion tasks such as *HalfCheetah-v4*, *Hopper-v4*, and *Walker-Walk*. The curves also show that smooth activations, especially Swish, are strong baselines in several

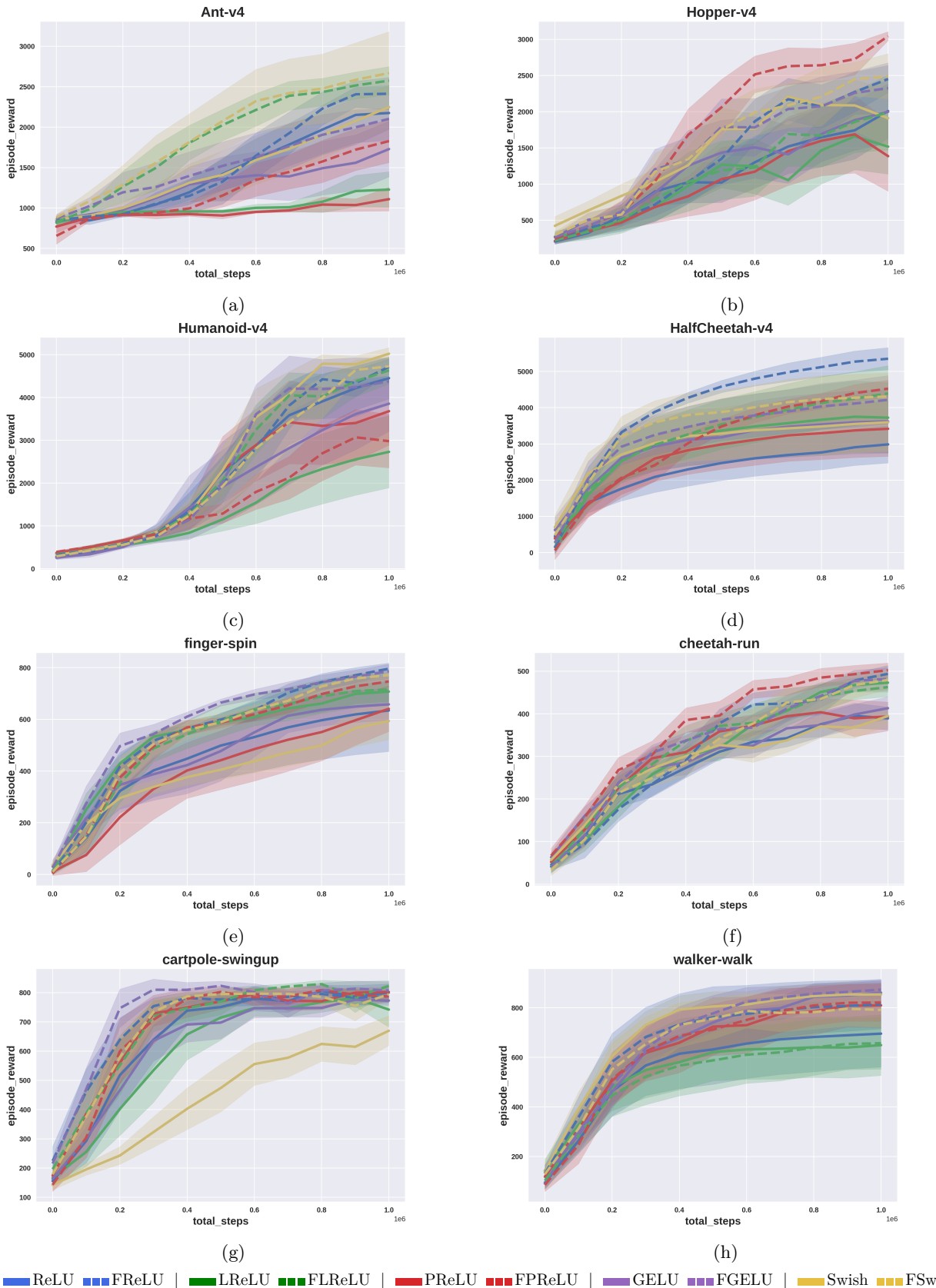

Figure 2: Learning curves for **TD3 with a single hidden layer**. Curves show the mean episodic return over five independent random seeds across one million training steps, with shaded regions indicating the standard deviation across seeds. Results are smoothed using a sliding window of 10 evaluation steps.

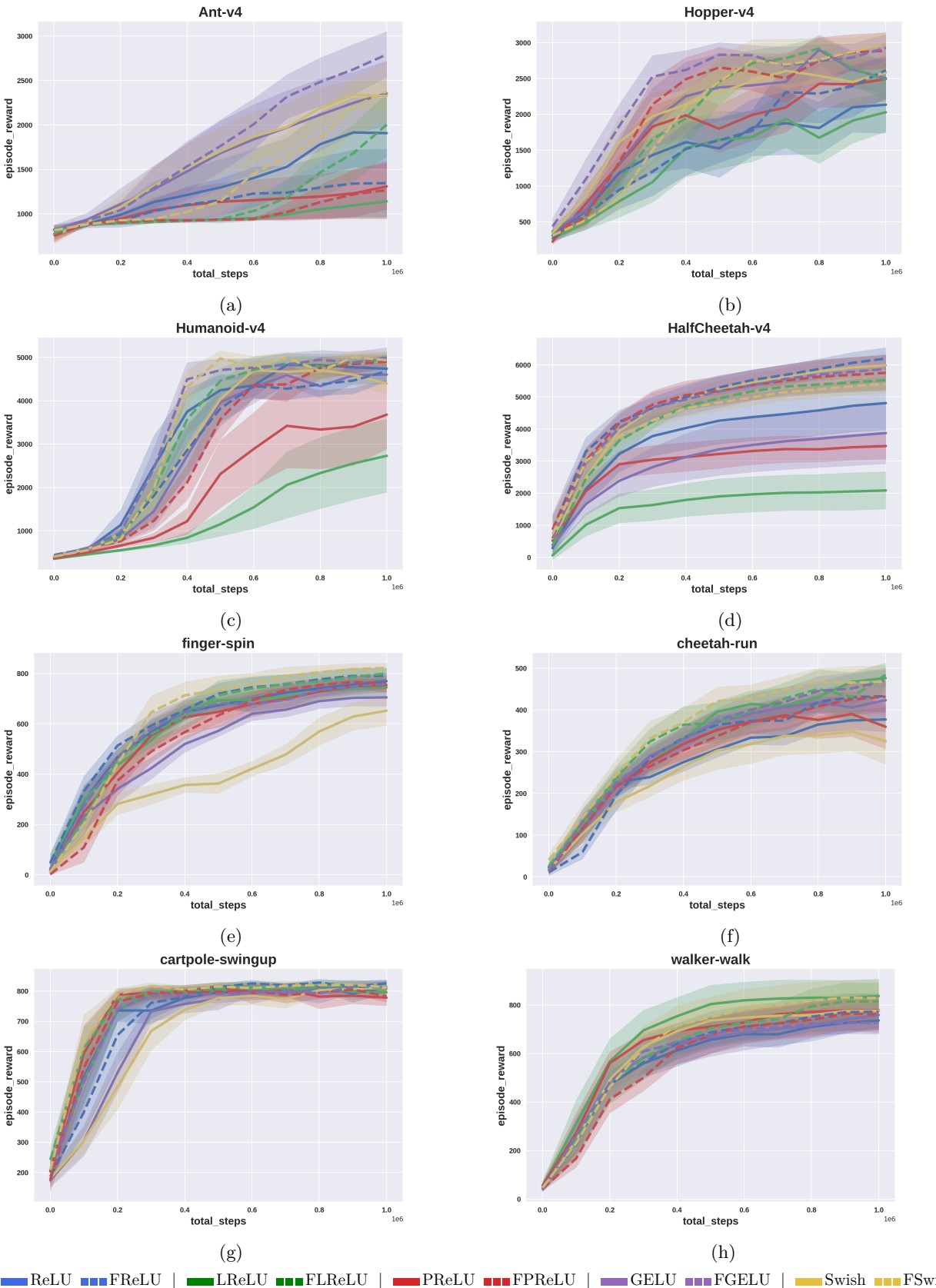

Figure 3: Learning curves for **SAC with a single hidden layer**. Curves show the mean episodic return over five independent random seeds, with shaded regions indicating the standard deviation across seeds. Results are smoothed using a sliding window of 10 evaluation steps.

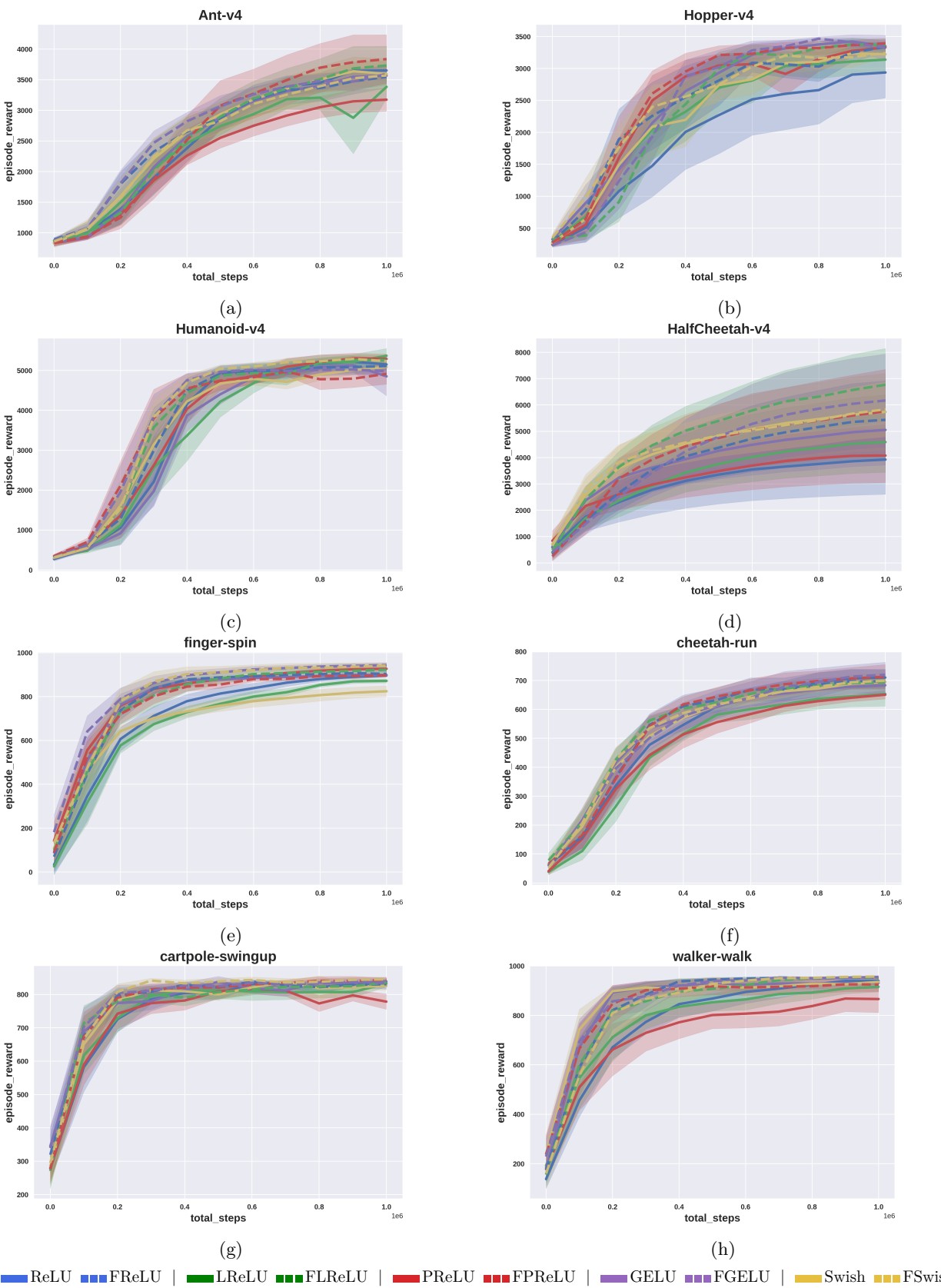

Figure 4: Learning curves for **TD3 with two hidden layers**. The best activation configuration for each task, including placement and $\alpha$, is compared with ReLU. Curves show the mean episodic return over five seeds, with shaded regions indicating standard deviation.

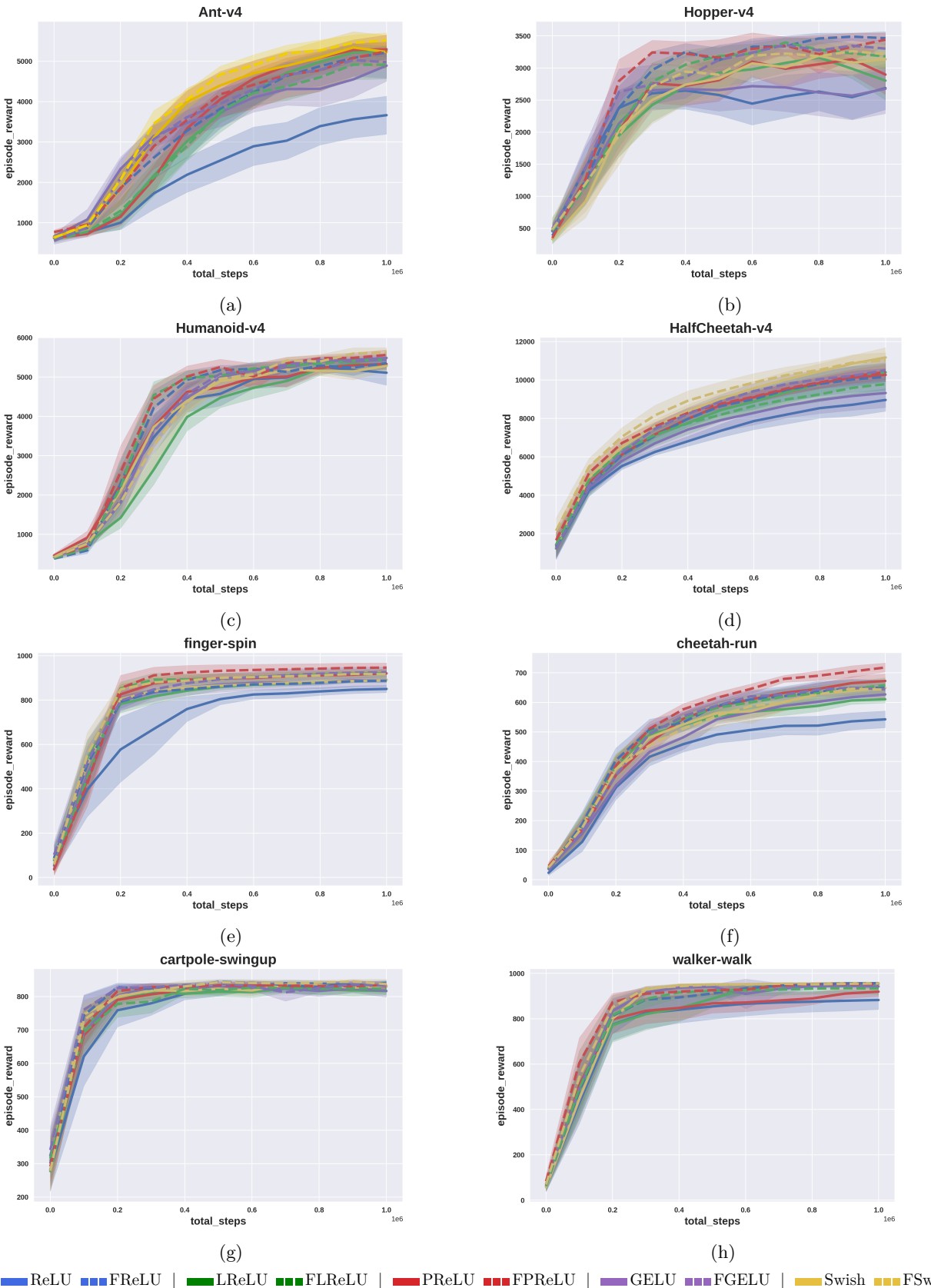

Figure 5: Learning curves for **SAC with two hidden layers**. The best activation configuration for each task, including placement and $\alpha$, is compared with ReLU. Curves show the mean episodic return over five seeds, with shaded regions indicating standard deviation.

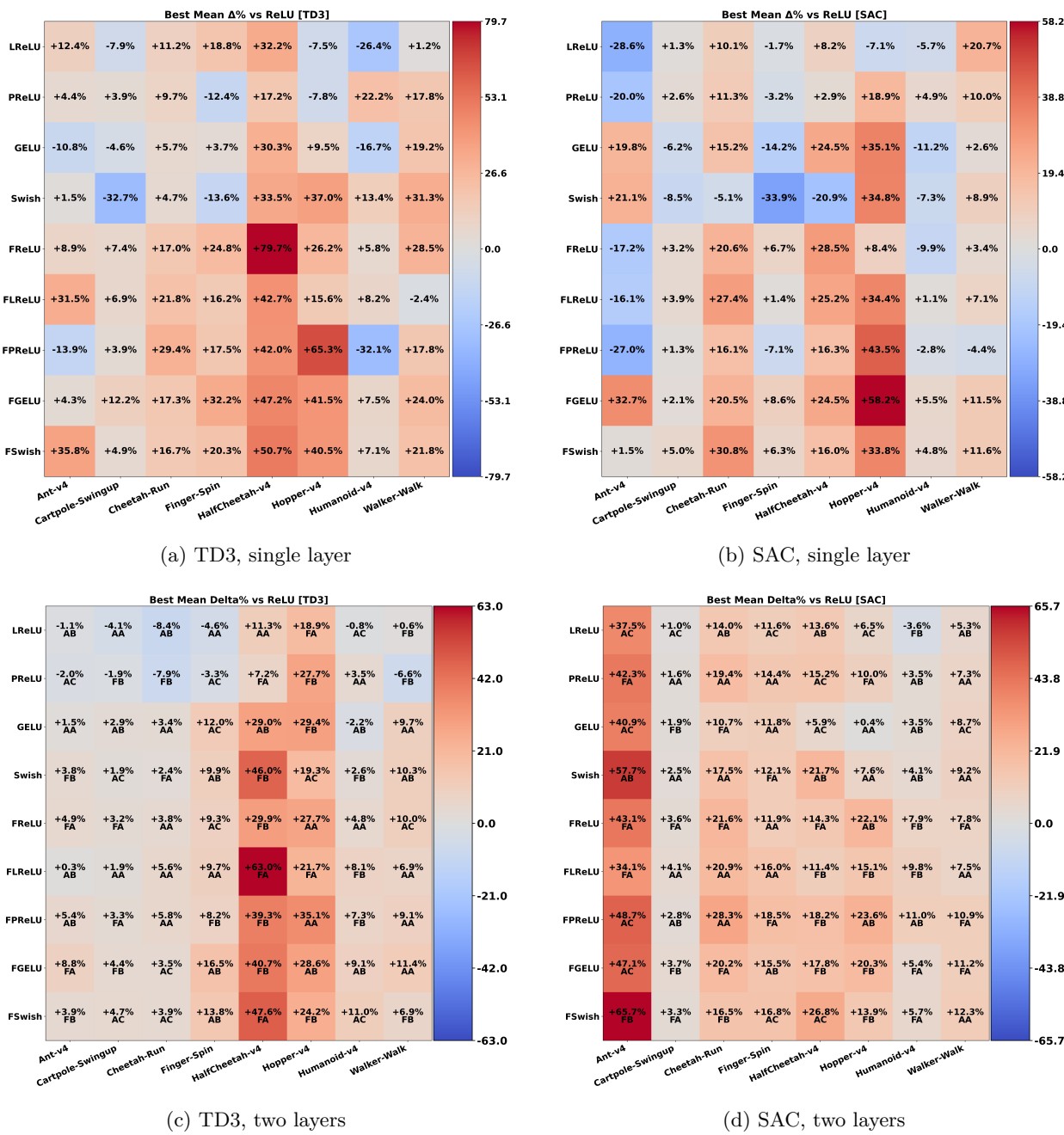

Figure 6: Relative performance ($\Delta\%$ AUC vs. the ReLU baseline) across TD3 and SAC for the single-layer and two-layer architectures. Each cell reports the best result for the corresponding activation and task, using the best $\alpha$ where applicable. Positive values indicate improvement over ReLU, while negative values indicate lower performance. For two-layer networks, results use the best activation-placement configuration.

tasks. However, their behavior is more task-dependent in the single-layer setting. In contrast, the fractional smooth variants, FGELU and FSwish, more consistently improve the corresponding learning curves, suggesting that fractional modulation can strengthen already smooth activation families rather than only improving rectifier-based functions.

Figure 6 provides a compact AUC-based comparison across TD3 and SAC for both network depths. Each heatmap cell reports the best mean improvement relative to the ReLU baseline for a given activation–task

Table 1: Best activation configuration for each task under TD3 and SAC. The first row reports the single-layer architecture, and the second row reports the two-layer architecture.

(a) TD3, one layer

| Task | Best Activation | Best $\alpha$ | $\Delta\%$ |
|---|---|---|---|
| Ant-v4 | FSwish | 0.3 | +35.75 |
| Cartpole-Swingup | FGELU | 0.2 | +12.16 |
| Cheetah-Run | FPReLU | 0.3 | +29.43 |
| Finger-Spin | FGELU | 0.5 | +32.19 |
| HalfCheetah-v4 | FReLU | 0.3 | +79.67 |
| Hopper-v4 | FPReLU | 0.1 | +65.26 |
| Humanoid-v4 | PReLU | – | +22.17 |
| Walker-Walk | Swish | – | +31.28 |

(b) SAC, one layer

| Task | Best Activation | Best $\alpha$ | $\Delta\%$ |
|---|---|---|---|
| Ant-v4 | FGELU | 0.1 | +32.66 |
| Cartpole-Swingup | FSwish | 0.4 | +5.03 |
| Cheetah-Run | FSwish | 0.2 | +30.76 |
| Finger-Spin | FGELU | 0.5 | +8.65 |
| HalfCheetah-v4 | FReLU | 0.2 | +28.54 |
| Hopper-v4 | FGELU | 0.1 | +58.23 |
| Humanoid-v4 | FGELU | 0.1 | +5.45 |
| Walker-Walk | LReLU | – | +20.65 |

(c) TD3, two layers

| Task | Best Activation | Best $\alpha$ | Placement | $\Delta\%$ |
|---|---|---|---|---|
| Ant-v4 | FGELU | 0.4 | first-actor | +8.79 |
| Cartpole-Swingup | FSwish | 0.4 | all-critic | +4.68 |
| Cheetah-Run | FPReLU | 0.4 | all-actor | +5.79 |
| Finger-Spin | FGELU | 0.1 | all-both | +16.48 |
| HalfCheetah-v4 | FLReLU | 0.1 | first-actor | +63.04 |
| Hopper-v4 | FPReLU | 0.1 | all-actor | +35.09 |
| Humanoid-v4 | FSwish | 0.5 | all-critic | +10.98 |
| Walker-Walk | FGELU | 0.3 | all-actor | +11.43 |

(d) SAC, two layers

| Task | Best Activation | Best $\alpha$ | Placement | $\Delta\%$ |
|---|---|---|---|---|
| Ant-v4 | FSwish | 0.5 | first-both | +65.67 |
| Cartpole-Swingup | FLReLU | 0.3 | all-actor | +4.08 |
| Cheetah-Run | FPReLU | 0.3 | all-actor | +28.34 |
| Finger-Spin | FPReLU | 0.3 | first-actor | +18.49 |
| HalfCheetah-v4 | FSwish | 0.1 | all-critic | +26.76 |
| Hopper-v4 | FPReLU | 0.2 | all-both | +23.65 |
| Humanoid-v4 | FPReLU | 0.3 | all-both | +10.96 |
| Walker-Walk | FSwish | 0.5 | all-actor | +12.30 |

pair. For the two-layer architecture, each cell also reports the placement strategy selected for the best-performing configuration. The heatmaps, therefore, summarize the combined influence of activation family, fractional order, and placement strategy.

The heatmaps show that smooth activations provide strong but task-dependent baselines. GELU and Swish improve over ReLU in several settings, especially in the two-layer architecture, but they are not uniformly better across all tasks and algorithms. This is particularly visible in the single-layer results, where smooth baselines can be competitive in some environments while underperforming ReLU in others. Therefore, the gains observed for fractional activations cannot be explained simply by replacing ReLU with a smoother nonlinearity.

In contrast, the fractional variants show a more consistent pattern of improvement across the heatmaps. This trend is especially clear for the smooth fractional activations, FGELU and FSwish, which remain competitive across both algorithms and network depths and frequently outperform their corresponding smooth baselines. These results indicate that fractional modulation can provide an additional benefit beyond smooth gated activations, rather than merely improving rectifier-style nonlinearities.

The heatmaps also show that the benefit of fractional activation design is not concentrated in a single activation family. Strong configurations are observed across FReLU, FLReLU, FPReLU, FGELU, and FSwish, suggesting that the fractional transformation serves as a broader activation-design mechanism. The two-layer results further show that activation placement matters, since the best configurations often depend on whether the fractional activation is applied to the actor, critic, both networks, or only the first layer.

Table 1 identifies the best activation configuration for each task. In the single-layer setting, fractional activations are selected in most tasks, although some non-fractional smooth activations remain competitive. This shows that the improvements are not simply a consequence of replacing ReLU with a smooth or curved activation. Even within an expanded and more competitive activation set, fractional variants are frequently selected as the strongest configurations, indicating that the fractional transformation provides an additional benefit beyond the base activation family.

The two-layer results show a clearer trend. Fractional activations are selected as the best configuration for all tasks under both TD3 and SAC. The selected activations include both rectifier-based variants, such as

FLReLU and FPReLU, and smooth variants, such as FGELU and FSwish. This result shows that fractional activations can become the preferred choice even in a competitive activation set that includes conventional rectifiers, modern smooth activations, and smooth fractional variants. In particular, FGELU and FSwish frequently appear among the top two-layer configurations, indicating that fractional modulation can improve smooth gated nonlinearities as well as ReLU-style activations.

Across these comparisons, the main finding is that smooth activations serve as strong, task-dependent baselines, whereas fractional variants yield a more consistent pattern of improvement across algorithms and network depths. The strongest configurations are not concentrated in a single activation family. Instead, they include FReLU, FLReLU, FPReLU, FGELU, and FSwish, suggesting that fractionalization serves as a more general activation-design mechanism rather than a correction specific to rectifier-based functions.

## 5.2 Alpha Sensitivity and Robustness

This section analyzes the sensitivity of fractional activation functions to the fractional-order parameter $\alpha$. Since GELU and Swish do not include a fractional-order parameter, this analysis focuses on the five fractional activation families: FReLU, FLReLU, FPReLU, FGELU, and FSwish. We evaluate $\alpha \in \{0.1, 0.2, 0.3, 0.4, 0.5\}$ and measure the resulting change in AUC relative to the ReLU baseline.

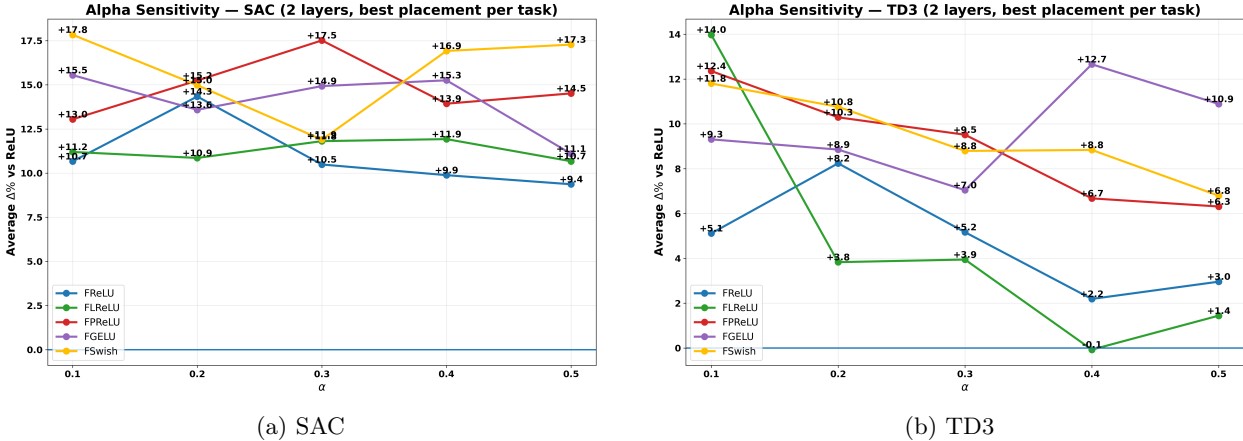

(a) SAC                                          (b) TD3

Figure 7: Sensitivity of performance to the fractional order $\alpha$ in the two-layer architecture. Curves show the average $\Delta\%$ AUC relative to the ReLU baseline across tasks. For each activation and $\alpha$, the best placement is selected per task.

Figure 7 shows that fractional activations remain beneficial across a broad range of $\alpha$ values, but the preferred value is not universal. This is expected because $\alpha$ controls the strength of the fractional transformation, and different activation families may require different degrees of deviation from their base nonlinearities.

For SAC, all five fractional activation families maintain positive average improvements across the full tested range. This indicates that SAC is relatively robust to the choice of $\alpha$. Smooth fractional variants are especially stable, with FGELU and FSwish remaining competitive across both small and larger fractional orders. FPReLU also achieves strong performance at intermediate values, suggesting that SAC can benefit from a wider range of fractional transformations when combined with optimized activation placement.

TD3 shows stronger sensitivity to $\alpha$. Rectifier-based variants generally prefer smaller fractional orders, particularly $\alpha = 0.1$ and $\alpha = 0.2$. This suggests that, for deterministic actor–critic learning, mild fractional transformations are often sufficient and may avoid overly strong changes to the activation shape. In contrast, FGELU reaches its strongest average performance at a larger value of $\alpha$, indicating that smooth fractional activations can tolerate, and sometimes benefit from, stronger fractional modulation.

To further examine preferred values of $\alpha$, Table 2 reports how often each fractional order is selected as the best-performing value across tasks. For the two-layer setting, the best placement is selected for each task, activation family, and $\alpha$.

Table 2: Best $\alpha$ frequency across tasks for fractional activations. Each cell reports the number of tasks, out of 8, for which a given $\alpha$ is selected as best. For the two-layer setting, the best placement is selected for each task, activation, and $\alpha$. A value of 0 indicates that the corresponding $\alpha$ was not selected for any task.

(a) SAC

| Activation | 1 layer | | | | | 2 layers | | | | |
|---|---|---|---|---|---|---|---|---|---|---|
| | 0.1 | 0.2 | 0.3 | 0.4 | 0.5 | 0.1 | 0.2 | 0.3 | 0.4 | 0.5 |
| FReLU | 2 | 4 | 2 | 0 | 0 | 2 | 3 | 0 | 0 | 3 |
| FLReLU | 3 | 2 | 0 | 3 | 0 | 2 | 0 | 4 | 1 | 1 |
| FPReLU | 3 | 2 | 2 | 0 | 1 | 0 | 2 | 4 | 0 | 2 |
| FGELU | 4 | 2 | 0 | 1 | 1 | 3 | 1 | 1 | 3 | 0 |
| FSwish | 3 | 2 | 0 | 2 | 1 | 3 | 0 | 1 | 0 | 4 |

(b) TD3

| Activation | 1 layer | | | | | 2 layers | | | | |
|---|---|---|---|---|---|---|---|---|---|---|
| | 0.1 | 0.2 | 0.3 | 0.4 | 0.5 | 0.1 | 0.2 | 0.3 | 0.4 | 0.5 |
| FReLU | 3 | 1 | 3 | 1 | 0 | 3 | 1 | 1 | 1 | 2 |
| FLReLU | 0 | 3 | 2 | 2 | 1 | 6 | 1 | 0 | 0 | 1 |
| FPReLU | 3 | 3 | 2 | 0 | 0 | 4 | 1 | 1 | 1 | 1 |
| FGELU | 0 | 1 | 2 | 1 | 4 | 1 | 1 | 1 | 4 | 1 |
| FSwish | 2 | 0 | 2 | 2 | 2 | 2 | 1 | 2 | 1 | 2 |

The frequency results support the trends observed in Figure 7. Smaller values of $\alpha$ are often selected for rectifier-based variants, especially under TD3. For example, in the two-layer TD3 setting, FLReLU and FPReLU most frequently select $\alpha = 0.1$. However, the preferred order is more diverse for SAC and for the smooth fractional variants. In the two-layer SAC setting, FPReLU and FLReLU often favor $\alpha = 0.3$, while FSwish most frequently selects $\alpha = 0.5$. FGELU also shows a broader distribution of preferred values, particularly across the two-layer settings.

These results show that $\alpha$ should not be interpreted as a parameter for which larger values are always better. Instead, it acts as a lightweight activation-specific hyperparameter that controls the strength of the fractional shape transformation. Small values are often effective for rectifier-based activations and TD3, while smooth fractional activations can benefit from a wider range of values. Overall, the tested range $\alpha \in [0.1, 0.5]$ provides stable and useful improvements across the evaluated tasks, with the best value depending on the activation family and algorithm.

## 5.3 Activation Functions and Placement Effects

This section examines the interaction between activation-family choice and activation placement in two-layer actor–critic networks. For each activation family and each algorithm–task pair, we select the best-performing placement and, for fractional activations, the best value of $\alpha$. Detailed task-level placement results are reported in Appendix C.4.

Table 3: Summary of activation performance in the two-layer placement experiments. For each activation family, we report the average improvement relative to the ReLU baseline across the 16 algorithm–task pairs, the number of pairs where the activation outperforms ReLU, and the maximum observed improvement.

| Activation | Avg $\Delta\%$ | Tasks > ReLU | Max $\Delta\%$ |
|---|---|---|---|
| LReLU | +6.1 | 10 / 16 | +37.54 |
| PReLU | +8.1 | 11 / 16 | +42.33 |
| GELU | +10.6 | 15 / 16 | +40.85 |
| Swish | +14.3 | 16 / 16 | +57.69 |
| FReLU | +14.1 | 16 / 16 | +43.08 |
| FLReLU | +14.8 | 16 / 16 | +63.04 |
| FPReLU | +17.2 | 16 / 16 | +48.72 |
| FGELU | +16.5 | 16 / 16 | +47.05 |
| FSwish | +17.3 | 16 / 16 | +65.67 |

Table 3 shows that smooth activations are strong baselines for off-policy actor–critic learning. GELU improves over ReLU in 15 out of 16 algorithm–task pairs, while Swish improves over ReLU in all 16 pairs. However, the fractional variants achieve stronger and more consistent performance. All five fractional acti-

vations improve over ReLU in all 16 pairs, and the largest average gains are obtained by FSwish, FPReLU, and FGELU.

The comparison also shows that the benefit of fractionalization is not limited to weaker rectifier baselines. FLReLU improves substantially over LReLU, and FPReLU improves over PReLU, even though PReLU is already adaptive. The same pattern appears for smooth activations, where FGELU improves over GELU, and FSwish improves over Swish on average. These results indicate that fractional activation design provides an additional benefit across rectifier-based, adaptive, and smooth activation families.

Table 4: Frequency of fractional activation placement appearing as the best configuration in the two-layer experiments. Counts are aggregated across 5 fractional activation families and 8 tasks for each algorithm, giving 40 comparisons per algorithm.

| Placement Strategy | SAC (2-layer) | TD3 (2-layer) |
|---|---|---|
| all-actor | 7 | 11 |
| first-actor | 13 | 7 |
| all-critic | 4 | 6 |
| all-both | 5 | 6 |
| first-both | 11 | 10 |

Table 4 summarizes how often each placement strategy is selected as the best configuration for the five fractional activation families in the two-layer experiments. Each algorithm contributes $5 \times 8 = 40$ placement selections.

The placement results show that fractional activations are most often beneficial when they affect the actor network or the early representation layers. Under SAC, `first-actor` and `first-both` are selected most frequently, while the purely critic-focused `all-critic` placement is selected much less often. Under TD3, the pattern is slightly more balanced, but `all-actor`, `first-actor`, and `first-both` still dominate the selections. This suggests that fractional activations primarily help by shaping the policy representation, while early placement in both the actor and the critic can also be useful when value estimation and policy improvement are tightly coupled.

These results show that both activation family and placement influence performance. Fractional activations are not only strong as function choices, but are also sensitive to where they are introduced in the actor–critic architecture. The dominant trend is that actor-side or early-layer placement is more effective than applying fractional activations only to the critic.

### 5.4 Statistical Validation

To strengthen the evaluation, we statistically compare the best fractional activation configuration against the matched ReLU baseline. The comparison is performed using seed-level normalized AUC values. For each algorithm and network depth, Table 5 reports the number of tasks with positive improvement, the number of tasks whose bootstrap confidence interval is strictly positive, the mean and median normalized AUC improvement, the median paired Wilcoxon signed-rank test $p$-value, the sign-test $p$-value, and the median Cliff's $\delta$ effect size across tasks.

The paired Wilcoxon signed-rank test is computed separately for each task using the five seed-level AUC values. Since each task has only five seeds, these per-task tests have limited statistical power. We therefore also report a sign test across tasks to evaluate whether improvements occur consistently across the benchmark suite. Cliff's $\delta$ is included to measure the practical size of the improvement, while the bootstrap confidence interval provides an additional seed-level estimate of whether the paired AUC difference is reliably positive.

Table 5 shows that the best fractional activation configurations improve normalized AUC in every task-level comparison across both algorithms and both network depths. In all four settings, fractional activations improve all 8 tasks. The sign-test results support this directional consistency, with $p = 0.0039$ in each

Table 5: Aggregate statistical validation comparing the best fractional activation configuration against the ReLU baseline. Pos. reports the number of tasks where the fractional configuration achieves higher normalized AUC than ReLU. CI> 0 reports the number of tasks whose bootstrap confidence interval for the paired AUC difference is strictly positive. Med. $p_{\mathrm{W}}$, $p_{\mathrm{sign}}$, and Med. $\delta$ denote the median paired Wilcoxon $p$-value, the sign-test $p$-value, and the median Cliff's $\delta$ effect size, respectively.

| Alg. | Depth | Pos. | CI> 0 | Mean $\Delta$ | Median $\Delta$ | Med. $p_{\mathrm{W}}$ | $p_{\mathrm{sign}}$ | Med. $\delta$ | Trend |
|------|-------|------|-------|---------------|-----------------|----------------------|--------------------|---------------|-------|
| SAC | 1 layer | 8/8 | 7/8 | +22.6% | +20.1% | 0.1250 | 0.0039 | 0.68 | Strong practical improvement |
| SAC | 2 layers | 8/8 | 8/8 | +23.8% | +21.1% | 0.0625 | 0.0039 | 0.80 | Strong consistent improvement |
| TD3 | 1 layer | 8/8 | 4/8 | +36.4% | +30.8% | 0.2500 | 0.0039 | 0.72 | Large practical gain |
| TD3 | 2 layers | 8/8 | 5/8 | +19.5% | +11.2% | 0.1250 | 0.0039 | 0.60 | Consistent practical improvement |

setting. This indicates that the improvements are not concentrated in a small subset of environments, but occur consistently across the evaluated benchmark suite.

The strongest seed-level evidence is observed for SAC, especially in the two-layer architecture. In this setting, every task improves, every bootstrap confidence interval is strictly positive, and the median Cliff's $\delta$ indicates a large practical effect. SAC also shows strong evidence in the single-layer architecture, where most bootstrap confidence intervals are strictly positive and the median effect size remains large. These results suggest that fractional activations provide stable benefits for SAC across both shallow and deeper actor–critic networks.

TD3 also improves across all tasks, although its seed-level evidence is more variable. The one-layer TD3 setting gives the largest average improvement, indicating strong practical gains. However, fewer bootstrap confidence intervals are strictly positive compared with SAC, and the median Wilcoxon $p$-value is less decisive. This pattern reflects greater seed-level variability in TD3 rather than a lack of improvement, since the sign test remains significant and the median effect sizes still indicate practically meaningful gains. The two-layer TD3 setting follows the same general trend, with positive improvements across all tasks and medium-to-large practical effects.

Taken together, these results provide directional, statistical, and practical support for the proposed fractional activations. The sign tests show that improvements are consistent across tasks, while Cliff's $\delta$ indicates meaningful effect sizes. Seed-level statistical evidence is strongest for SAC, whereas TD3 shows larger variability but still improves consistently across environments. Full per-task statistical results, including bootstrap confidence intervals, paired Wilcoxon signed-rank tests, Cliff's $\delta$ effect sizes, selected activation configurations, fractional orders, and activation placements, are provided in Appendix C.1.

## 6 Conclusion

This paper examined five fractional activation functions for off-policy deep RL, covering FReLU, FLReLU, FPReLU, FSwish, and FGELU. These activations span rectifier-based and smooth activation families, enabling a controlled evaluation of whether fractional transformations remain useful beyond ReLU-style non-linearities. We integrated the activations into TD3 and SAC and evaluated them on continuous-control benchmarks from MuJoCo and the DeepMind Control Suite. The evaluation compared each fractional activation with its corresponding non-fractional baseline, including ReLU, LReLU, PReLU, Swish, and GELU.

The results show that fractional activations frequently improve normalized AUC across tasks, algorithms, and network depths. The evaluation further shows that the benefits are not limited to rectifier-based activations, as FGELU and FSwish often outperform their corresponding smooth baselines. Statistical validation using paired seed-level AUC comparisons, bootstrap confidence intervals, Wilcoxon signed-rank tests, Cliff's $\delta$, and aggregate sign tests supports the consistency and practical relevance of the observed improvements.

The analysis also shows that the fractional order $\alpha$ should be treated as a lightweight activation-specific hyperparameter rather than as a parameter to be maximized. Small and moderate fractional orders often provide the most reliable gains, while activation placement in deeper actor–critic networks influences performance. Overall, these findings suggest that fractional activation functions offer a simple, computationally lightweight design choice to improve empirical performance in off-policy actor–critic RL.

Future work will study fractional activations in broader RL settings, including offline RL, non-Gaussian policy classes, image-based agents, and larger network architectures. Further analysis of gradient behavior, value-function approximation, and policy representation may also help explain when and why fractional activations are most beneficial.

## Broader Impact Statement

This work investigates fractional-order activation functions for RL in continuous-control tasks. Improvements in activation design may enhance the learning efficiency and representational capacity of neural networks, with potential benefits for applications in robotics, control systems, and autonomous decision-making. At the same time, RL methods may be deployed in sensitive domains, including autonomous systems, where responsible development and appropriate oversight are essential. This research does not involve human subjects or personal data, and all experiments are conducted in standard simulated benchmark environments.

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

## Appendix

This appendix provides supplementary material supporting the methodology, experimental setup, and empirical results. It summarizes the evaluated activation functions, presents the benchmark environments, and reports additional analyses, including statistical validation, fractional-order sensitivity, task-level preferred orders, complete AUC tables, and best-configuration results.

## A    Activation Function Summary

This section summarizes the activation functions evaluated in this study. Table 6 groups the non-fractional baselines and fractional variants into rectifier baselines, smooth baselines, fractional rectifier activations, and fractional smooth activations. It also reports whether each activation uses a fractional order, whether it includes learnable parameters, and how it is used in the controlled comparison.

Table 6: Summary of baseline and fractional activation functions evaluated in this study. The full mathematical definitions of the fractional variants are provided in Eqs. 9–11, 17, and 19.

| Family | Activation | Fractional order | Learnable parameter | Role in the study |
|---|---|---|---|---|
| Rectifier baseline | ReLU | No | No | Standard rectifier baseline used in TD3 and SAC. |
| Rectifier baseline | LReLU | No | No, fixed negative slope $k$ | Leaky rectifier baseline with a fixed non-zero negative-side response. |
| Rectifier baseline | PReLU | No | Learnable negative coefficient $p$ | Parametric rectifier baseline with adaptive negative-side response. |
| Smooth baseline | GELU | No | No | Smooth probabilistic-gated activation used as a stronger non-rectifier baseline. |
| Smooth baseline | Swish | No | No | Smooth self-gated activation used as a stronger non-rectifier baseline. |
| Fractional rectifier | FReLU | Fixed $\alpha$ | No | Positive-only fractional rectifier used to test fractional curvature relative to ReLU. |
| Fractional rectifier | FLReLU | Fixed $\alpha$ | No, fixed negative slope $k$ | Two-sided fractional rectifier used to test fixed leakage relative to LReLU. |
| Fractional rectifier | FPReLU | Fixed $\alpha$ | Learnable bounded negative coefficient $p_c$ | Two-sided fractional parametric rectifier used to test adaptive negative-side response relative to PReLU. |
| Fractional smooth | FGELU | Fixed $\alpha$ | Learnable bounded $\beta_c$ | Fractional smooth activation based on an adaptive GELU core, used to test fractional modulation relative to GELU. |
| Fractional smooth | FSwish | Fixed $\alpha$ | Learnable bounded $\beta_c$ | Fractional smooth activation based on an adaptive Swish core, used to test fractional modulation relative to Swish. |

Here $k$ denotes the fixed LReLU and FLReLU negative slope, set to 0.1 in this study; $p_c$ denotes the bounded learnable negative coefficient; and $\beta_c = \text{clip}(\text{softplus}(\beta_{\text{raw}}), 0.1, 10.0)$.

## B    Benchmark Tasks

This section illustrates the benchmark environments used in the experimental evaluation. Figure 8 shows representative tasks from the MuJoCo and DeepMind Control Suite benchmarks. These environments cover a range of continuous-control challenges, including locomotion, balance, running, manipulation, and swing-up

control. This diversity enables the evaluation to assess whether fractional activation functions yield benefits across diverse state-action structures, reward landscapes, and control objectives.

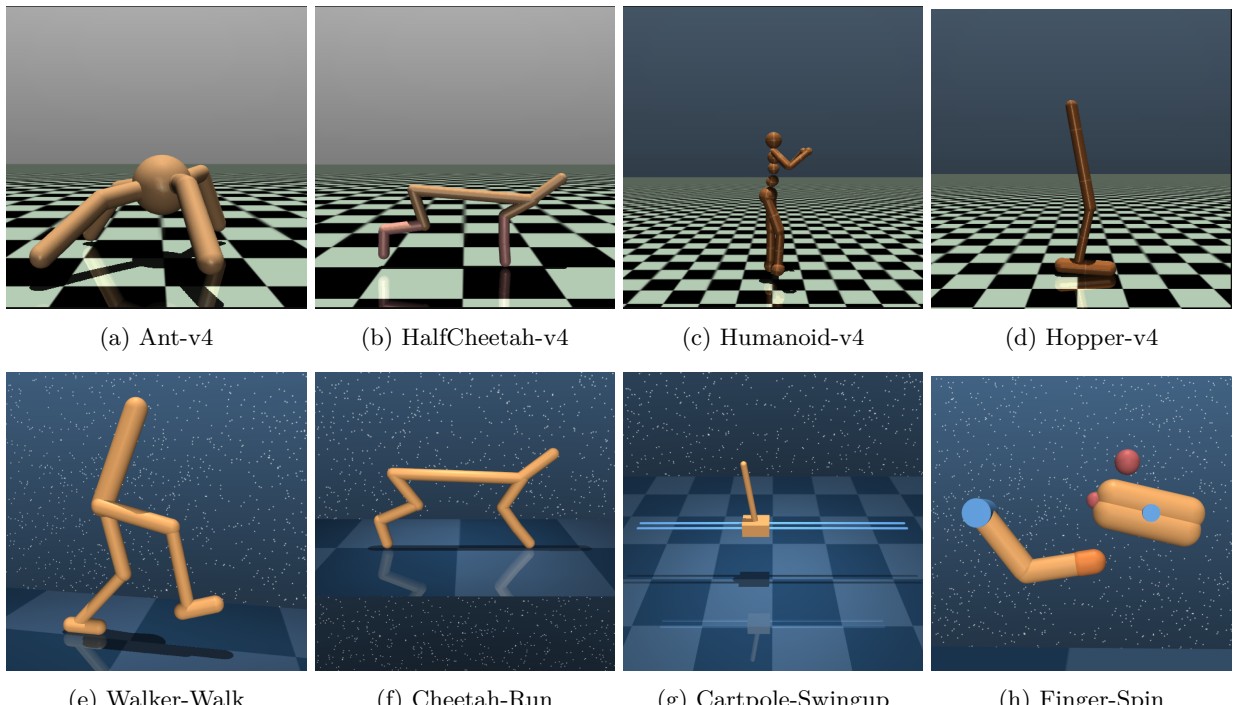

(a) Ant-v4      (b) HalfCheetah-v4      (c) Humanoid-v4      (d) Hopper-v4

(e) Walker-Walk      (f) Cheetah-Run      (g) Cartpole-Swingup      (h) Finger-Spin

Figure 8: Continuous-control benchmark tasks used in the experiments. The first row shows MuJoCo environments accessed through OpenAI Gym, while the second row shows tasks from the DeepMind Control Suite.

## C  Additional Experimental Results

This appendix provides additional experimental material supporting the aggregate results reported in the main paper. It includes the full per-task statistical validation, extended sensitivity analyses for the fractional-order parameter, task-level preferred fractional orders, task-level activation-placement behavior, complete AUC results, and detailed best-configuration tables. These results provide a more detailed view of how activation design affects performance across tasks, algorithms, network depths, fractional orders, and activation-placement strategies.

The appendix is organized as follows. Appendix C.1 reports the full per-task statistical validation. Appendix C.2 presents additional sensitivity analyses for the fractional order parameter $\alpha$. Appendix C.3 reports the task-level preferred fractional orders. Appendix C.4 analyzes task-level activation-placement behavior in the two-layer architecture. Appendix C.5 provides the complete AUC tables. Finally, Appendix C.6 reports the best configuration for each activation and task.

### C.1  Per-Task Statistical Validation

This section provides the full per-task statistical validation supporting the aggregate summary in Table 5. For each task, algorithm, and network depth, the best fractional activation configuration is compared against the matched ReLU baseline using seed-level normalized AUC values. The best configuration is selected from FReLU, FLReLU, FPReLU, FSwish, and FGELU.

Table 7 reports the normalized AUC improvement over ReLU, the bootstrap confidence interval for the paired AUC difference, the paired Wilcoxon signed-rank test $p$-value, Cliff's $\delta$ effect size, the selected fractional

Table 7: Per-task statistical validation for the best fractional activation configuration relative to the ReLU baseline. $\Delta$ denotes the normalized AUC improvement over ReLU. CI denotes the bootstrap confidence interval of the paired AUC difference across seeds, reported in the original AUC-difference scale. The Wilcoxon signed-rank test and Cliff's $\delta$ are computed from matched seed-level AUC values.

| Alg. | Depth | Task | Best config. | $\alpha$ | Placement | $\Delta$ | CI | $p_{\mathrm{W}}$ | Cliff's $\delta$ |
|---|---|---|---|---|---|---|---|---|---|
| SAC | 1 layer | Ant-v4 | FGELU | 0.1 | both | +32.7% | [1.76e+08, 7.06e+08] | 0.1250 | 0.60 |
| SAC | 1 layer | Cartpole-Swingup | FSwish | 0.4 | both | +5.0% | [1.96e+07, 5.57e+07] | 0.0625 | 0.92 |
| SAC | 1 layer | Cheetah-Run | FSwish | 0.2 | both | +30.8% | [3.52e+07, 1.15e+08] | 0.1250 | 0.84 |
| SAC | 1 layer | Finger-Spin | FGELU | 0.5 | both | +8.6% | [2.34e+07, 8.39e+07] | 0.0625 | 0.76 |
| SAC | 1 layer | HalfCheetah-v4 | FReLU | 0.2 | both | +28.5% | [1.91e+08, 2.03e+09] | 0.1875 | 0.52 |
| SAC | 1 layer | Hopper-v4 | FGELU | 0.1 | both | +58.2% | [3.95e+08, 1.26e+09] | 0.1250 | 0.84 |
| SAC | 1 layer | Humanoid-v4 | FGELU | 0.1 | both | +5.5% | [6.79e+07, 2.97e+08] | 0.0625 | 0.36 |
| SAC | 1 layer | Walker-Walk | FSwish | 0.2 | both | +11.6% | [-5.79e+07, 1.91e+08] | 0.6250 | 0.36 |
| SAC | 2 layers | Ant-v4 | FSwish | 0.5 | first-both | +65.7% | [8.95e+08, 2.08e+09] | 0.0625 | 1.00 |
| SAC | 2 layers | Cartpole-Swingup | FLReLU | 0.3 | all-actor | +4.1% | [8.20e+06, 5.42e+07] | 0.1250 | 0.68 |
| SAC | 2 layers | Cheetah-Run | FPReLU | 0.3 | all-actor | +28.3% | [8.99e+07, 1.57e+08] | 0.0625 | 1.00 |
| SAC | 2 layers | Finger-Spin | FPReLU | 0.3 | first-actor | +18.5% | [5.97e+07, 2.02e+08] | 0.0625 | 0.76 |
| SAC | 2 layers | HalfCheetah-v4 | FSwish | 0.1 | all-critic | +26.8% | [1.29e+09, 2.38e+09] | 0.0625 | 0.84 |
| SAC | 2 layers | Hopper-v4 | FPReLU | 0.2 | all-both | +23.6% | [3.94e+08, 7.79e+08] | 0.0625 | 0.76 |
| SAC | 2 layers | Humanoid-v4 | FPReLU | 0.3 | all-both | +11.0% | [2.51e+08, 6.45e+08] | 0.0625 | 0.92 |
| SAC | 2 layers | Walker-Walk | FSwish | 0.5 | all-actor | +12.3% | [1.77e+07, 1.73e+08] | 0.0625 | 0.76 |
| TD3 | 1 layer | Ant-v4 | FSwish | 0.3 | both | +35.8% | [-3.16e+08, 1.31e+09] | 0.3125 | 0.60 |
| TD3 | 1 layer | Cartpole-Swingup | FGELU | 0.2 | both | +12.2% | [-9.52e+06, 1.69e+08] | 0.3125 | 0.76 |
| TD3 | 1 layer | Cheetah-Run | FPReLU | 0.3 | both | +29.4% | [3.50e+07, 1.28e+08] | 0.0625 | 0.92 |
| TD3 | 1 layer | Finger-Spin | FGELU | 0.5 | both | +32.2% | [8.50e+05, 3.82e+08] | 0.1875 | 0.68 |
| TD3 | 1 layer | HalfCheetah-v4 | FReLU | 0.3 | both | +79.7% | [1.12e+09, 2.53e+09] | 0.0625 | 1.00 |
| TD3 | 1 layer | Hopper-v4 | FPReLU | 0.1 | both | +65.3% | [3.97e+08, 1.03e+09] | 0.0625 | 0.92 |
| TD3 | 1 layer | Humanoid-v4 | FLReLU | 0.2 | both | +8.2% | [-7.08e+08, 1.07e+09] | 0.8125 | 0.20 |
| TD3 | 1 layer | Walker-Walk | FReLU | 0.1 | both | +28.5% | [-2.98e+07, 3.61e+08] | 0.3125 | 0.52 |
| TD3 | 2 layers | Ant-v4 | FGELU | 0.4 | first-actor | +8.8% | [7.80e+07, 4.38e+08] | 0.0625 | 0.52 |
| TD3 | 2 layers | Cartpole-Swingup | FSwish | 0.4 | all-critic | +4.7% | [9.92e+06, 6.06e+07] | 0.1250 | 0.84 |
| TD3 | 2 layers | Cheetah-Run | FPReLU | 0.4 | all-actor | +5.8% | [-8.28e+07, 1.18e+08] | 0.8125 | 0.12 |
| TD3 | 2 layers | Finger-Spin | FGELU | 0.1 | all-both | +16.5% | [5.23e+07, 2.06e+08] | 0.0625 | 1.00 |
| TD3 | 2 layers | HalfCheetah-v4 | FLReLU | 0.1 | first-actor | +63.0% | [3.56e+08, 3.65e+09] | 0.1250 | 0.60 |
| TD3 | 2 layers | Hopper-v4 | FPReLU | 0.1 | all-actor | +35.1% | [-2.40e+07, 1.56e+09] | 0.3125 | 0.60 |
| TD3 | 2 layers | Humanoid-v4 | FSwish | 0.5 | all-critic | +11.0% | [-9.60e+07, 7.72e+08] | 0.1875 | 0.60 |
| TD3 | 2 layers | Walker-Walk | FGELU | 0.3 | all-actor | +11.4% | [6.55e+07, 1.07e+08] | 0.0625 | 0.76 |

activation, the fractional order $\alpha$, and the activation placement. The percentage improvement $\Delta$ is reported relative to the ReLU baseline, while the confidence interval is reported in the original paired AUC-difference scale.

The bootstrap confidence interval is computed from paired seed-level AUC differences. The Wilcoxon signed-rank test provides a non-parametric seed-level test for each task, while Cliff's $\delta$ measures the practical magnitude of the effect. Since each task uses five seeds, individual per-task tests have limited statistical power and should be interpreted together with the aggregate sign-test results reported in the main text.

Overall, the per-task results confirm that the updated fractional activation set yields positive AUC improvements across all task-level comparisons. The strongest seed-level consistency is observed for SAC, particularly in the two-layer setting, while TD3 shows large practical gains in several tasks.

## C.2  Fractional Order Sensitivity

This section presents additional sensitivity analyses of the fractional-order parameter $\alpha$. The five fractional activation families, FReLU, FLReLU, FPReLU, FGELU, and FSwish are evaluated across $\alpha \in \{0.1, 0.2, 0.3, 0.4, 0.5\}$. Since ReLU, LReLU, PReLU, Swish, and GELU do not use a fractional order, they are not included in this analysis.

Figures 9 and 10 show task-wise sensitivity for SAC and TD3 under different two-layer placement strategies. For each placement, activation family, and value of $\alpha$, the plotted value is based on the strongest task-level improvement relative to the ReLU baseline. The curves are normalized within each activation family by the largest absolute improvement observed for that activation. Therefore, the vertical position of a curve

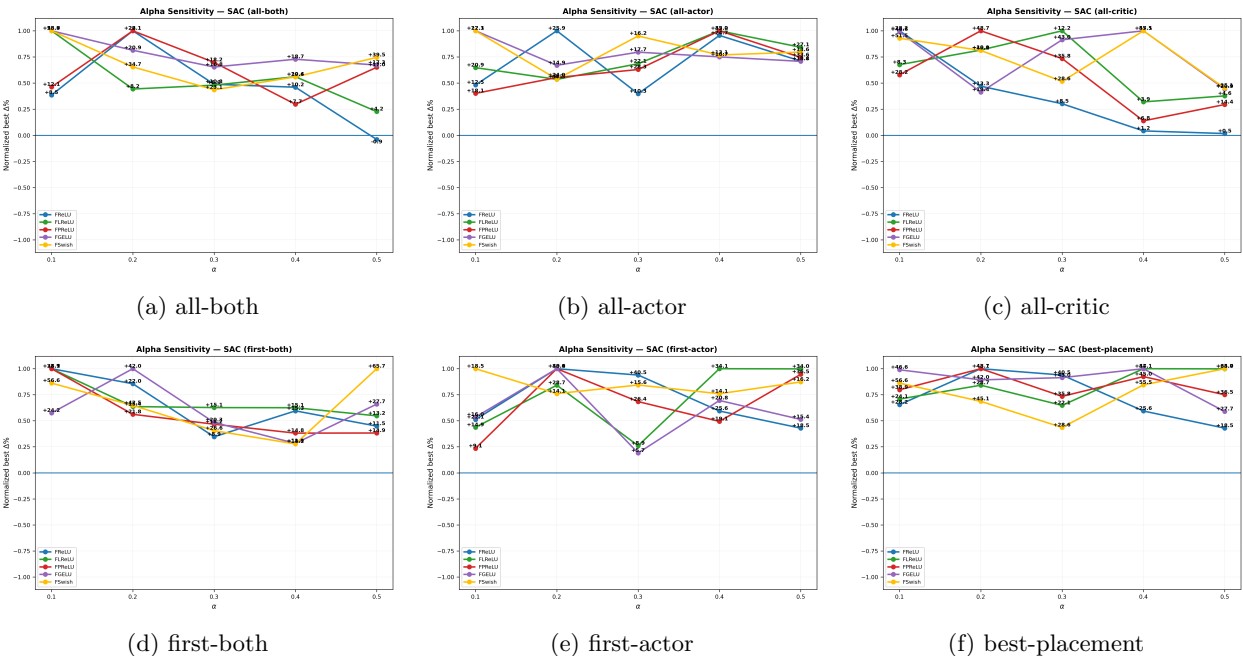

Figure 9: Task-wise fractional order sensitivity for SAC with two-layer networks across placement strategies. Each panel reports the best relative improvement over the ReLU baseline across tasks as a function of $\alpha$ for FReLU, FLReLU, FPReLU, FGELU, and FSwish. Values are normalised within each activation family, so vertical positions show relative sensitivity within each activation rather than absolute comparison across activations. Numeric labels report the original unnormalised $\Delta\%$ values.

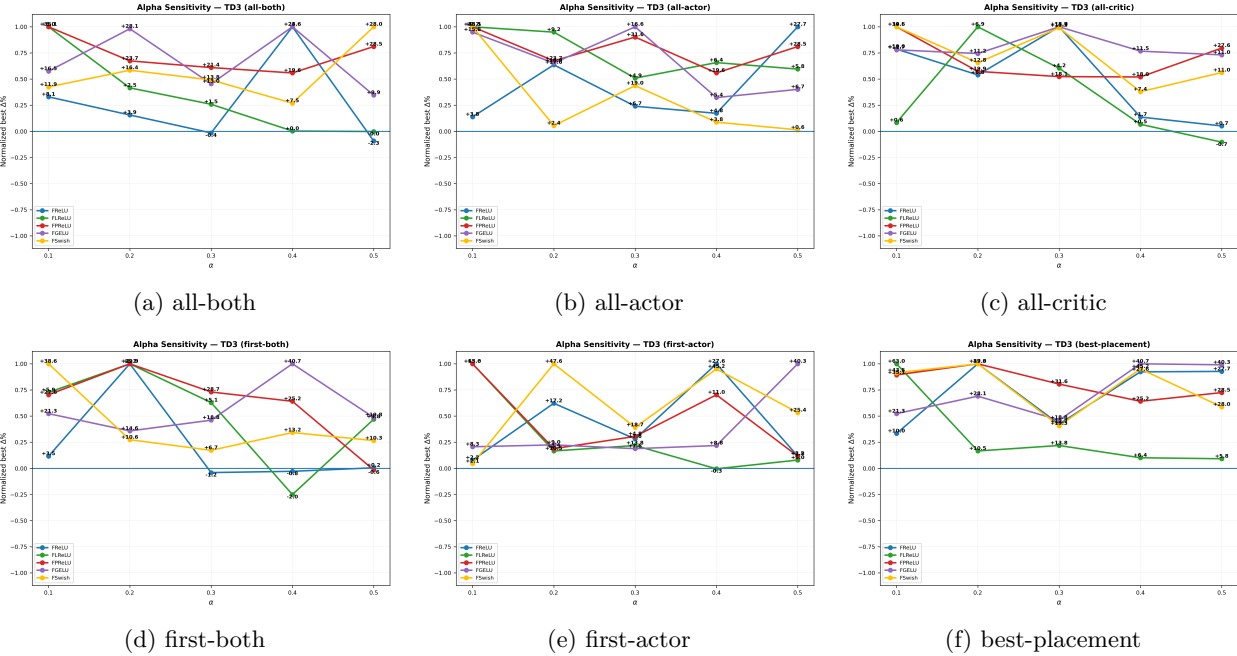

Figure 10: Task-wise fractional order sensitivity for TD3 with two-layer networks across placement strategies. Each panel reports the best relative improvement over the ReLU baseline across tasks as a function of $\alpha$ for FReLU, FLReLU, FPReLU, FGELU, and FSwish. Values are normalized within each activation family, so vertical positions show relative sensitivity within each activation rather than absolute comparison across activations. Numeric labels report the original unnormalized $\Delta\%$ values.

Table 8: Preferred fractional order $\alpha$ per task in the two-layer experiments. For each task and algorithm, the value corresponds to the most frequently selected $\alpha$ across the five fractional activation families. The count in parentheses indicates how many activation families selected that value.

| Task | SAC Preferred $\alpha$ | TD3 Preferred $\alpha$ |
|---|---|---|
| Ant-v4 | 0.2 (2/5) | 0.1 (3/5) |
| Cartpole-Swingup | 0.5 (3/5) | 0.4 (2/5) |
| Cheetah-Run | 0.1 (2/5) | 0.1 (2/5) |
| Finger-Spin | 0.3 (3/5) | 0.1 (3/5) |
| HalfCheetah-v4 | 0.1 (2/5) | 0.2 (3/5) |
| Hopper-v4 | 0.2 (2/5) | 0.1 (3/5) |
| Humanoid-v4 | 0.5 (2/5) | 0.2 (2/5) |
| Walker-Walk | 0.1 (3/5) | 0.1 (2/5) |

shows the relative sensitivity pattern within the same activation family, rather than an absolute ranking between different activation families. The numeric annotations report the original unnormalized $\Delta\%$ values and should be used when comparing the magnitude of improvements across activations.

The SAC results show that fractional activations can achieve strong improvements across a broad range of fractional orders and placement strategies. The strongest gains are not restricted to one activation family, which suggests that fractionalization can be useful for both rectifier-based and smooth nonlinearities. The smooth fractional activations, especially FGELU and FSwish, remain competitive across several placements, showing that the benefit of fractional transformations is not simply due to replacing ReLU with a smooth activation.

The TD3 results show stronger placement dependence. This is consistent with the deterministic policy update in TD3, where changes in the actor representation can directly influence policy improvement. In several cases, strong gains occur when the fractional activation is applied to the actor or to the first layer, suggesting that early nonlinear transformations can affect the quality of policy and value representations.

These sensitivity results reinforce the main conclusion that $\alpha$ should be treated as a lightweight activation-specific hyperparameter. Fractional activations are not effective because of a single fixed value of $\alpha$. Instead, their benefit comes from introducing a controllable nonlinear family whose curvature can be tuned according to activation type, task, algorithm, and placement. Small and moderate fractional orders are often effective, while larger values can also be beneficial in specific settings, particularly for smooth fractional activations.

### C.3   Task-Level Preferred Fractional Orders

Table 8 reports the preferred fractional order $\alpha$ for each task in the two-layer architecture. For each task and algorithm, the preferred value is computed as the most frequently selected $\alpha$ across the five fractional activation families. The count in parentheses indicates how many activation families selected that value. When counts are tied, the value with the higher mean improvement is selected.

The task-level results support the aggregate conclusions from the main text. Small fractional orders are frequently selected, particularly under TD3, where $\alpha = 0.1$ is preferred in several tasks. However, the preferred value is not identical across all environments. For example, some tasks favor larger fractional orders, especially under SAC and for smooth fractional variants. This confirms that the fractional order is task dependent and that a small search over $\alpha$ is useful.

Overall, these results show that there is no single universally optimal value of $\alpha$. Instead, the strongest behavior is obtained by searching over a compact range of fractional orders. This is consistent with the main result that values in the range $\alpha \in [0.1, 0.5]$ provide stable and useful improvements, with smaller values often preferred for rectifier-based variants and larger values sometimes beneficial for smooth fractional activations.

### C.4 Task-Level Placement Behavior

This section provides the task-level placement analysis for the two-layer fractional activation experiments. While Section 5.3 summarizes the overall placement trends, here we report how often each placement strategy is selected as the best configuration for each task. The analysis includes the five fractional activation families, FReLU, FLReLU, FPReLU, FGELU, and FSwish, under both TD3 and SAC. Therefore, each task has 10 selections in total, corresponding to five fractional activation families evaluated under two algorithms.

Table 9: Most frequent best placement per task across both algorithms and all five fractional activation families in the two-layer placement experiments. Each row sums to 10, corresponding to five fractional activation families evaluated under TD3 and SAC.

| Task | all-actor | first-actor | all-critic | all-both | first-both |
|---|---|---|---|---|---|
| Ant-v4 | 0 | 4 | 2 | 2 | 2 |
| Cartpole-Swingup | 2 | 4 | 1 | 1 | 2 |
| Cheetah-Run | 5 | 2 | 2 | 0 | 1 |
| Finger-Spin | 3 | 1 | 2 | 3 | 1 |
| HalfCheetah-v4 | 0 | 3 | 1 | 0 | 6 |
| Hopper-v4 | 2 | 1 | 0 | 3 | 4 |
| Humanoid-v4 | 1 | 2 | 1 | 2 | 4 |
| Walker-Walk | 5 | 3 | 1 | 0 | 1 |

Table 9 shows that placement behavior is task-dependent, but the dominant trend remains consistent with the main-text summary. Fractional activations are most often used when applied to the actor network or the first layers of the actor–critic architecture. Purely critic-focused placement is less frequently selected across most tasks.

For *Cheetah-Run* and *Walker-Walk*, the `all-actor` placement is selected most often, with five selections in each task. When combined with `first-actor`, actor-only placement accounts for most selections in these two environments. This suggests that, for these locomotion tasks, modifying the policy representation is particularly important.

A different pattern appears in *HalfCheetah-v4*, *Hopper-v4*, and *Humanoid-v4*, where `first-both` is frequently selected. This indicates that early fractional transformations in both the actor and critic can be beneficial in tasks where policy improvement and value estimation are closely coupled.

The *Ant-v4* and *Cartpole-Swingup* tasks show a stronger preference for `first-actor`. This suggests that applying fractional activations early in the actor network can provide a useful representational modification without requiring the same activation to be applied throughout the full actor network.

These task-level results support the main conclusion that fractional activation placement matters. The strongest configurations are rarely restricted to the critic alone. Instead, they usually involve actor-side placement or early placement across both networks, reinforcing the interpretation that fractional activations improve learning partly by shaping the policy representation.

### C.5 Complete AUC Results

This section reports the complete AUC results for all activation functions, algorithms, tasks, and architecture settings evaluated in the study. AUC values are computed over the full training horizon and aggregated across five random seeds using the interquartile mean (IQM). IQM is used because it provides a robust estimate of central performance while reducing sensitivity to unusually high or low seed outcomes.

Within each fractional activation family, FReLU, FLReLU, FPReLU, FSwish, and FGELU, the best-performing fractional order $\alpha$ is shown in **bold**. The overall best configuration for each task, including the ReLU baseline, is highlighted in purple. These complete tables are intended to make the experimental results transparent and to show the full pattern behind the summary results reported in the main text.

### C.5.1 Single-Layer Architecture

Table 10 reports the complete single-layer results for TD3 and SAC. In this architecture, each activation function is applied uniformly within the single hidden layer of the actor and critic networks. Therefore, the comparison isolates the effect of activation-function choice without introducing placement variation.

Table 10: **Full results for the 1-layer architecture.** Normalised area under the learning curve (AUC, $\times 10^9$) up to total training steps (interquartile mean over five seeds). Best $\alpha$ per activation type (FReLU, FLReLU, FPReLU, FSwish, FGELU) is shown in **bold**; the overall best configuration for each task is highlighted in purple.

(a) TD3

| Task | ReLU | FReLU $\alpha$=0.1 | 0.2 | 0.3 | 0.4 | 0.5 | FLReLU 0.1 | 0.2 | 0.3 | 0.4 | 0.5 | FPReLU 0.1 | 0.2 | 0.3 | 0.4 | 0.5 | FSwish 0.1 | 0.2 | 0.3 | 0.4 | 0.5 | FGELU 0.1 | 0.2 | 0.3 | 0.4 | 0.5 |
|---|---|---|---|---|---|---|---|---|---|---|---|---|---|---|---|---|---|---|---|---|---|---|---|---|---|---|
| Ant-v4 | 1.51 | 1.47 | 1.40 | **1.62** | 1.30 | 1.26 | 1.47 | 1.52 | **1.93** | 1.38 | 1.13 | 1.18 | **1.23** | 0.96 | 0.55 | -0.02 | 1.26 | 1.20 | 2.06 | 1.49 | 1.44 | 1.19 | 1.20 | 1.36 | 0.99 | **1.53** |
| Cartpole-Swingup | 0.66 | 0.66 | **0.71** | 0.63 | 0.66 | 0.62 | 0.69 | 0.70 | **0.70** | 0.66 | 0.65 | 0.63 | **0.70** | 0.69 | 0.61 | 0.65 | 0.67 | 0.68 | 0.68 | 0.65 | **0.68** | 0.72 | 0.76 | 0.72 | 0.71 | 0.69 |
| Cheetah-Run | 0.28 | 0.29 | 0.31 | 0.32 | **0.33** | 0.30 | 0.32 | 0.31 | 0.31 | **0.35** | 0.31 | 0.29 | 0.29 | 0.37 | **0.37** | 0.31 | 0.30 | 0.27 | **0.33** | 0.31 | 0.29 | 0.30 | 0.32 | 0.31 | 0.32 | **0.32** |
| Finger-Spin | 0.54 | **0.56** | 0.50 | 0.50 | 0.44 | 0.52 | 0.51 | **0.52** | 0.40 | 0.18 | 0.21 | **0.52** | 0.37 | 0.32 | 0.26 | 0.14 | 0.53 | **0.54** | 0.50 | 0.53 | 0.54 | 0.53 | 0.53 | 0.52 | 0.56 | 0.59 |
| HalfCheetah-v4 | 2.40 | 3.87 | 3.00 | 4.00 | 3.56 | 2.92 | 2.18 | 3.39 | 2.28 | 2.06 | **3.66** | **3.19** | 2.39 | 3.10 | 3.10 | 2.42 | **3.66** | 3.52 | 3.41 | 3.07 | 2.69 | 1.02 | 2.65 | 2.47 | 2.82 | **3.76** |
| Hopper-v4 | 1.17 | **1.32** | 0.97 | 0.69 | 0.35 | 0.58 | 1.12 | **1.27** | 0.75 | 0.68 | 0.41 | 1.79 | 1.31 | 0.91 | 0.83 | 1.19 | **1.64** | 1.55 | 1.10 | 1.37 | 0.90 | 1.50 | 1.54 | 1.13 | **1.58** | 1.45 |
| Humanoid-v4 | 2.38 | 2.03 | 2.23 | **2.40** | 0.91 | 0.76 | 2.36 | **2.55** | 2.28 | 1.37 | 0.83 | 1.60 | 1.06 | 0.81 | **1.62** | 0.73 | 2.14 | 2.24 | 2.12 | 2.57 | 1.73 | 1.69 | 2.06 | 2.54 | 1.42 | **2.57** |
| Walker-Walk | 0.59 | **0.69** | 0.55 | 0.60 | 0.42 | 0.60 | 0.50 | 0.50 | 0.44 | **0.55** | 0.27 | 0.62 | **0.65** | 0.47 | 0.52 | 0.34 | 0.64 | 0.63 | 0.61 | **0.68** | 0.65 | 0.53 | 0.62 | 0.69 | 0.68 | 0.64 |

(b) SAC

| Task | ReLU | FReLU $\alpha$=0.1 | 0.2 | 0.3 | 0.4 | 0.5 | FLReLU 0.1 | 0.2 | 0.3 | 0.4 | 0.5 | FPReLU 0.1 | 0.2 | 0.3 | 0.4 | 0.5 | FSwish 0.1 | 0.2 | 0.3 | 0.4 | 0.5 | FGELU 0.1 | 0.2 | 0.3 | 0.4 | 0.5 |
|---|---|---|---|---|---|---|---|---|---|---|---|---|---|---|---|---|---|---|---|---|---|---|---|---|---|---|
| Ant-v4 | 1.22 | **0.94** | 0.91 | 0.92 | 0.91 | 0.90 | **1.11** | 0.92 | 0.92 | 0.92 | 0.91 | 0.91 | **0.92** | 0.91 | 0.91 | 0.90 | **1.33** | 0.94 | 0.90 | 0.95 | 0.91 | 1.71 | 1.12 | 1.01 | 0.94 | 0.97 |
| Cartpole-Swingup | 0.73 | 0.74 | **0.75** | 0.74 | 0.70 | 0.72 | 0.73 | **0.76** | 0.72 | 0.70 | 0.66 | 0.72 | 0.70 | **0.74** | 0.72 | 0.70 | 0.74 | 0.73 | 0.73 | 0.76 | 0.72 | 0.72 | **0.75** | 0.73 | 0.73 | 0.73 |
| Cheetah-Run | 0.27 | **0.33** | 0.32 | 0.31 | 0.32 | 0.30 | **0.34** | 0.31 | 0.32 | 0.32 | 0.31 | **0.32** | 0.26 | 0.30 | 0.30 | 0.27 | 0.31 | 0.37 | 0.34 | 0.29 | 0.30 | 0.29 | **0.33** | 0.30 | 0.28 | 0.32 |
| Finger-Spin | 0.60 | 0.59 | 0.59 | **0.63** | 0.60 | 0.57 | 0.54 | 0.58 | 0.59 | **0.63** | 0.57 | **0.57** | 0.53 | 0.56 | 0.53 | 0.55 | 0.59 | 0.54 | 0.57 | **0.63** | 0.60 | 0.43 | 0.59 | 0.63 | 0.64 | 0.65 |
| HalfCheetah-v4 | 4.30 | 4.81 | **4.89** | 4.53 | 4.01 | 4.28 | 2.93 | 4.31 | 4.63 | **4.86** | 3.12 | 4.48 | **4.65** | 4.40 | 4.31 | 3.63 | 4.50 | 3.60 | 4.45 | **4.64** | 4.43 | 4.47 | 4.68 | 4.47 | **4.93** | 4.45 |
| Hopper-v4 | 1.42 | 1.22 | 1.59 | **1.64** | 1.15 | 0.81 | 1.59 | **1.99** | 1.56 | 1.48 | 1.47 | **2.10** | 1.80 | 1.67 | 1.99 | 1.46 | **2.01** | 1.79 | 1.49 | 1.43 | 0.96 | 2.36 | 2.26 | 2.15 | 2.37 | 1.91 |
| Humanoid-v4 | 3.33 | **3.17** | 2.97 | 2.94 | 2.61 | 2.61 | **3.40** | 3.11 | 3.11 | 3.39 | 2.63 | 3.08 | **3.30** | 2.41 | 2.49 | 2.93 | 3.64 | 3.11 | 3.38 | 3.20 | 3.41 | **3.52** | 3.42 | 3.51 | 3.50 | 3.21 |
| Walker-Walk | 0.53 | **0.56** | 0.51 | 0.56 | 0.49 | 0.54 | **0.60** | 0.50 | 0.59 | 0.54 | 0.48 | 0.50 | **0.53** | 0.46 | 0.49 | 0.41 | 0.52 | 0.65 | 0.54 | 0.59 | 0.52 | **0.63** | 0.58 | 0.46 | 0.51 | 0.50 |

### C.5.2 Two-Layer Architectures

The following tables report the complete two-layer results for all activation placement strategies. In the two-layer setting, activations can be applied in different locations within the actor and critic networks. This allows the analysis to examine whether fractional nonlinearities are more useful when applied to the actor, the critic, both networks, or only the first hidden layer.

Each table corresponds to one placement strategy. The results show that activation placement can substantially influence performance, especially in the deeper architecture. This supports the main observation that fractional activation functions should be evaluated not only by activation family and fractional order, but also by where they are introduced within actor–critic networks.

### C.6 Best Configuration per Activation and Task

Tables 21–24 report the best configuration for each activation on each benchmark task. For each task and activation, the table reports the configuration that achieves the highest mean normalized AUC relative to the ReLU baseline. The reported values, therefore, show the strongest task-level improvement obtained by each activation under the corresponding algorithm and network depth.

These tables complement the aggregate results by showing how the preferred fractional order $\alpha$ and, for two-layer networks, the activation placement vary across tasks and algorithms. The tables include the non-fractional baselines ReLU, LReLU, PReLU, Swish, and GELU, as well as the fractional activations FReLU, FLReLU, FPReLU, FSwish, and FGELU. In the two-layer tables, placement abbreviations follow

Table 11: Normalised area under the learning curve (AUC) ($\times 10^9$) (interquartile mean over five seeds) for **SAC** with **2layers** under placement **all-actor**. Best $\alpha$ per activation type in **bold**; overall best per task in purple.

| Task | ReLU | FReLU | | | | | FLReLU | | | | | FPReLU | | | | | FSwish | | | | | FGELU | | | | |
|---|---|---|---|---|---|---|---|---|---|---|---|---|---|---|---|---|---|---|---|---|---|---|---|---|---|---|
| | | α=0.1 | 0.2 | 0.3 | 0.4 | 0.5 | 0.1 | 0.2 | 0.3 | 0.4 | 0.5 | 0.1 | 0.2 | 0.3 | 0.4 | 0.5 | 0.1 | 0.2 | 0.3 | 0.4 | 0.5 | 0.1 | 0.2 | 0.3 | 0.4 | 0.5 |
| Ant-v4 | 2.37 | 2.69 | **2.92** | 2.50 | 2.84 | 2.52 | 2.43 | 2.84 | 2.80 | **3.37** | 3.19 | 2.48 | 2.99 | 2.74 | **3.41** | 3.27 | 2.55 | 2.29 | 2.37 | **2.73** | 2.57 | **2.96** | 1.97 | 2.80 | 2.41 | 2.72 |
| Cartpole-Swingup | 0.77 | 0.76 | 0.78 | 0.78 | **0.79** | 0.77 | 0.78 | 0.77 | **0.80** | 0.78 | 0.78 | 0.77 | 0.77 | 0.76 | **0.78** | 0.77 | 0.78 | 0.77 | **0.78** | 0.77 | 0.78 | 0.78 | **0.79** | 0.76 | 0.76 | 0.78 |
| Cheetah-Run | 0.44 | 0.45 | 0.48 | 0.46 | 0.46 | **0.50** | 0.50 | 0.47 | **0.50** | 0.47 | 0.47 | 0.50 | 0.48 | **0.54** | 0.47 | 0.52 | **0.47** | 0.47 | 0.45 | 0.38 | 0.45 | 0.46 | 0.48 | 0.48 | **0.49** | 0.47 |
| Finger-Spin | 0.73 | 0.77 | 0.74 | 0.74 | 0.75 | **0.79** | 0.76 | 0.75 | **0.81** | 0.79 | 0.76 | 0.75 | 0.76 | 0.76 | 0.76 | **0.78** | 0.75 | 0.73 | 0.76 | 0.77 | **0.80** | 0.77 | 0.76 | **0.80** | 0.77 | 0.76 |
| HalfCheetah-v4 | 6.81 | 7.18 | 6.52 | **7.60** | 6.82 | 5.80 | 6.75 | **7.33** | 5.43 | 6.45 | 6.97 | 6.57 | 7.45 | 7.36 | **7.52** | 7.45 | **8.06** | 7.71 | 8.01 | 7.81 | 6.96 | 7.53 | 7.35 | 7.28 | **7.57** | 6.91 |
| Hopper-v4 | 2.35 | 2.27 | 2.39 | 2.34 | 2.50 | **2.66** | 2.56 | 2.40 | 2.53 | **2.63** | 2.35 | 2.19 | 2.53 | **2.85** | 2.37 | 2.67 | 2.31 | **2.61** | 2.37 | 2.47 | 2.41 | 2.53 | 2.37 | **2.72** | 2.45 | 2.20 |
| Humanoid-v4 | 3.83 | 3.57 | 3.70 | **3.91** | 3.66 | 3.87 | 3.67 | 3.58 | 3.76 | 3.82 | **3.95** | 3.72 | 3.63 | 3.32 | **3.81** | 3.71 | 3.61 | **3.61** | 3.39 | 3.48 | 3.49 | 3.48 | 3.70 | 3.52 | 3.61 | **3.85** |
| Walker-Walk | 0.78 | 0.79 | 0.76 | **0.81** | 0.78 | 0.70 | 0.84 | **0.86** | 0.82 | 0.77 | 0.81 | 0.83 | 0.77 | **0.85** | 0.84 | 0.84 | 0.82 | 0.80 | 0.79 | 0.84 | **0.86** | 0.77 | 0.81 | 0.69 | **0.83** | 0.72 |

Table 12: Normalised area under the learning curve (AUC) ($\times 10^9$) (interquartile mean over five seeds) for **SAC** with **2layers** under placement **all-both**. Best $\alpha$ per activation type in **bold**; overall best per task in purple.

| Task | ReLU | FReLU | | | | | FLReLU | | | | | FPReLU | | | | | FSwish | | | | | FGELU | | | | |
|---|---|---|---|---|---|---|---|---|---|---|---|---|---|---|---|---|---|---|---|---|---|---|---|---|---|---|
| | | α=0.1 | 0.2 | 0.3 | 0.4 | 0.5 | 0.1 | 0.2 | 0.3 | 0.4 | 0.5 | 0.1 | 0.2 | 0.3 | 0.4 | 0.5 | 0.1 | 0.2 | 0.3 | 0.4 | 0.5 | 0.1 | 0.2 | 0.3 | 0.4 | 0.5 |
| Ant-v4 | 2.37 | **2.45** | 2.36 | 2.07 | 1.47 | 1.03 | **2.70** | 2.21 | 2.49 | 1.61 | 1.27 | 2.31 | **2.99** | 1.41 | 1.14 | 1.06 | **3.58** | 3.06 | 2.88 | 2.95 | 3.44 | **3.09** | 2.97 | 2.62 | 2.45 | 2.57 |
| Cartpole-Swingup | 0.77 | **0.78** | 0.74 | 0.77 | 0.77 | 0.76 | **0.78** | 0.77 | 0.75 | 0.78 | 0.77 | 0.74 | 0.77 | 0.76 | 0.77 | **0.79** | 0.69 | 0.76 | **0.78** | 0.76 | 0.78 | **0.78** | 0.74 | 0.78 | 0.76 | 0.76 |
| Cheetah-Run | 0.44 | **0.44** | 0.43 | 0.39 | 0.39 | 0.33 | 0.45 | **0.46** | 0.41 | 0.41 | 0.37 | 0.44 | **0.45** | 0.41 | 0.39 | 0.36 | 0.45 | **0.48** | 0.47 | 0.43 | 0.46 | 0.48 | 0.44 | 0.45 | **0.49** | 0.43 |
| Finger-Spin | 0.73 | 0.76 | 0.77 | **0.78** | 0.77 | 0.68 | **0.78** | 0.75 | 0.74 | 0.77 | 0.74 | 0.71 | **0.78** | 0.73 | 0.74 | 0.73 | **0.80** | 0.78 | 0.76 | 0.80 | 0.80 | **0.80** | 0.77 | 0.78 | 0.79 | 0.80 |
| HalfCheetah-v4 | 6.81 | **6.90** | 6.47 | 6.17 | 5.72 | 5.04 | 7.19 | 6.28 | **7.25** | 6.51 | 6.66 | 6.52 | **7.12** | 6.14 | 4.90 | 4.74 | 8.29 | **8.54** | 7.42 | 7.16 | 8.22 | 7.76 | 6.89 | **8.02** | 7.90 | 7.33 |
| Hopper-v4 | 2.35 | 2.51 | **2.86** | 2.38 | 2.53 | 2.48 | 2.63 | 2.34 | 1.97 | 2.44 | **2.66** | 2.56 | **2.99** | 2.74 | 2.71 | 2.88 | **2.36** | 2.24 | 2.30 | 2.23 | 2.06 | 2.42 | 2.15 | 2.43 | 2.67 | **2.76** |
| Humanoid-v4 | 3.83 | 3.73 | 3.52 | 3.58 | **3.87** | 3.56 | 3.98 | 3.65 | 3.70 | **4.12** | 3.71 | 3.98 | 3.87 | **4.25** | 4.07 | 4.17 | 3.66 | 3.43 | 3.61 | 3.65 | **3.70** | **3.85** | 3.54 | 3.61 | 3.74 | 3.58 |
| Walker-Walk | 0.78 | 0.77 | **0.81** | 0.75 | 0.77 | 0.59 | 0.77 | 0.75 | **0.80** | 0.75 | 0.57 | 0.67 | 0.63 | 0.77 | 0.67 | **0.78** | **0.84** | 0.83 | 0.81 | 0.83 | 0.69 | 0.83 | **0.83** | 0.79 | 0.69 | 0.74 |

Table 13: Normalised area under the learning curve (AUC) ($\times 10^9$) (interquartile mean over five seeds) for **SAC** with **2layers** under placement **all-critic**. Best $\alpha$ per activation type in **bold**; overall best per task in purple.

| Task | ReLU | FReLU | | | | | FLReLU | | | | | FPReLU | | | | | FSwish | | | | | FGELU | | | | |
|---|---|---|---|---|---|---|---|---|---|---|---|---|---|---|---|---|---|---|---|---|---|---|---|---|---|---|
| | | α=0.1 | 0.2 | 0.3 | 0.4 | 0.5 | 0.1 | 0.2 | 0.3 | 0.4 | 0.5 | 0.1 | 0.2 | 0.3 | 0.4 | 0.5 | 0.1 | 0.2 | 0.3 | 0.4 | 0.5 | 0.1 | 0.2 | 0.3 | 0.4 | 0.5 |
| Ant-v4 | 2.37 | **3.17** | 2.39 | 2.14 | 1.78 | 1.20 | **2.38** | 2.07 | 2.04 | 1.12 | 1.27 | 3.17 | **3.76** | 3.45 | 1.36 | 1.20 | 3.66 | 3.45 | 3.05 | **3.77** | 2.84 | 3.45 | 2.85 | 3.44 | **3.58** | 2.82 |
| Cartpole-Swingup | 0.77 | 0.77 | 0.77 | 0.78 | **0.78** | 0.78 | 0.78 | 0.78 | 0.76 | **0.78** | 0.77 | 0.74 | 0.76 | 0.77 | **0.78** | 0.78 | **0.78** | 0.78 | 0.77 | 0.78 | 0.78 | 0.77 | 0.76 | 0.75 | **0.77** | 0.72 |
| Cheetah-Run | 0.44 | 0.46 | 0.43 | 0.41 | 0.37 | 0.38 | **0.45** | 0.44 | 0.42 | 0.38 | 0.38 | 0.39 | **0.42** | 0.39 | 0.38 | 0.36 | **0.47** | 0.47 | 0.47 | 0.44 | 0.45 | **0.47** | 0.46 | 0.44 | 0.44 | 0.45 |
| Finger-Spin | 0.73 | 0.78 | **0.79** | 0.73 | 0.74 | 0.69 | 0.76 | 0.76 | **0.79** | 0.73 | 0.74 | 0.70 | 0.73 | 0.76 | **0.76** | 0.73 | 0.70 | **0.81** | 0.80 | 0.75 | 0.76 | **0.80** | 0.78 | 0.76 | 0.79 | 0.78 |
| HalfCheetah-v4 | 6.81 | 7.13 | **7.31** | 6.11 | 7.01 | 6.79 | 7.39 | **7.67** | 6.91 | 6.46 | 5.91 | **6.45** | 5.76 | 6.05 | 3.78 | 5.48 | **8.77** | 8.12 | 8.46 | 8.50 | 8.75 | 7.94 | **8.15** | 7.03 | 7.42 | 6.33 |
| Hopper-v4 | 2.35 | 2.76 | 2.64 | 2.52 | 2.21 | 2.27 | 2.39 | **2.55** | 2.36 | 2.38 | 2.35 | **2.76** | 2.75 | 2.43 | 2.36 | 2.68 | 2.36 | **2.56** | 2.17 | 2.10 | 2.24 | 2.37 | 2.51 | **2.64** | 2.46 | 2.30 |
| Humanoid-v4 | 3.83 | 3.50 | 3.53 | 3.39 | **3.60** | 3.50 | 3.45 | 3.53 | 3.57 | 3.52 | **3.76** | 3.97 | 3.92 | 4.04 | 3.80 | **4.10** | 3.76 | **3.91** | 3.76 | 3.74 | 3.77 | **3.91** | 3.80 | 3.72 | 3.86 | 3.86 |
| Walker-Walk | 0.78 | **0.75** | 0.74 | 0.73 | 0.54 | 0.69 | 0.78 | 0.73 | **0.83** | 0.71 | 0.69 | 0.62 | **0.71** | 0.67 | 0.68 | 0.63 | **0.84** | 0.70 | 0.72 | 0.77 | 0.83 | **0.80** | 0.77 | 0.75 | 0.79 | 0.76 |

Table 14: Normalised area under the learning curve (AUC) ($\times 10^9$) (interquartile mean over five seeds) for **SAC** with **2layers** under placement **first-actor**. Best $\alpha$ per activation type in **bold**; overall best per task in purple.

| Task | ReLU | FReLU | | | | | FLReLU | | | | | FPReLU | | | | | FSwish | | | | | FGELU | | | | |
|---|---|---|---|---|---|---|---|---|---|---|---|---|---|---|---|---|---|---|---|---|---|---|---|---|---|---|
| | | α=0.1 | 0.2 | 0.3 | 0.4 | 0.5 | 0.1 | 0.2 | 0.3 | 0.4 | 0.5 | 0.1 | 0.2 | 0.3 | 0.4 | 0.5 | 0.1 | 0.2 | 0.3 | 0.4 | 0.5 | 0.1 | 0.2 | 0.3 | 0.4 | 0.5 |
| Ant-v4 | 2.37 | 3.15 | 3.48 | 3.48 | 2.99 | 2.58 | 2.35 | 3.12 | 2.56 | 3.18 | **3.22** | 2.16 | **3.37** | 2.75 | 2.57 | 3.26 | **2.93** | 2.05 | 2.70 | 0.03 | 2.18 | 2.59 | **3.04** | 2.37 | 2.82 | 2.60 |
| Cartpole-Swingup | 0.77 | 0.77 | 0.78 | 0.75 | 0.79 | **0.79** | 0.77 | 0.78 | 0.78 | 0.80 | 0.78 | 0.78 | 0.78 | 0.78 | **0.79** | 0.77 | 0.79 | 0.79 | 0.77 | 0.78 | **0.79** | **0.79** | 0.77 | 0.77 | 0.78 | 0.78 |
| Cheetah-Run | 0.44 | 0.52 | 0.49 | 0.49 | 0.50 | 0.49 | 0.48 | 0.48 | 0.47 | 0.49 | **0.50** | 0.43 | 0.48 | 0.48 | **0.50** | 0.48 | 0.45 | 0.46 | 0.48 | **0.48** | 0.47 | 0.49 | **0.50** | 0.44 | 0.45 | 0.46 |
| Finger-Spin | 0.73 | 0.76 | 0.76 | 0.76 | 0.72 | **0.78** | **0.77** | 0.76 | 0.76 | 0.75 | 0.75 | 0.74 | 0.75 | 0.84 | 0.79 | 0.78 | 0.77 | **0.79** | 0.77 | 0.79 | 0.77 | **0.77** | 0.73 | 0.75 | 0.76 | 0.75 |
| HalfCheetah-v4 | 6.81 | 7.56 | 8.11 | 6.90 | 6.96 | 7.28 | 7.24 | 7.13 | **7.26** | 7.08 | 7.19 | 7.34 | 7.35 | **7.58** | 7.38 | 6.83 | 7.23 | 7.78 | 7.67 | 7.55 | **7.91** | **7.69** | 7.22 | 6.97 | 6.18 | 6.79 |
| Hopper-v4 | 2.35 | 2.58 | 2.41 | 2.38 | 2.74 | 2.49 | 2.28 | 2.49 | 2.51 | **2.54** | 2.36 | 2.54 | **2.61** | 2.56 | 2.51 | 2.61 | 2.46 | 2.41 | 2.46 | 2.36 | **2.55** | 2.35 | 2.36 | 2.49 | 2.63 | **2.65** |
| Humanoid-v4 | 3.83 | 3.62 | **3.82** | 3.57 | 3.60 | 3.68 | 3.59 | 3.59 | **3.66** | 3.63 | 3.51 | 3.66 | 3.68 | 3.73 | 3.57 | **3.87** | 4.09 | 3.86 | 3.91 | 3.97 | 3.75 | **4.01** | 3.85 | 3.97 | 3.98 | 3.75 |
| Walker-Walk | 0.78 | **0.85** | 0.81 | 0.74 | 0.83 | 0.84 | 0.83 | 0.80 | **0.84** | 0.78 | 0.70 | 0.80 | 0.71 | 0.81 | 0.81 | **0.85** | **0.85** | 0.74 | 0.83 | 0.79 | 0.84 | **0.85** | 0.79 | 0.84 | 0.82 | 0.76 |

Table 15: Normalised area under the learning curve (AUC) ($\times 10^9$) (interquartile mean over five seeds) for **SAC** with **2layers** under placement **first-both**. Best $\alpha$ per activation type in **bold**; overall best per task in purple.

| Task | ReLU | FReLU | | | | | FLReLU | | | | | FPReLU | | | | | FSwish | | | | | FGELU | | | | |
|---|---|---|---|---|---|---|---|---|---|---|---|---|---|---|---|---|---|---|---|---|---|---|---|---|---|---|
| | | α=0.1 | 0.2 | 0.3 | 0.4 | 0.5 | 0.1 | 0.2 | 0.3 | 0.4 | 0.5 | 0.1 | 0.2 | 0.3 | 0.4 | 0.5 | 0.1 | 0.2 | 0.3 | 0.4 | 0.5 | 0.1 | 0.2 | 0.3 | 0.4 | 0.5 |
| Ant-v4 | 2.37 | **3.07** | 2.88 | 1.87 | 1.38 | 1.59 | **2.88** | 2.76 | 1.42 | 1.15 | 1.19 | **3.34** | 2.11 | 2.39 | 1.24 | 1.41 | 3.69 | 3.41 | 2.95 | 2.80 | **3.88** | 2.99 | **3.33** | 2.66 | 2.50 | 3.03 |
| Cartpole-Swingup | 0.77 | 0.78 | 0.78 | 0.78 | 0.78 | **0.78** | **0.79** | 0.77 | 0.78 | 0.77 | 0.79 | 0.76 | **0.79** | 0.78 | 0.77 | 0.78 | 0.76 | 0.77 | 0.77 | 0.78 | **0.78** | 0.77 | 0.74 | 0.77 | 0.79 | 0.80 |
| Cheetah-Run | 0.44 | 0.51 | 0.46 | 0.36 | 0.48 | 0.43 | 0.47 | 0.43 | 0.46 | **0.48** | 0.44 | 0.47 | 0.46 | 0.46 | **0.49** | 0.46 | 0.45 | 0.46 | 0.47 | 0.47 | **0.49** | **0.46** | 0.44 | 0.44 | 0.46 | 0.44 |
| Finger-Spin | 0.73 | 0.78 | 0.75 | 0.78 | 0.76 | **0.79** | **0.78** | 0.77 | 0.76 | 0.78 | 0.78 | 0.77 | 0.77 | **0.79** | 0.79 | 0.71 | 0.76 | 0.74 | 0.79 | 0.81 | 0.79 | 0.77 | 0.78 | **0.78** | 0.77 | 0.76 |
| HalfCheetah-v4 | 6.81 | 7.23 | 6.85 | 7.01 | 6.70 | 5.89 | 7.35 | 6.54 | **7.75** | 7.25 | 6.49 | 7.39 | 7.63 | 8.22 | 7.49 | 6.91 | 8.02 | 7.32 | 7.81 | **8.21** | 8.18 | **8.17** | 6.94 | 7.07 | 6.88 | 6.64 |
| Hopper-v4 | 2.35 | 2.47 | 2.39 | 2.11 | 2.61 | **2.64** | 2.64 | **2.77** | 2.72 | 2.67 | 2.67 | 2.33 | 2.87 | 2.53 | 2.31 | 2.69 | **2.63** | 2.24 | 2.48 | 2.59 | 2.18 | 2.57 | 2.55 | **2.82** | 2.61 | 2.73 |
| Humanoid-v4 | 3.83 | 3.70 | 3.72 | 3.72 | 3.76 | **4.15** | 3.56 | 3.75 | 3.67 | 3.83 | **4.19** | 4.25 | 4.05 | 4.19 | 4.06 | 4.06 | 3.73 | 3.68 | **3.74** | 3.71 | 3.65 | 3.81 | **3.84** | 3.80 | 3.75 | 3.64 |
| Walker-Walk | 0.78 | 0.79 | 0.80 | **0.81** | 0.75 | 0.68 | 0.85 | 0.80 | 0.83 | 0.82 | 0.78 | 0.74 | **0.82** | 0.80 | 0.72 | 0.60 | 0.72 | 0.81 | 0.80 | **0.83** | 0.81 | 0.81 | 0.82 | **0.82** | 0.81 | 0.80 |

Table 16: Normalised area under the learning curve (AUC) ($\times 10^9$) (interquartile mean over five seeds) for **TD3** with **2layers** under placement **all-actor**. Best $\alpha$ per activation type in **bold**; overall best per task in purple.

| Task | ReLU | FReLU | | | | | FLReLU | | | | | FPReLU | | | | | FSwish | | | | | FGELU | | | | |
|---|---|---|---|---|---|---|---|---|---|---|---|---|---|---|---|---|---|---|---|---|---|---|---|---|---|---|
| | | α=0.1 | 0.2 | 0.3 | 0.4 | 0.5 | 0.1 | 0.2 | 0.3 | 0.4 | 0.5 | 0.1 | 0.2 | 0.3 | 0.4 | 0.5 | 0.1 | 0.2 | 0.3 | 0.4 | 0.5 | 0.1 | 0.2 | 0.3 | 0.4 | 0.5 |
| Ant-v4 | 2.54 | 2.19 | **2.44** | 2.23 | 2.12 | 2.17 | 2.39 | **2.44** | 2.43 | 2.22 | 2.39 | 2.33 | **2.33** | 2.15 | 2.15 | 2.17 | 2.25 | 2.42 | **2.50** | 2.13 | 2.28 | 2.36 | **2.53** | 2.31 | 2.30 | 2.48 |
| Cartpole-Swingup | 0.75 | 0.75 | 0.76 | 0.74 | **0.77** | 0.75 | 0.75 | 0.74 | 0.76 | 0.75 | **0.77** | 0.76 | **0.76** | 0.74 | 0.76 | 0.75 | **0.77** | 0.75 | 0.75 | 0.75 | 0.77 | 0.76 | 0.75 | 0.75 | **0.77** | 0.77 |
| Cheetah-Run | 0.52 | **0.56** | 0.52 | 0.51 | 0.55 | 0.53 | **0.55** | 0.48 | 0.55 | 0.52 | 0.49 | 0.47 | 0.52 | 0.54 | **0.57** | 0.52 | 0.50 | **0.54** | 0.53 | 0.53 | 0.45 | **0.54** | 0.51 | 0.51 | 0.45 | 0.52 |
| Finger-Spin | 0.72 | 0.69 | 0.73 | 0.77 | **0.78** | 0.76 | 0.78 | 0.77 | 0.76 | 0.77 | 0.76 | 0.23 | 0.61 | 0.60 | 0.66 | **0.71** | 0.72 | 0.74 | **0.76** | 0.71 | 0.66 | 0.75 | **0.77** | 0.75 | 0.74 | 0.75 |
| HalfCheetah-v4 | 2.69 | 2.41 | **3.54** | 2.63 | 2.80 | 2.04 | 1.54 | **2.70** | 2.44 | 2.63 | 2.67 | 2.60 | 3.56 | 4.47 | 2.69 | 3.61 | **4.17** | 2.63 | 3.72 | 1.60 | 2.47 | 1.49 | **2.99** | 2.62 | 1.71 | 2.88 |
| Hopper-v4 | 2.16 | 1.76 | 2.28 | 1.91 | 1.67 | **2.62** | 1.90 | 2.25 | **2.25** | 2.18 | 1.57 | 2.72 | 2.43 | 2.36 | 2.32 | 2.47 | 1.74 | 2.01 | 2.20 | **2.23** | 2.20 | **2.31** | 1.78 | 2.27 | 2.07 | 1.99 |
| Humanoid-v4 | 3.51 | 3.60 | 3.65 | 3.62 | **3.74** | 3.46 | 3.51 | 3.78 | 3.47 | **3.78** | 3.76 | 0.92 | 3.04 | 3.27 | **3.71** | 3.32 | 3.40 | 3.40 | **3.62** | 3.50 | 3.55 | 3.77 | 3.59 | 3.77 | 3.54 | 3.78 |
| Walker-Walk | 0.79 | 0.70 | 0.77 | **0.86** | 0.80 | 0.77 | **0.87** | 0.62 | 0.81 | 0.67 | 0.82 | 0.85 | 0.81 | 0.89 | 0.87 | 0.89 | 0.76 | 0.79 | **0.82** | 0.81 | 0.78 | 0.85 | 0.82 | **0.88** | 0.85 | 0.81 |

Table 17: Normalised area under the learning curve (AUC) ($\times 10^9$) (interquartile mean over five seeds) for **TD3** with **2layers** under placement **all-both**. Best $\alpha$ per activation type in **bold**; overall best per task in purple.

| Task | ReLU | FReLU | | | | | FLReLU | | | | | FPReLU | | | | | FSwish | | | | | FGELU | | | | |
|---|---|---|---|---|---|---|---|---|---|---|---|---|---|---|---|---|---|---|---|---|---|---|---|---|---|---|
| | | α=0.1 | 0.2 | 0.3 | 0.4 | 0.5 | 0.1 | 0.2 | 0.3 | 0.4 | 0.5 | 0.1 | 0.2 | 0.3 | 0.4 | 0.5 | 0.1 | 0.2 | 0.3 | 0.4 | 0.5 | 0.1 | 0.2 | 0.3 | 0.4 | 0.5 |
| Ant-v4 | 2.54 | **2.41** | 1.97 | 2.37 | 1.05 | 0.90 | 2.42 | **2.43** | 1.94 | 1.07 | 0.86 | **2.44** | 2.28 | 1.55 | 0.24 | -0.02 | **2.50** | 2.31 | 2.33 | 1.99 | 1.71 | 2.30 | 2.26 | **2.41** | 2.38 | 2.30 |
| Cartpole-Swingup | 0.75 | 0.75 | **0.76** | 0.75 | 0.76 | 0.74 | 0.75 | 0.73 | **0.77** | 0.75 | 0.76 | 0.75 | 0.75 | 0.73 | **0.76** | 0.74 | 0.77 | 0.76 | 0.78 | 0.76 | 0.77 | 0.77 | **0.78** | 0.75 | 0.77 | 0.76 |
| Cheetah-Run | 0.52 | 0.50 | 0.47 | 0.50 | **0.51** | 0.42 | 0.48 | **0.51** | 0.48 | 0.47 | 0.43 | 0.47 | **0.50** | 0.49 | 0.46 | 0.41 | **0.51** | 0.45 | 0.48 | 0.48 | 0.49 | 0.47 | 0.48 | **0.51** | 0.49 | 0.49 |
| Finger-Spin | 0.72 | 0.71 | **0.76** | 0.69 | 0.63 | 0.55 | **0.78** | 0.73 | 0.42 | 0.62 | 0.52 | 0.63 | **0.69** | 0.58 | 0.67 | 0.46 | 0.80 | 0.81 | **0.81** | 0.78 | 0.76 | 0.82 | 0.82 | 0.81 | 0.80 | 0.77 |
| HalfCheetah-v4 | 2.69 | 2.75 | 1.53 | 2.31 | **3.74** | 2.61 | 1.56 | 1.60 | **3.00** | 1.20 | 2.99 | **3.00** | 1.71 | 2.62 | 2.76 | 2.79 | 1.72 | 1.70 | 1.65 | 2.66 | **3.80** | **3.07** | 3.00 | 1.73 | 1.75 | 1.56 |
| Hopper-v4 | 2.16 | **2.21** | 1.45 | 1.96 | 1.52 | 1.37 | 1.94 | 1.90 | 1.69 | **2.03** | 1.79 | 2.72 | 2.43 | 2.36 | 2.32 | 2.47 | 2.38 | **2.38** | 2.14 | 1.68 | 2.07 | 2.26 | **2.50** | 2.15 | 2.47 | 2.18 |
| Humanoid-v4 | 3.51 | 3.50 | **3.88** | 3.16 | 2.26 | 1.11 | **3.64** | 3.44 | 3.15 | 3.35 | 2.48 | 2.36 | **3.60** | 1.87 | 2.38 | 0.99 | 3.74 | 3.66 | 3.47 | 3.69 | **3.88** | 3.44 | **3.89** | 3.41 | 3.60 | 3.49 |
| Walker-Walk | 0.79 | 0.81 | 0.75 | 0.76 | **0.81** | 0.67 | 0.79 | **0.80** | 0.77 | 0.79 | 0.61 | 0.78 | **0.85** | 0.70 | 0.73 | 0.68 | 0.82 | **0.84** | 0.81 | 0.79 | 0.75 | 0.74 | **0.85** | 0.76 | 0.77 | 0.68 |

Table 18: Normalised area under the learning curve (AUC) ($\times 10^9$) (interquartile mean over five seeds) for **TD3** with **2layers** under placement **all-critic**. Best $\alpha$ per activation type in **bold**; overall best per task in purple.

| Task | ReLU | FReLU | | | | | FLReLU | | | | | FPReLU | | | | | FSwish | | | | | FGELU | | | | |
|---|---|---|---|---|---|---|---|---|---|---|---|---|---|---|---|---|---|---|---|---|---|---|---|---|---|---|
| | | α=0.1 | 0.2 | 0.3 | 0.4 | 0.5 | 0.1 | 0.2 | 0.3 | 0.4 | 0.5 | 0.1 | 0.2 | 0.3 | 0.4 | 0.5 | 0.1 | 0.2 | 0.3 | 0.4 | 0.5 | 0.1 | 0.2 | 0.3 | 0.4 | 0.5 |
| Ant-v4 | 2.54 | 2.60 | 2.50 | 2.13 | 2.27 | 1.33 | 2.33 | **2.43** | 2.38 | 2.19 | 1.21 | **2.48** | 2.43 | 1.97 | 0.82 | 0.40 | 1.94 | 2.19 | **2.34** | 2.27 | 1.96 | 2.24 | 2.34 | **2.49** | 2.38 | 2.39 |
| Cartpole-Swingup | 0.75 | 0.75 | 0.74 | 0.76 | **0.76** | 0.76 | 0.76 | **0.77** | 0.77 | 0.76 | 0.74 | 0.75 | 0.76 | **0.76** | 0.74 | 0.73 | 0.76 | 0.79 | 0.75 | 0.79 | 0.77 | 0.73 | 0.76 | 0.75 | **0.77** | 0.77 |
| Cheetah-Run | 0.52 | 0.47 | 0.49 | **0.51** | 0.49 | 0.49 | **0.53** | 0.52 | 0.50 | 0.48 | 0.48 | 0.47 | **0.47** | 0.46 | 0.43 | 0.39 | 0.45 | 0.48 | **0.54** | 0.50 | 0.49 | 0.54 | 0.49 | 0.52 | 0.44 | 0.57 |
| Finger-Spin | 0.72 | 0.78 | **0.79** | 0.70 | 0.63 | 0.53 | 0.70 | 0.74 | **0.77** | 0.63 | 0.58 | **0.76** | 0.72 | 0.74 | 0.66 | 0.50 | 0.80 | **0.80** | 0.79 | 0.77 | 0.76 | 0.80 | 0.81 | 0.80 | 0.78 | 0.77 |
| HalfCheetah-v4 | 2.69 | **1.56** | 1.54 | 1.34 | 1.40 | 1.15 | 1.66 | 1.56 | 2.21 | **2.21** | 1.29 | 4.15 | 3.01 | 3.29 | 4.00 | 1.91 | **3.33** | 3.16 | 2.97 | 2.93 | 2.67 | 1.78 | 3.01 | 2.87 | **3.11** | 2.51 |
| Hopper-v4 | 2.16 | 1.95 | 1.49 | **2.37** | 1.95 | 1.63 | 1.31 | **2.05** | 1.94 | 1.85 | 1.70 | **2.45** | 2.29 | 2.33 | 2.25 | 2.42 | 2.45 | 1.90 | 2.40 | 1.90 | 2.28 | 1.78 | 1.85 | **2.30** | 2.00 | 2.24 |
| Humanoid-v4 | 3.51 | **3.82** | 3.32 | 3.03 | 2.32 | 0.54 | 3.47 | **3.51** | 3.24 | 2.16 | 0.67 | **3.67** | 2.02 | 3.20 | 2.22 | 0.36 | 3.58 | 3.65 | 3.56 | 3.81 | 3.94 | 3.77 | 3.54 | 3.34 | **3.84** | 3.34 |
| Walker-Walk | 0.79 | 0.88 | 0.85 | 0.79 | 0.68 | 0.66 | 0.73 | 0.81 | **0.85** | 0.79 | 0.62 | **0.87** | 0.81 | 0.86 | 0.85 | 0.75 | **0.80** | 0.68 | 0.69 | 0.75 | 0.80 | 0.62 | 0.74 | 0.77 | 0.77 | **0.81** |

Table 19: Normalised area under the learning curve (AUC) ($\times 10^9$) (interquartile mean over five seeds) for **TD3** with **2layers** under placement **first-actor**. Best $\alpha$ per activation type in **bold**; overall best per task in purple.

| Task | ReLU | FReLU | | | | | FLReLU | | | | | FPReLU | | | | | FSwish | | | | | FGELU | | | | |
|---|---|---|---|---|---|---|---|---|---|---|---|---|---|---|---|---|---|---|---|---|---|---|---|---|---|---|
| | | α=0.1 | 0.2 | 0.3 | 0.4 | 0.5 | 0.1 | 0.2 | 0.3 | 0.4 | 0.5 | 0.1 | 0.2 | 0.3 | 0.4 | 0.5 | 0.1 | 0.2 | 0.3 | 0.4 | 0.5 | 0.1 | 0.2 | 0.3 | 0.4 | 0.5 |
| Ant-v4 | **2.54** | 2.34 | 2.31 | **2.59** | 2.24 | 2.52 | 2.32 | 2.35 | 2.42 | **2.51** | 2.08 | 2.31 | 2.36 | 2.33 | 2.38 | **2.54** | **2.60** | 2.40 | 2.39 | 2.30 | 2.39 | 2.39 | 2.44 | 2.47 | 2.72 | 2.66 |
| Cartpole-Swingup | **0.75** | 0.76 | 0.75 | 0.75 | 0.77 | **0.78** | 0.70 | **0.73** | 0.71 | 0.70 | 0.70 | 0.78 | 0.75 | 0.76 | 0.77 | 0.76 | 0.75 | **0.76** | 0.74 | 0.74 | 0.75 | **0.76** | 0.76 | 0.75 | 0.76 | 0.76 |
| Cheetah-Run | **0.52** | **0.54** | 0.52 | 0.51 | 0.49 | 0.49 | 0.49 | 0.49 | 0.54 | 0.51 | 0.53 | 0.51 | 0.53 | **0.54** | 0.46 | 0.48 | **0.52** | 0.50 | 0.49 | 0.52 | 0.51 | 0.49 | 0.52 | 0.51 | **0.53** | 0.52 |
| Finger-Spin | **0.72** | 0.71 | **0.75** | 0.72 | 0.72 | 0.70 | 0.73 | 0.71 | 0.76 | 0.73 | **0.76** | 0.70 | 0.72 | 0.73 | 0.72 | **0.75** | 0.74 | 0.72 | 0.74 | **0.78** | 0.68 | 0.80 | 0.78 | 0.78 | 0.71 | 0.75 |
| HalfCheetah-v4 | **2.69** | 1.51 | 2.62 | 2.97 | **3.81** | 2.73 | 5.47 | 2.73 | 1.69 | 2.46 | 2.61 | 1.60 | 1.71 | 2.44 | 1.93 | **2.66** | 2.85 | **5.09** | 3.47 | 4.58 | 3.94 | 1.55 | 1.62 | 1.59 | 1.60 | **4.30** |
| Hopper-v4 | **2.16** | 1.65 | **2.26** | 1.53 | 1.73 | 2.16 | 2.35 | 2.12 | 2.14 | 1.83 | 2.08 | **2.33** | 1.87 | 2.08 | 2.19 | 1.92 | **2.24** | 1.97 | 1.65 | 1.81 | 2.16 | 1.98 | 1.93 | 1.98 | 1.87 | **2.19** |
| Humanoid-v4 | **3.51** | 1.75 | 1.31 | 1.20 | 1.45 | **1.95** | 3.38 | 3.43 | **3.68** | 3.43 | 3.42 | 3.45 | 3.36 | 3.74 | 3.53 | 3.58 | 3.11 | 3.36 | **3.69** | 3.38 | 3.51 | **3.59** | 3.42 | 2.89 | 3.26 | 3.03 |
| Walker-Walk | **0.79** | 0.76 | 0.87 | 0.84 | 0.74 | 0.75 | **0.80** | 0.75 | 0.78 | 0.76 | 0.78 | 0.83 | 0.84 | **0.86** | 0.84 | 0.81 | 0.78 | **0.85** | 0.85 | 0.75 | 0.83 | 0.66 | 0.68 | 0.72 | **0.86** | 0.81 |

Table 20: Normalised area under the learning curve (AUC) ($\times 10^9$) (interquartile mean over five seeds) for **TD3** with **2layers** under placement **first-both**. Best $\alpha$ per activation type in **bold**; overall best per task in purple.

| Task | ReLU | FReLU | | | | | FLReLU | | | | | FPReLU | | | | | FSwish | | | | | FGELU | | | | |
|---|---|---|---|---|---|---|---|---|---|---|---|---|---|---|---|---|---|---|---|---|---|---|---|---|---|---|
| | | α=0.1 | 0.2 | 0.3 | 0.4 | 0.5 | 0.1 | 0.2 | 0.3 | 0.4 | 0.5 | 0.1 | 0.2 | 0.3 | 0.4 | 0.5 | 0.1 | 0.2 | 0.3 | 0.4 | 0.5 | 0.1 | 0.2 | 0.3 | 0.4 | 0.5 |
| Ant-v4 | **2.54** | 2.56 | 2.33 | 2.26 | 1.99 | 1.95 | **2.50** | 2.39 | 2.37 | 1.95 | 1.30 | 2.43 | **2.49** | 2.49 | 1.83 | 1.23 | **2.50** | 2.39 | 2.27 | 2.05 | 2.21 | **2.48** | 2.36 | 2.33 | 2.40 | 2.10 |
| Cartpole-Swingup | **0.75** | 0.75 | **0.76** | 0.74 | 0.75 | 0.75 | 0.75 | 0.75 | **0.75** | 0.74 | 0.74 | 0.76 | 0.77 | 0.74 | **0.77** | 0.75 | 0.78 | 0.77 | **0.78** | 0.77 | 0.78 | 0.76 | 0.77 | 0.75 | 0.79 | 0.75 |
| Cheetah-Run | **0.52** | **0.53** | 0.53 | 0.50 | 0.51 | 0.47 | 0.49 | 0.45 | 0.44 | **0.52** | 0.49 | 0.52 | 0.49 | **0.53** | 0.52 | 0.49 | 0.48 | 0.46 | 0.55 | 0.51 | 0.49 | 0.49 | 0.48 | 0.48 | **0.49** | 0.48 |
| Finger-Spin | **0.72** | 0.70 | **0.72** | 0.66 | 0.52 | 0.68 | **0.75** | 0.71 | 0.74 | 0.69 | 0.64 | 0.75 | 0.71 | **0.78** | 0.69 | 0.62 | 0.76 | 0.76 | 0.75 | **0.77** | 0.75 | 0.80 | 0.77 | 0.82 | 0.78 | 0.81 |
| HalfCheetah-v4 | **2.69** | 1.53 | **3.87** | 1.46 | 1.51 | 1.40 | 2.60 | **2.80** | 2.23 | 1.52 | 1.37 | 3.22 | 4.09 | **4.12** | 1.55 | 2.07 | **4.17** | 3.05 | 1.72 | 2.80 | 1.54 | 3.77 | 1.66 | 1.54 | 4.44 | 3.26 |
| Hopper-v4 | **2.16** | 1.79 | **2.10** | 1.57 | 1.70 | 1.73 | 2.05 | **2.09** | 1.88 | 1.80 | 1.57 | 2.55 | 2.32 | 2.32 | 2.40 | 1.91 | 2.38 | 2.02 | 1.94 | 2.35 | **2.51** | 2.23 | 2.43 | 2.32 | 2.32 | **2.48** |
| Humanoid-v4 | **3.51** | 3.38 | 3.35 | **3.50** | 3.00 | 2.07 | 3.68 | **3.84** | 3.30 | 3.42 | 2.42 | 3.94 | 3.76 | 3.25 | 3.23 | 1.66 | 3.70 | 3.50 | **3.79** | 3.77 | 3.74 | 3.30 | 3.23 | **3.54** | 3.32 | 3.14 |
| Walker-Walk | **0.79** | **0.82** | 0.68 | 0.74 | 0.74 | 0.75 | 0.82 | 0.71 | **0.84** | 0.74 | 0.82 | 0.81 | 0.73 | **0.82** | 0.68 | 0.71 | 0.85 | 0.86 | 0.81 | 0.78 | 0.86 | 0.82 | **0.83** | 0.82 | 0.78 | 0.76 |

the captions: A denotes the actor network, C denotes the critic network, and B denotes both the actor and critic networks.

The detailed configuration tables also help distinguish two effects that are combined in the main aggregate results. First, they show which fractional order is preferred for each activation family. Second, they show whether the best result in the two-layer architecture comes from actor-side, critic-side, or combined actor–critic placement. This provides a clearer view of how activation-function design interacts with actor–critic architecture.

Table 21: Best configuration per activation and task for TD3 with a 1-layer network. Values report mean normalized AUC improvement relative to the ReLU baseline. Task names are shortened for compactness.

| Task | Act. | $\alpha$ | $\Delta\%$ | Task | Act. | $\alpha$ | $\Delta\%$ |
|------|------|----------|-----------|------|------|----------|-----------|
| Ant | ReLU | base | +0.00 | HalfCheetah | ReLU | base | +0.00 |
| Ant | LReLU | base | +12.41 | HalfCheetah | LReLU | base | +32.22 |
| Ant | PReLU | base | +4.38 | HalfCheetah | PReLU | base | +17.17 |
| Ant | Swish | base | +1.48 | HalfCheetah | Swish | base | +33.47 |
| Ant | GELU | base | -10.84 | HalfCheetah | GELU | base | +30.28 |
| Ant | FReLU | 0.3 | +8.89 | HalfCheetah | FReLU | 0.3 | +79.67 |
| Ant | FLReLU | 0.3 | +31.54 | HalfCheetah | FLReLU | 0.5 | +42.69 |
| Ant | FPReLU | 0.2 | -13.94 | HalfCheetah | FPReLU | 0.3 | +42.03 |
| Ant | FSwish | 0.3 | +35.75 | HalfCheetah | FSwish | 0.1 | +50.73 |
| Ant | FGELU | 0.5 | +4.26 | HalfCheetah | FGELU | 0.5 | +47.19 |
| Cartpole | ReLU | base | +0.00 | Hopper | ReLU | base | +0.00 |
| Cartpole | LReLU | base | -7.89 | Hopper | LReLU | base | -7.55 |
| Cartpole | PReLU | base | +3.91 | Hopper | PReLU | base | -7.84 |
| Cartpole | Swish | base | -32.71 | Hopper | Swish | base | +37.01 |
| Cartpole | GELU | base | -4.61 | Hopper | GELU | base | +9.54 |
| Cartpole | FReLU | 0.2 | +7.42 | Hopper | FReLU | 0.1 | +26.17 |
| Cartpole | FLReLU | 0.3 | +6.93 | Hopper | FLReLU | 0.2 | +15.63 |
| Cartpole | FPReLU | 0.2 | +3.90 | Hopper | FPReLU | 0.1 | +65.26 |
| Cartpole | FSwish | 0.5 | +4.85 | Hopper | FSwish | 0.1 | +40.50 |
| Cartpole | FGELU | 0.2 | +12.16 | Hopper | FGELU | 0.4 | +41.53 |
| Cheetah | ReLU | base | +0.00 | Humanoid | ReLU | base | +0.00 |
| Cheetah | LReLU | base | +11.25 | Humanoid | LReLU | base | -26.40 |
| Cheetah | PReLU | base | +9.66 | Humanoid | PReLU | base | +22.17 |
| Cheetah | Swish | base | +4.74 | Humanoid | Swish | base | +13.45 |
| Cheetah | GELU | base | +5.71 | Humanoid | GELU | base | -16.72 |
| Cheetah | FReLU | 0.4 | +16.99 | Humanoid | FReLU | 0.3 | +5.76 |
| Cheetah | FLReLU | 0.4 | +21.78 | Humanoid | FLReLU | 0.2 | +8.15 |
| Cheetah | FPReLU | 0.3 | +29.43 | Humanoid | FPReLU | 0.1 | -32.10 |
| Cheetah | FSwish | 0.3 | +16.68 | Humanoid | FSwish | 0.4 | +7.15 |
| Cheetah | FGELU | 0.5 | +17.28 | Humanoid | FGELU | 0.3 | +7.48 |
| Finger | ReLU | base | +0.00 | Walker | ReLU | base | +0.00 |
| Finger | LReLU | base | +18.85 | Walker | LReLU | base | +1.16 |
| Finger | PReLU | base | -12.42 | Walker | PReLU | base | +17.85 |
| Finger | Swish | base | -13.62 | Walker | Swish | base | +31.28 |
| Finger | GELU | base | +3.71 | Walker | GELU | base | +19.17 |
| Finger | FReLU | 0.1 | +24.80 | Walker | FReLU | 0.1 | +28.48 |
| Finger | FLReLU | 0.2 | +16.17 | Walker | FLReLU | 0.4 | -2.41 |
| Finger | FPReLU | 0.1 | +17.53 | Walker | FPReLU | 0.2 | +17.76 |
| Finger | FSwish | 0.5 | +20.25 | Walker | FSwish | 0.4 | +21.76 |
| Finger | FGELU | 0.5 | +32.19 | Walker | FGELU | 0.3 | +23.96 |

Table 22: Best configuration per activation and task for SAC with a 1-layer network. Values report mean normalized AUC improvement relative to the ReLU baseline.

| Task | Act. | $\alpha$ | $\Delta\%$ | Task | Act. | $\alpha$ | $\Delta\%$ |
|------|------|----------|------------|------|------|----------|------------|
| Ant-v4 | ReLU | base | +0.00 | HalfCheetah | ReLU | base | +0.00 |
| Ant-v4 | LReLU | base | -28.55 | HalfCheetah | LReLU | base | +8.17 |
| Ant-v4 | PReLU | base | -19.98 | HalfCheetah | PReLU | base | +2.87 |
| Ant-v4 | Swish | base | +21.11 | HalfCheetah | Swish | base | -20.94 |
| Ant-v4 | GELU | base | +19.79 | HalfCheetah | GELU | base | +24.52 |
| Ant-v4 | FReLU | 0.2 | -17.20 | HalfCheetah | FReLU | 0.2 | +28.54 |
| Ant-v4 | FLReLU | 0.1 | -16.10 | HalfCheetah | FLReLU | 0.4 | +25.21 |
| Ant-v4 | FPReLU | 0.3 | -27.00 | HalfCheetah | FPReLU | 0.1 | +16.28 |
| Ant-v4 | FSwish | 0.1 | +1.45 | HalfCheetah | FSwish | 0.5 | +16.02 |
| Ant-v4 | FGELU | 0.1 | +32.66 | HalfCheetah | FGELU | 0.4 | +24.51 |
| Cartpole | ReLU | base | +0.00 | Hopper | ReLU | base | +0.00 |
| Cartpole | LReLU | base | +1.29 | Hopper | LReLU | base | -7.08 |
| Cartpole | PReLU | base | +2.55 | Hopper | PReLU | base | +18.87 |
| Cartpole | Swish | base | -8.54 | Hopper | Swish | base | +34.79 |
| Cartpole | GELU | base | -6.24 | Hopper | GELU | base | +35.08 |
| Cartpole | FReLU | 0.2 | +3.21 | Hopper | FReLU | 0.3 | +8.44 |
| Cartpole | FLReLU | 0.2 | +3.94 | Hopper | FLReLU | 0.2 | +34.44 |
| Cartpole | FPReLU | 0.3 | +1.27 | Hopper | FPReLU | 0.1 | +43.51 |
| Cartpole | FSwish | 0.4 | +5.03 | Hopper | FSwish | 0.1 | +33.85 |
| Cartpole | FGELU | 0.2 | +2.05 | Hopper | FGELU | 0.1 | +58.23 |
| Cheetah | ReLU | base | +0.00 | Humanoid | ReLU | base | +0.00 |
| Cheetah | LReLU | base | +10.15 | Humanoid | LReLU | base | -5.71 |
| Cheetah | PReLU | base | +11.33 | Humanoid | PReLU | base | +4.88 |
| Cheetah | Swish | base | -5.12 | Humanoid | Swish | base | -7.33 |
| Cheetah | GELU | base | +15.22 | Humanoid | GELU | base | -11.18 |
| Cheetah | FReLU | 0.1 | +20.59 | Humanoid | FReLU | 0.2 | -9.93 |
| Cheetah | FLReLU | 0.1 | +27.43 | Humanoid | FLReLU | 0.4 | +1.15 |
| Cheetah | FPReLU | 0.1 | +16.06 | Humanoid | FPReLU | 0.2 | -2.78 |
| Cheetah | FSwish | 0.2 | +30.76 | Humanoid | FSwish | 0.1 | +4.75 |
| Cheetah | FGELU | 0.2 | +20.51 | Humanoid | FGELU | 0.1 | +5.45 |
| Finger | ReLU | base | +0.00 | Walker | ReLU | base | +0.00 |
| Finger | LReLU | base | -1.70 | Walker | LReLU | base | +20.65 |
| Finger | PReLU | base | -3.25 | Walker | PReLU | base | +9.99 |
| Finger | Swish | base | -33.92 | Walker | Swish | base | +8.90 |
| Finger | GELU | base | -14.16 | Walker | GELU | base | +2.64 |
| Finger | FReLU | 0.3 | +6.70 | Walker | FReLU | 0.1 | +3.36 |
| Finger | FLReLU | 0.4 | +1.35 | Walker | FLReLU | 0.1 | +7.09 |
| Finger | FPReLU | 0.5 | -7.10 | Walker | FPReLU | 0.2 | -4.43 |
| Finger | FSwish | 0.4 | +6.28 | Walker | FSwish | 0.2 | +11.62 |
| Finger | FGELU | 0.5 | +8.65 | Walker | FGELU | 0.1 | +11.51 |

Table 23: Best configuration per activation and task for TD3 with a 2-layer network. Values report mean normalized AUC improvement relative to the ReLU baseline. Placement abbreviations are A for actor, C for critic, and B for both. Task names are shortened for compactness.

| Task | Act. | $\alpha$ | Place | $\Delta\%$ | Task | Act. | $\alpha$ | Place | $\Delta\%$ |
|---|---|---|---|---|---|---|---|---|---|
| Ant | ReLU | base | r | +0.00 | HalfCheetah | ReLU | base | r | +0.00 |
| Ant | LReLU | base | all-B | -1.14 | HalfCheetah | LReLU | base | all-A | +11.27 |
| Ant | PReLU | base | all-C | -2.03 | HalfCheetah | PReLU | base | first-A | +7.17 |
| Ant | Swish | base | first-B | +3.81 | HalfCheetah | Swish | base | first-B | +46.00 |
| Ant | GELU | base | all-A | +1.47 | HalfCheetah | GELU | base | all-B | +28.96 |
| Ant | FReLU | 0.3 | first-A | +4.88 | HalfCheetah | FReLU | 0.2 | first-B | +29.91 |
| Ant | FLReLU | 0.1 | all-B | +0.33 | HalfCheetah | FLReLU | 0.1 | first-A | +63.04 |
| Ant | FPReLU | 0.1 | all-B | +5.38 | HalfCheetah | FPReLU | 0.2 | first-B | +39.27 |
| Ant | FSwish | 0.1 | first-B | +3.90 | HalfCheetah | FSwish | 0.2 | first-A | +47.57 |
| Ant | FGELU | 0.4 | first-A | +8.79 | HalfCheetah | FGELU | 0.4 | first-B | +40.70 |
| Cartpole | ReLU | base | r | +0.00 | Hopper | ReLU | base | r | +0.00 |
| Cartpole | LReLU | base | all-A | -4.10 | Hopper | LReLU | base | first-A | +18.86 |
| Cartpole | PReLU | base | first-B | -1.89 | Hopper | PReLU | base | first-B | +27.69 |
| Cartpole | Swish | base | all-C | +1.87 | Hopper | Swish | base | all-C | +19.25 |
| Cartpole | GELU | base | all-B | +2.88 | Hopper | GELU | base | first-B | +29.37 |
| Cartpole | FReLU | 0.5 | first-A | +3.24 | Hopper | FReLU | 0.5 | all-A | +27.73 |
| Cartpole | FLReLU | 0.5 | all-A | +1.94 | Hopper | FLReLU | 0.1 | first-A | +21.73 |
| Cartpole | FPReLU | 0.1 | first-A | +3.30 | Hopper | FPReLU | 0.1 | all-A | +35.09 |
| Cartpole | FSwish | 0.4 | all-C | +4.68 | Hopper | FSwish | 0.1 | first-B | +24.16 |
| Cartpole | FGELU | 0.4 | first-B | +4.38 | Hopper | FGELU | 0.4 | all-B | +28.64 |
| Cheetah | ReLU | base | r | +0.00 | Humanoid | ReLU | base | r | +0.00 |
| Cheetah | LReLU | base | all-B | -8.42 | Humanoid | LReLU | base | all-C | -0.81 |
| Cheetah | PReLU | base | first-B | -7.86 | Humanoid | PReLU | base | all-A | +3.51 |
| Cheetah | Swish | base | first-A | +2.38 | Humanoid | Swish | base | first-B | +2.56 |
| Cheetah | GELU | base | all-A | +3.39 | Humanoid | GELU | base | all-B | -2.24 |
| Cheetah | FReLU | 0.1 | all-A | +3.83 | Humanoid | FReLU | 0.4 | all-A | +4.79 |
| Cheetah | FLReLU | 0.1 | all-A | +5.64 | Humanoid | FLReLU | 0.2 | first-B | +8.07 |
| Cheetah | FPReLU | 0.4 | all-A | +5.79 | Humanoid | FPReLU | 0.1 | first-B | +7.26 |
| Cheetah | FSwish | 0.3 | all-C | +3.91 | Humanoid | FSwish | 0.5 | all-C | +10.98 |
| Cheetah | FGELU | 0.5 | all-C | +3.51 | Humanoid | FGELU | 0.2 | all-B | +9.09 |
| Finger | ReLU | base | r | +0.00 | Walker | ReLU | base | r | +0.00 |
| Finger | LReLU | base | all-A | -4.65 | Walker | LReLU | base | first-B | +0.58 |
| Finger | PReLU | base | all-C | -3.31 | Walker | PReLU | base | first-B | -6.64 |
| Finger | Swish | base | all-B | +9.93 | Walker | Swish | base | all-B | +10.34 |
| Finger | GELU | base | all-C | +12.04 | Walker | GELU | base | all-A | +9.67 |
| Finger | FReLU | 0.1 | all-C | +9.34 | Walker | FReLU | 0.1 | all-C | +9.97 |
| Finger | FLReLU | 0.1 | all-A | +9.72 | Walker | FLReLU | 0.1 | all-A | +6.93 |
| Finger | FPReLU | 0.3 | first-B | +8.19 | Walker | FPReLU | 0.5 | all-A | +9.10 |
| Finger | FSwish | 0.3 | all-B | +13.76 | Walker | FSwish | 0.5 | first-B | +6.89 |
| Finger | FGELU | 0.1 | all-B | +16.48 | Walker | FGELU | 0.3 | all-A | +11.43 |

Table 24: Best configuration per activation and task for SAC with a 2-layer network. Values report mean normalized AUC improvement relative to the ReLU baseline. Placement abbreviations are A for actor, C for critic, and B for both.

| Task | Act. | $\alpha$ | Place | $\Delta\%$ | Task | Act. | $\alpha$ | Place | $\Delta\%$ |
|------|------|----------|-------|-----------|------|------|----------|-------|-----------|
| Ant | ReLU | base | r | +0.00 | HalfCheetah | ReLU | base | r | +0.00 |
| Ant | LReLU | base | all-C | +37.54 | HalfCheetah | LReLU | base | all-B | +13.58 |
| Ant | PReLU | base | first-A | +42.33 | HalfCheetah | PReLU | base | all-C | +15.16 |
| Ant | Swish | base | all-B | +57.69 | HalfCheetah | Swish | base | all-B | +21.69 |
| Ant | GELU | base | all-C | +40.85 | HalfCheetah | GELU | base | all-C | +5.91 |
| Ant | FReLU | 0.2 | first-A | +43.08 | HalfCheetah | FReLU | 0.2 | first-A | +14.33 |
| Ant | FLReLU | 0.4 | first-A | +34.09 | HalfCheetah | FLReLU | 0.3 | first-B | +11.38 |
| Ant | FPReLU | 0.2 | all-C | +48.72 | HalfCheetah | FPReLU | 0.3 | first-B | +18.15 |
| Ant | FSwish | 0.5 | first-B | +65.67 | HalfCheetah | FSwish | 0.1 | all-C | +26.76 |
| Ant | FGELU | 0.4 | all-C | +47.05 | HalfCheetah | FGELU | 0.1 | first-B | +17.76 |
| Cartpole | ReLU | base | r | +0.00 | Hopper | ReLU | base | r | +0.00 |
| Cartpole | LReLU | base | all-C | +0.96 | Hopper | LReLU | base | all-C | +6.45 |
| Cartpole | PReLU | base | all-A | +1.61 | Hopper | PReLU | base | first-A | +9.95 |
| Cartpole | Swish | base | all-A | +2.49 | Hopper | Swish | base | all-A | +7.59 |
| Cartpole | GELU | base | first-B | +1.88 | Hopper | GELU | base | all-A | +0.39 |
| Cartpole | FReLU | 0.5 | first-A | +3.58 | Hopper | FReLU | 0.2 | all-B | +22.06 |
| Cartpole | FLReLU | 0.3 | all-A | +4.08 | Hopper | FLReLU | 0.3 | first-B | +15.13 |
| Cartpole | FPReLU | 0.5 | all-B | +2.78 | Hopper | FPReLU | 0.2 | all-B | +23.65 |
| Cartpole | FSwish | 0.5 | first-A | +3.26 | Hopper | FSwish | 0.1 | first-B | +13.89 |
| Cartpole | FGELU | 0.4 | first-B | +3.68 | Hopper | FGELU | 0.3 | first-B | +20.33 |
| Cheetah | ReLU | base | r | +0.00 | Humanoid | ReLU | base | r | +0.00 |
| Cheetah | LReLU | base | all-B | +14.04 | Humanoid | LReLU | base | first-B | -3.55 |
| Cheetah | PReLU | base | all-A | +19.43 | Humanoid | PReLU | base | all-B | +3.48 |
| Cheetah | Swish | base | all-A | +17.52 | Humanoid | Swish | base | all-B | +4.06 |
| Cheetah | GELU | base | first-A | +10.65 | Humanoid | GELU | base | all-B | +3.46 |
| Cheetah | FReLU | 0.1 | first-A | +21.59 | Humanoid | FReLU | 0.5 | first-B | +7.88 |
| Cheetah | FLReLU | 0.1 | all-A | +20.86 | Humanoid | FLReLU | 0.5 | first-B | +9.75 |
| Cheetah | FPReLU | 0.3 | all-A | +28.34 | Humanoid | FPReLU | 0.3 | all-B | +10.96 |
| Cheetah | FSwish | 0.5 | first-B | +16.55 | Humanoid | FSwish | 0.1 | first-A | +5.72 |
| Cheetah | FGELU | 0.2 | first-A | +20.16 | Humanoid | FGELU | 0.4 | first-A | +5.39 |
| Finger | ReLU | base | r | +0.00 | Walker | ReLU | base | r | +0.00 |
| Finger | LReLU | base | all-C | +11.61 | Walker | LReLU | base | all-B | +5.32 |
| Finger | PReLU | base | all-A | +14.44 | Walker | PReLU | base | all-A | +7.28 |
| Finger | Swish | base | first-A | +12.14 | Walker | Swish | base | all-A | +9.23 |
| Finger | GELU | base | all-A | +11.77 | Walker | GELU | base | all-C | +8.67 |
| Finger | FReLU | 0.5 | all-A | +11.85 | Walker | FReLU | 0.1 | first-A | +7.76 |
| Finger | FLReLU | 0.3 | all-A | +15.97 | Walker | FLReLU | 0.1 | all-A | +7.49 |
| Finger | FPReLU | 0.3 | first-A | +18.49 | Walker | FPReLU | 0.5 | first-A | +10.89 |
| Finger | FSwish | 0.3 | all-C | +16.76 | Walker | FSwish | 0.5 | all-A | +12.30 |
| Finger | FGELU | 0.1 | all-B | +15.48 | Walker | FGELU | 0.1 | first-A | +11.24 |

