# OpenReview forum: "Fractional Activation Functions for Off-Policy Reinforcement Learning: A Systematic Empirical Study"
_TMLR — Under review for TMLR_

### Review · Reviewer_tu44 · 2026-05-13

**Summary Of Contributions:**

This paper studies whether fractional-order variants of rectifier activation functions can improve off-policy deep reinforcement learning performance in continuous-control tasks. The paper addresses an underexplored question in RL: the role of activation-function design independent of algorithmic modifications. However, the work is almost entirely empirical and provides limited theoretical or mechanistic insight into why fractional activations improve RL optimization or representation quality. The comparison space is restricted mainly to rectifier-family activations and omits stronger modern baselines such as GELU, Swish/SiLU, or Mish.

**Audience:**

Yes

**Audience Explanation:**

The paper addresses a topic that is likely relevant to portions of the TMLR audience working on deep reinforcement learning, neural network architectures, and optimization. While activation functions are fundamental components of neural networks, they are comparatively underexplored in RL relative to algorithmic innovations, critic regularization, or exploration methods. The paper therefore contributes to an area that has practical and methodological relevance.

**Claims And Evidence:**

Yes

**Claims Explanation:**

The paper provides reasonably convincing empirical evidence for its main claims. The authors evaluate the proposed activations across two standard off-policy RL algorithms (TD3 and SAC), multiple continuous-control environments from MuJoCo and DMControl, and different architectural configurations. That said, several limitations reduce the overall strength of the evidence:

1. The paper primarily reports relative percentage improvements over ReLU rather than statistical significance tests or confidence intervals on the aggregate metrics.

2. The comparisons focus mainly on ReLU-family baselines. Since smoother activations such as GELU or SiLU are now common in deep learning, including them would help contextualize whether the benefits arise specifically from fractional calculus or more generally from introducing curvature/smoothness.

Overall, however, the empirical study is sufficiently extensive and carefully controlled to support the paper’s primary claim that fractional rectifier activations can improve performance in off-policy continuous-control RL.

**Requested Changes:**

1. Strengthen the statistical rigor of the evaluation. The paper should provide stronger statistical evidence supporting the reported improvements. At minimum, the authors should report confidence intervals or statistical significance tests for the main AUC comparisons. Increasing the number of seeds would further strengthen the conclusions.

2. Include stronger activation baselines beyond the ReLU family. The current comparisons are largely restricted to ReLU, Leaky ReLU, and PReLU. Since smoother activations such as GELU, Swish/SiLU, and Mish are widely used and often outperform ReLU in supervised learning, including at least one or two such baselines would help determine whether the observed gains are truly specific to fractional activations rather than simply due to smoother nonlinearities.

3. Provide deeper analysis of the mechanism behind the improvements. The paper demonstrates empirical gains but gives limited insight into why fractional activations help RL training. Additional analyses would substantially improve the contribution.

4. Clarify whether hyperparameter tuning was balanced across baselines. It is not entirely clear whether all activation functions received comparable tuning effort. Since activation functions can interact strongly with optimization hyperparameters, the paper should explicitly discuss whether learning rates or initialization schemes were re-tuned for each activation family.

---

> ### Author Response · Authors · 2026-05-18
> **Revisions for Statistical Validation**
>
> Thank you for the detailed and constructive feedback. We appreciate the reviewer’s suggestions regarding statistical validation, stronger activation baselines, mechanistic analysis, and clarification of the experimental protocol. We are currently awaiting the remaining reviewer feedback before preparing the final revised manuscript. Nevertheless, we plan to strengthen the paper in the revision along the following directions.
>
> First, to improve the statistical rigor of the evaluation, we plan to include additional statistical validation for the main AUC comparisons. Specifically, the revised paper will report bootstrap confidence intervals, paired Wilcoxon signed-rank tests, Cliff’s \\(\\delta\\) effect sizes, and sign-test consistency analyses across tasks for paired seed-level AUC comparisons. We use the Wilcoxon signed-rank test because it is a robust, nonparametric paired statistical test that assesses whether the observed improvements are consistently positive across seeds without requiring strong distributional assumptions about RL performance metrics.
>
> The table below presents the aggregate statistical summary that we plan to include in the main paper.
>
> | Algo | Layers | Positive | Mean \\(\\Delta\\%\\) | Median \\(\\Delta\\%\\) | Median \\(p\\) | Sign-test \\(p\\) | Median Cliff's \\(\\delta\\) | Overall Performance Trend |
> |---|---|---|---|---|---|---|---|---|
> | SAC | 1 layer | 7/8 | +12.8 | +6.9 | 0.2500 | 0.0352 | 0.40 | Moderate improvement |
> | SAC | 2 layers | 8/8 | +20.4 | +18.3 | 0.0625 | 0.0039 | 0.76 | Strong consistent improvement |
> | TD3 | 1 layer | 8/8 | +34.3 | +29.0 | 0.3125 | 0.0039 | 0.60 | Strong practical improvement |
> | TD3 | 2 layers | 8/8 | +17.5 | +8.9 | 0.1875 | 0.0039 | 0.60 | Consistent positive improvement |
>
> We additionally plan to provide detailed per-task statistical validation results in the appendix, including bootstrap confidence intervals, paired Wilcoxon signed-rank tests, Cliff’s $\delta$ effect sizes, activation placements, and the best-performing fractional activation configurations for each task.
>
> While additional seeds could further reduce variance and strengthen per-task statistical significance, the aggregate analysis already demonstrates a highly consistent pattern of improvement across environments. In particular, the sign-test results (e.g., 8/8 positive outcomes with \\(p=0.0039\\)) and moderate-to-large Cliff’s \\(\\delta\\) effect sizes suggest that the observed gains are systematic and practically meaningful rather than isolated to a small number of favourable runs. We also note that the study's experimental scope is relatively large, including two RL algorithms, eight benchmark tasks, six activation functions, three fractional activation families with multiple \\(\\alpha\\) values, and additional activation-placement experiments in deeper two-layer architectures. In this experimental setting, using five paired random seeds provides a fair and computationally tractable evaluation protocol.

---

> ### Author Response · Authors · 2026-05-18
> **Extended Activation Baselines, RL Mechanistic Discussion, and Experimental Protocol Clarification**
>
> We agree that comparisons against smoother modern activations can further strengthen the study. We are currently extending the experiments to include Swish/SiLU and GELU baselines. We selected these activations because they represent two distinct smooth activation function families. Swish/SiLU represents self-gated, smooth, non-monotonic activations with strong relevance to neural network function approximation in RL, while GELU represents a probabilistic, smooth activation with stable gradient propagation and strong convergence behaviour. Mish is also a strong activation function, but prior work commonly describes it as being inspired by Swish and based on similar self-gating principles. Therefore, our current view is that Swish/SiLU and GELU provide broader activation-family coverage than selecting both Swish and Mish. We would also welcome the reviewer’s perspective on whether including Mish in addition to these two baselines would be necessary.
>
> We also agree that the paper would benefit from a clearer discussion regarding why fractional activations may improve RL training. While the current Related Work section already discusses activation functions, smooth nonlinear transformations, and fractional activations in deep learning, we agree that the RL-specific mechanistic motivation can be explained more clearly. To address this concern, we plan to expand the Related Work section with a new subsection titled “Why Fractional Activations May Benefit Reinforcement Learning”. In this section, we plan to discuss how activation functions influence RL optimisation through gradient propagation, representation smoothness, and optimisation stability in actor--critic learning. We further plan to explain how smoother nonlinear transformations introduced by fractional activations may help stabilise actor--critic optimisation and produce smoother policy and value representations during RL training.
>
> Finally, we will clarify the experimental protocol regarding hyperparameter tuning. In the current experiments, all activation functions were evaluated using the same optimiser settings, learning rates, initialisation schemes, architectures, and training protocols. The baseline hyperparameters used for TD3 and SAC were adopted from modern tuned implementations commonly used in contemporary RL benchmarks and prior work. This was intended to provide strong and stable baseline training settings rather than activation-specific optimisation. Our goal was to isolate the effect of the activation function itself while avoiding confounding factors introduced by activation-aware tuning, in line with common recommendations for controlled and reproducible RL evaluation settings. We will make this protocol more explicit in the revised manuscript. We also acknowledge that different activation functions may interact differently with optimisation hyperparameters such as learning rates and initialisation schemes, and we will discuss activation-specific hyperparameter tuning as an important direction for future work.

---

### Review · Reviewer_q1Lp · 2026-05-20

**Summary Of Contributions:**

The authors evaluate 3 need fractional calculus inspired activation functions for RL and benchmark them empirically. On continuous control tasks, the authors argue these functions perform better than ReLU.

**Audience:**

Yes

**Audience Explanation:**

RL and architectural improvements to the algorithms are broadly relevant to a lot of the TMLR community.

**Claims And Evidence:**

No

**Claims Explanation:**

- The results are not compelling. Further statistical analysis is needed to make claims about the results (see e.g. https://openreview.net/forum?id=uqv8-U4lKBe), simple AUC metrics are insufficient. Additionally, numerically the numbers don't seem that impressive, but it is hard to tell what is actually a statistically significant result.
- The results are not sufficiently justified. Specifically, it is not clear why the FRELU variants should be any better. The analysis of the results is mostly just describing facts about the graphs, not like why the results are the way they are.
- Fundamentally, this is just a simple activation change, and so beyond very strong evidence that it works, it also needs compelling arguments why it works or why it is interesting. There are many many dozens of optimizers and activation functions that work slightly better on certain things, and just being slightly better on a benchmark is no longer sufficient.

**Requested Changes:**

In addition to the above strengthening of results:
- Figure 2 isn't needed in the main text, these are standard benchmarks.
- Figure 3, total_steps x label doesn't need underscore
- There are too many figures that show very similar things. Perhaps they can be consolidated or averaged, then the full results can be put in the appendix

---

> ### Author Response · Authors · 2026-05-21
> **Response to Reviewer q1Lp**
>
> We thank the reviewer for the feedback. We are awaiting the remaining reviews before preparing the final revision. Our preliminary responses to the current comments are provided below.
>
> 1. We agree that stronger statistical evaluation would improve the rigor of the study. In the revised version, we plan to follow established RL evaluation recommendations [1,2] by incorporating paired Wilcoxon signed-rank tests, bootstrap confidence intervals, and Cliff’s $\delta$ effect sizes.
>
> The experimental scope of this study is substantially larger than a standard benchmark comparison, involving:
> (i) two RL algorithms (TD3 and SAC),
> (ii) multiple activation families, including fractional activations evaluated across multiple fractional orders $\alpha$,
> (iii) multiple activation placement strategies,
> (iv) eight benchmark environments, and
> (v) multiple random seeds per configuration.
>
> Given this large combinatorial space, the normalised AUC metric was intentionally selected because it provides a compact and consistent measure that captures both learning speed and final performance across the entire training trajectory, rather than relying solely on terminal returns.
>
> Importantly, the manuscript does not rely exclusively on AUC summaries. We additionally provide full reward-versus-training-step learning curves, which remain standard practice in RL evaluation [1,2]. These plots allow direct inspection of optimisation dynamics, convergence behaviour, stability, and sample efficiency throughout training. In particular, Figures 3--6 present representative learning curves over one million environment steps for the strongest configurations of each activation family.
>
> Accordingly, the current presentation provides three complementary evaluation perspectives:
> (i) aggregate quantitative comparison through normalised AUC,
> (ii) task-level summaries through heatmaps and best-configuration tables, and
> (iii) optimisation dynamics through full learning curves.
>
> 2. We agree that the mechanistic motivation for fractional activations in RL can be strengthened. In the revised version, we plan to expand the discussion on how fractional nonlinearities may influence optimisation behaviour in actor-critic learning.
>
> 3. While the proposed modification changes only the activation function and may appear simple, the central message of this work is that activation design represents an important but comparatively underexplored component in RL research. Historically, relatively small changes to activation nonlinearities have produced improvements in optimisation stability, gradient propagation, and representation learning in deep neural networks [3--5].
>
> This issue is particularly relevant in RL, where optimisation is considerably more unstable than in supervised learning due to bootstrapped targets, non-stationary data distributions, correlated samples, and high-variance updates [1,6]. Actor-critic methods are particularly sensitive because value-estimation errors directly influence policy updates via bootstrapping [7]. Prior studies have further shown that RL performance can vary significantly across random seeds, hyperparameters, and implementation details [1,2]. Consequently, architectural components such as activation functions can meaningfully affect optimisation stability and learning dynamics.
>
> Our objective is therefore not to propose a fundamentally new RL algorithm, but rather to systematically investigate whether fractional-calculus-inspired nonlinearities provide a useful and underexplored mechanism for improving actor-critic optimisation and function approximation.
>
> 4. We appreciate the reviewer’s suggestion regarding figure redundancy. The current presentation was motivated by the large experimental scope, where heatmaps and aggregate visualisations were used to compactly summarise the high-dimensional experimental space. Nevertheless, we agree that the presentation can be streamlined further. In the revised manuscript, we plan to consolidate similar figures into more compact summary visualisations and move additional configuration-specific results to the appendix. We also agree that Figure 2 can likely be moved to the appendix, and we will correct the formatting issue in Figure 3.
>
> References
>
> [1] P. Henderson et al. *Deep Reinforcement Learning That Matters*. AAAI, 2018.
>
> [2] R. Agarwal et al. *Deep Reinforcement Learning at the Edge of the Statistical Precipice*. NeurIPS, 2021.
>
> [3] X. Glorot, A. Bordes, and Y. Bengio. *Deep Sparse Rectifier Neural Networks*. AISTATS, 2011.
>
> [4] K. He et al. *Delving Deep into Rectifiers: Surpassing Human-Level Performance on ImageNet Classification*. ICCV, 2015.
>
> [5] P. Ramachandran, B. Zoph, and Q. V. Le. *Searching for Activation Functions*. ICLR Workshop, 2018.
>
> [6] V. Mnih et al. *Human-Level Control Through Deep Reinforcement Learning*. Nature, 2015.
>
> [7] S. Fujimoto, H. van Hoof, and D. Meger. *Addressing Function Approximation Error in Actor-Critic Methods*. ICML, 2018.

---

### Review · Reviewer_5mqU · 2026-06-14

**Summary Of Contributions:**

This paper presents fractional variants of rectifier-based activation functions for off-policy reinforcement learning. Specifically, the paper investigates fractional variants of ReLU, Leaky ReLU, and Parametric ReLU, namely FReLU, FLReLU, and FPReLU in the online Reinforcement Learning setting. The authors argue that conventional rectifier-based activations have a linear structure that may limit representational flexibility, which can be particularly important in reinforcement learning where neural networks must approximate complex value functions and model expressive policy distributions.The paper evaluates these activation functions on continuous-control benchmarks from MuJoCo and the DeepMind Control Suite.

< Strenghths >

S1 (Broad empirical evaluation).
The paper provides empirical results across two widely used off-policy RL algorithms, TD3 and SAC, on MuJoCo and DMC environments.

S2 (Various activation usage and placement).
Beyond simply replacing ReLU with a fractional variant, the paper studies different ways of applying fractional variants in the actor-critic algorithms. In particular, Sections 3.4 and 5 considers different placement strategies, such as applying fractional activations to the actor, critic, both networks, or only the first layer. This provides additional empirical insight into how activation functions may interact with actor-critic algorithms.

< Weaknesses >

W1 (Insufficient evidence for the core motivation behind fractional activations).
A central motivation of the paper is that conventional rectifier-based activations may be limited by their piecewise-linear structure, restricting nonlinear function representation in reinforcement learning. However, the paper does not provide sufficient evidence for this claim in the RL setting. The current evidence mainly comes from final benchmark performance comparisons, which makes it difficult to determine whether the improvements are indeed caused by improved nonlinear representation, better optimization dynamics. To strengthen this claim, the paper would benefit from a more controlled analysis. For example, one could include a simple tabular or low-dimensional toy example showing how different activations affect value-function approximation, Bellman error propagation, or policy representation. Alternatively, a theoretical discussion measuring representation quality, gradient behavior, or function approximation error would make the motivation more convincing.

W2 (The choice of the fractional-order parameter $\alpha$ does not clearly support the main claim).
As in Section 3.2.2, $\alpha$ is in the range {0.1, 0.2, 0.3, 0.4, 0.5}, and in particular, the results in Figure 9 indicate that TD3 tends to perform best around $\alpha=$0.1 or 0.2, while performance often decreases as $\alpha$ becomes larger. Since larger $\alpha$ values correspond to stronger deviations from the conventional rectifier shape and stronger nonlinearity, this empirical trend might weaken the paper’s main claim.

W3 (Some experimental results are difficult to interpret).
The two-hidden-layer experiments introduce several activation placement strategies, including all-actor (AA), all-critic (AC), all-both (AB), first-actor (FA), and first-both (FB). While this analysis is potentially interesting, it also makes the results more difficult to interpret. For example, for Ant-v4 in Figure 8-(a), the best configurations for PReLU (AC) and FPReLU (AB) are different even on the same task.

W4 (Limited algorithmic scope and limited comparison to modern activation choices).
The paper focuses on TD3 and SAC in the online RL setting. While these are standard algorithms, the scope is still limited given the broad claim that fractional activations improve off-policy RL function approximation. Recent RL research also studies offline RL setting or other expressive policy classes (diffusion or flow policies). Many of these methods commonly use smoother activations such as Swish or GELU rather than ReLU. Therefore, it is not yet clear whether fractional rectifier activations are preferable to more modern activation functions, or whether their benefits are specific to MLPs with rectifier-based activations. To make the empirical claim more convincing, the paper should compare against stronger non-rectifier activation baselines such as Swish and GELU. It would also be valuable to evaluate the fractional variants in offline RL or with non-Gaussian policy classes, where representation and distribution modeling issues may be more pronounced.

W5 (Writing and presentation need improvement).
The paper would benefit from more careful editing. Some claims about the limitations of rectifier-based activations are repeated multiple times, which makes the manuscript longer than necessary. In addition, most citations appear to be formatted incorrectly, for example through inappropriate use of citation commands such as citep and citet.

**Audience:**

No

**Audience Explanation:**

I lean toward no in its current form.

The current paper focuses only on rectifier-based activations in TD3 and SAC, without comparison to stronger modern activation functions such as Swish or GELU. Given that many recent RL methods, including diffusion or flow policy-based methods, often use smoother nonlinearities, it is not yet clear why the community should prefer fractional rectifier variants over these existing alternatives.

**Broader Impact Concerns:**

I do not see major negative broader impact concerns from the provided empirical study. However, the paper should more clearly acknowledge the origin of the fractional activation formulations in Section 3.2. In particular, Equations 5, 6, and 7 should be clearly presented either as newly proposed formulations or as adaptations from prior fractional activation work.

**Claims And Evidence:**

No

**Claims Explanation:**

The evidence does not fully support the main claim that conventional rectifier-based activations limit RL performance specifically because of their piecewise-linear structure. As discussed in W1, W2, and W3, the benchmark results alone are insufficient to determine whether the gains come from improved nonlinear representation of fractional variants.

**Requested Changes:**

1. Could you provide additional experimental or theoretical evidence supporting the main motivation discussed in W1 and W2?
2. Please consider adding comparisons with stronger activation-function baselines such as Swish and GELU. This would help establish whether fractional rectifier activations are competitive with recent smooth activation functions.
3. Please clarify whether FReLU, FLReLU, and FPReLU are proposed in this paper or adapted from prior work.
4. The writing and presentation could be improved. Some claims are repeated several times, and the manuscript would benefit from a more concise discussion of the motivation, experimental setup, and conclusions. Citation formatting should also be corrected.

---

> ### Author Response · Authors · 2026-07-07
> **Response to Review of Paper8086 by Reviewer 5mqU**
>
> We thank the reviewer for the detailed and constructive feedback. Following our preliminary response, we have now prepared a revised paper that addresses the concerns raised in the review. Our responses to the weaknesses and requested changes are provided below:
>
> **W1**: We thank the reviewer for this important comment. In the revised manuscript, we clarify that fractional activations are intended as a lightweight modification of activation shape, not as a claim that stronger nonlinearity is always better. They modify the nonlinear transformation and gradient pathway inside the actor and critic networks while keeping TD3, SAC, replay, target updates, network width, policy objective, and training setup unchanged. For rectifier-based activations, fractionalization introduces curvature while preserving the basic ReLU-style gating structure. For smooth activations, FGELU and FSwish add fractional modulation while retaining smooth gated behavior. This is important in off-policy Actor-Critic RL because policy and value networks are trained with bootstrapped targets, replayed transitions, evolving data distributions, and coupled Actor-Critic updates. To clarify this motivation, we added a subsection, **Motivation for Fractional Activations in Actor-Critic RL**, in which we explain why activation-function design can affect policy representation, value estimation, and optimization behavior in this setting.
>
> In the Results section, the revised experiments compare rectifier baselines, smooth baselines, fractional rectifier variants, and fractional smooth variants under matched TD3 and SAC settings. The results show strong configurations across FReLU, FLReLU, FPReLU, FGELU, and FSwish, suggesting that fractionalization acts as a broader activation-design mechanism rather than only a correction to ReLU-style activations. We also added seed-level statistical validation using normalized AUC, bootstrap confidence intervals, paired Wilcoxon signed-rank tests, Cliff’s delta effect sizes, and aggregate sign tests. These additions provide stronger evidence for the consistency and practical size of the improvements. At the same time, we keep the claim careful by not claiming to fully isolate the mechanism behind the gains, but showing that fractional activation design provides a controlled and useful modification for off-policy Actor-Critic learning.
>
> We also appreciate the reviewer’s suggestion to include a controlled toy example or theoretical analysis. We agree that such an analysis would be valuable for isolating mechanisms related to value-function approximation, representation quality, and gradient behavior. In this work, our main focus was to evaluate fractional activations in modern off-policy actor-critic algorithms widely used in current deep RL research. The positive results under TD3 and SAC show that the proposed activation design is relevant in practical continuous-control settings. A more controlled low-dimensional or theoretical analysis would complement these results and could be considered an important direction for future work.
>
> **W2**: The fractional-order parameter alpha is an activation-shape hyperparameter. For rectifier-based fractional activations, alpha changes the exponent 1-alpha of the fractional-power branch. Smaller values keep the activation closer to its corresponding non-fractional baseline, while larger values introduce stronger curvature and larger deviations from the original activation shape. For smooth fractional variants, alpha controls the strength of the fractional modulation applied to the adaptive smooth core. We evaluate alpha in {0.1, 0.2, 0.3, 0.4, 0.5} to examine mild-to-moderate fractional transformations while keeping the activations close to their corresponding non-fractional baselines. The goal is to test whether controlled changes in activation shape can improve learning while preserving stability, rather than making the activation as nonlinear as possible.
>
> The revised alpha-sensitivity results support this interpretation. For rectifier-based fractional activations under TD3, smaller values such as alpha = 0.1 and alpha = 0.2 are often preferred, suggesting that mild fractional curvature can be more stable for deterministic Actor-Critic updates. However, after adding the smooth fractional variants FGELU and FSwish, we also observe that larger values, such as alpha = 0.3, 0.4, and 0.5, can perform well across several tasks and settings. For example, FGELU selects alpha = 0.5 in Finger-Spin in the single-layer setting, FSwish selects alpha = 0.5 in Ant-v4 and Walker-Walk under two-layer SAC, and FSwish selects alpha = 0.5 in Humanoid-v4 under two-layer TD3. We revised the Methodology and Results sections to clarify that alpha should be treated as a tunable activation-shape parameter whose optimal value depends on the activation family, task, algorithm, architecture, and placement, rather than as a parameter whose larger values are expected to monotonically improve performance.

---

> > ### Author Response · Authors · 2026-07-07
> > **Clarifications for W3, W4, and W5 and Requested Changes**
> >
> > **W3**:
> > We revised the presentation of the placement analysis to make the Results section clearer and more focused. The main text now reports the key aggregate trends, while more detailed configuration-level results are moved to the appendix. We also explain the purpose of each placement strategy more explicitly: actor-only placements test the effect of activation design on policy representation, critic-only placement tests the effect on value-function approximation, all-both placement tests the effect of applying the activation throughout the Actor-Critic system, and first-layer placements test whether modifying the initial state or state-action representation is sufficient to affect learning.
> >
> > We also revised the Results section to focus more on aggregate placement trends rather than isolated task-level examples. The updated results show that actor-side and early-layer placements are selected more often than purely critic-side placement. We also added task-level placement behavior in the appendix to make the analysis more transparent.
> >
> > The reviewer’s example, where PReLU and FPReLU select different placements on the same task, is now addressed through this interpretation. Different activation families may interact differently with the actor and critic because these networks play different roles in off-policy Actor-Critic learning. The revised manuscript now presents placement variation as an empirical property of the Actor-Critic architecture rather than as an inconsistency.
> >
> >  **W4**:
> > We agree with this concern and made a major revision to address it. The original manuscript focused on fractional rectifier activations, namely FReLU, FLReLU, and FPReLU, and compared them mainly against ReLU, LReLU, and PReLU. In the revised manuscript, we added GELU and Swish as stronger modern smooth activation baselines.
> >
> > We also conducted a full implementation and experimental evaluation of two smooth fractional variants, FGELU and FSwish. These variants were added to test whether fractional modulation remains useful on top of already strong smooth gated activations, rather than only within rectifier-based activation families. As a result, the revised paper now compares ReLU, LReLU, PReLU, GELU, Swish, FReLU, FLReLU, FPReLU, FGELU, and FSwish under matched TD3 and SAC settings. The revised results show that GELU and Swish are strong baselines, but fractional variants remain competitive and often outperform their corresponding non-fractional baselines. This directly addresses the concern that the original empirical comparison was limited to rectifier-style activations.
> >
> > **W5**:
> > We revised the manuscript for clarity and conciseness. Repeated claims about the limitations of rectifier activations were reduced, and the motivation was rewritten to avoid overstatement. We also revised the abstract, introduction, related work, methodology, experiments, results, conclusion, and appendix to reflect the expanded activation set. Citation formatting was corrected throughout the manuscript, including the use of appropriate parenthetical and textual citation commands. We also reorganized the appendix to improve transparency and readability.
> >
> > **Response to requested change 1. Additional evidence for W1 and W2**
> >
> > We added statistical validation, alpha sensitivity analysis, activation-family comparison, and placement analysis. We also revised the interpretation of alpha to treat it as a tunable activation-shape parameter rather than a monotonic performance parameter.
> >
> > **Response to requested change 2. Add Swish and GELU baselines**
> >
> > We added both Swish and GELU as modern smooth activation baselines. We also added FGELU and FSwish as smooth fractional variants and conducted a comprehensive experimental evaluation on the same TD3 and SAC benchmarks.
> >
> > **Response to requested change 3. Clarify whether FReLU, FLReLU, and FPReLU are proposed or adapted**
> >
> > We clarified that FReLU, FLReLU, and FPReLU are adaptations of prior fractional activation studies and are implemented here for off-policy Actor-Critic RL. We also clarified that practical implementation in RL requires additional numerical safeguards, bounded learnable parameters where applicable, and scale-control mechanisms to support stable repeated actor and critic updates. In addition, we clarified that FGELU and FSwish are introduced in this work as RL-oriented smooth fractional activation variants. This distinction is now stated explicitly in the Methodology section and summarized in the activation-function table in the appendix.
> >
> > **Response to requested change 4. Improve writing, presentation, and citations**
> >
> > We revised the writing throughout the paper, reduced repetition, reorganized the methodology and results sections, expanded the appendix, and corrected the citation formatting.

---

> > > ### Comment · Reviewer_5mqU · 2026-07-08
> > >
> > > Thank you for the detailed response and the revised manuscript. I appreciate the authors’ efforts to address the concerns, especially by adding smooth activation baselines such as GELU and Swish, introducing their fractional variants, and improving the organization of the manuscript.
> > >
> > > After carefully reading the revised version and the authors’ response, I acknowledge that the revision improves the paper in several respects. However, I still have several unresolved concerns regarding the proposed activation functions, the interpretation of the empirical results, and the strength of the evidence supporting the paper’s main claims. Therefore, I maintain my final recommendation. I believe that addressing the following concerns would further strengthen the work.
> > >
> > > 1. Behavior of the newly introduced FGELU and FSwish at zero input:
> > > In the revision, FGELU and FSwish are introduced as fractional variants of smooth activation functions. From the definitions and the activation plots in Figure 1, it appears that these functions can output a positive value even when the input is zero, especially for larger values of the fractional-order parameter $\alpha>0$. This behavior differs from many commonly used activation functions, including ReLU, GELU, and Swish, which preserve the property that zero input produces zero output. So, These newly proposed activation functions seem to introduce an implicit translation or bias. I would like the authors to clarify whether this behavior is intended, and whether it is consistent with the usual design philosophy of activation functions.
> > >
> > > 2. Potential similarity to saturating activations for large fractional orders:
> > > In Section 2.1, the manuscript discusses that classical activations such as sigmoid and tanh can suffer from vanishing gradients and slow optimization. Like these classical activations, for fractional variants with relatively large values of the fractional-order parameter, the activation curves become increasingly saturated for $x>1$. This raises the question of whether fractional variants with a high $\alpha$ may exhibit behavior similar to the classical activations. If so, they may also introduce similar optimization difficulties.
> > >
> > > 3. Need for more isolated analysis:
> > > This work still relies mainly on RL benchmark results to support the usefulness of the fractional variants. However, as noted in my original review, RL performance can be affected by many interacting factors, including exploration, bootstrapping, environmental noise, optimization noise, and hyperparameter choices. Therefore, it remains difficult to isolate the effect of the activation function itself from the current best-configuration results. In particular, for the two-layer experiments shown in Figure 6-(c, d), the results are summarized using the best configuration over both the fractional-order parameter $\alpha$ and the activation placement choices such as AA, AC, AB, FA, and FB. This best-case results makes the comparison less transparent. I believe the paper would be strengthened by providing an aggregated analysis over all configurations rather than only the best-performing cases.
> > >
> > > 4. Compared with the original manuscript, Eq. (9) in the revised manuscript defines FReLU without the normalization factor $1/\Gamma(2-\alpha)$, is this a typo or an intentional design choice?

---

> > > > ### Author Response · Authors · 2026-07-13
> > > > **Response to Remaining Concerns on Fractional Activation Behavior and Evaluation part1**
> > > >
> > > > We thank the reviewer for the careful review and constructive comments and address the remaining concerns below.
> > > >
> > > > **1. FGELU and FSwish do not produce zero at zero**
> > > >
> > > > We agree that the smooth fractional variants are **not necessarily zero-preserving**. This behavior can be interpreted as an **implicit activation offset near the origin**, and we will clarify this explicitly in the updated manuscript. It is a normal consequence of the adopted fractional formulation, rather than a plotting or implementation error.
> > > >
> > > > Our FGELU implementation follows the Grunwald--Letnikov-style fractional GELU formulation used in prior fractional-activation work [1]. In this formulation, the fractional derivative is approximated through shifted evaluations of the base function, with terms of the form \\(f(x-ih)\\). Therefore, even though standard GELU satisfies \\(GELU(0)=0\\), the fractional approximation at \\(x=0\\) also depends on shifted values such as \\(GELU(-h)\\), \\(GELU(-2h)\\), and so on. Thus, FGELU can be nonzero at the origin because of the shifted finite-difference terms, not because of a plotting or implementation error.
> > > >
> > > > For FSwish, the behavior follows from the FALU-style fractional correction applied to a Swish-like core [2]. Prior work also treats the fractional order as a tunable parameter that modifies the activation response, rather than necessarily preserving every structural property of the base activation [2,3]. In our implementation,
> > > >
> > > > $$ S_{\\alpha,\\beta}(x)=x\\sigma(\\beta_c x)+\\alpha\\sigma(\\beta_c x)(1-x\\sigma(\\beta_c x)). $$
> > > >
> > > > At \\(x=0\\), since \\(\\sigma(0)=0.5\\) and \\(x\\sigma(\\beta_c x)=0\\), we obtain
> > > >
> > > > $$ S_{\\alpha,\\beta}(0)=\\frac{\\alpha}{2}. $$
> > > >
> > > > Therefore, FSwish introduces an **origin offset** whenever \\(\\alpha>0\\). The learnable gate parameter \\(\\beta_c\\) does not change this value because \\(\\sigma(\\beta_c \\cdot 0)=\\sigma(0)=0.5\\).
> > > >
> > > > In the updated manuscript, we will describe this as an **inherent property of the adopted fractional formulation**, and we will clarify how it affects comparison with zero-preserving activations such as ReLU, GELU, and Swish. The results should be interpreted as evaluating the complete fractional activation design, including fractional modulation and possible origin offset, rather than isolating curvature alone. A centered variant, such as \\(f_{c,\\alpha}(x)=f_{\\alpha}(x)-f_{\\alpha}(0)\\), could preserve the zero-output property and is a useful direction for future work.
> > > >
> > > > **2. Large fractional orders may cause saturation-like behavior**
> > > >
> > > > We agree that large fractional orders can make the fractional variants **compression-like or saturation-like**, and we will clarify this in the updated manuscript. This concern is related to the well-known optimization difficulty of saturating nonlinearities such as sigmoid and tanh, which can produce small gradients and slow learning [5].
> > > >
> > > > For the rectifier-based fractional variants, the positive branch is
> > > >
> > > > $$ f_{\\alpha}(x)=x^{1-\\alpha}, \\quad x>0. $$
> > > >
> > > > Its derivative is
> > > >
> > > > $$ f'_{\\alpha}(x)=(1-\\alpha)x^{-\\alpha}. $$
> > > >
> > > > Thus, for \\(0<\\alpha<1\\), the slope tends to zero as \\(x\\) grows, and the activation becomes more compressed as \\(\\alpha\\) increases. These functions are not bounded like sigmoid or tanh, but they can still show **saturation-like behavior** through reduced sensitivity to large positive inputs. In the ideal form, the derivative can also become sharp near zero, which further motivates avoiding values close to one.
> > > >
> > > > The same intuition applies to FSwish. For large positive \\(x\\), \\(\\sigma(\\beta_c x)\\to 1\\), so
> > > >
> > > > $$ S_{\\alpha,\\beta}(x)\\approx \\alpha+(1-\\alpha)x. $$
> > > >
> > > > The large-positive slope is therefore approximately \\(1-\\alpha\\), which becomes small when \\(\\alpha\\) is close to one. For FGELU, the shifted Grunwald--Letnikov terms can similarly reduce the effective slope for larger \\(\\alpha\\) [1].
> > > >
> > > > This is why our experiments use the bounded low-to-mid range \\(\\alpha \\in [0.1,0.5]\\), rather than values close to one. This range allows us to study fractional activation-shape modification while avoiding the stronger saturation-like behavior expected at high fractional orders. We will add this rationale to the updated manuscript and make clear that \\(\\alpha\\) should be treated as a task-dependent hyperparameter.

---

> > > > > ### Author Response · Authors · 2026-07-13
> > > > > **Response to Remaining Concerns on Fractional Activation Behavior and Evaluation part2**
> > > > >
> > > > > **3. Activation-function isolation and best-configuration reporting**
> > > > >
> > > > > We agree that RL performance can be influenced by exploration, bootstrapping, environment stochasticity, optimization noise, and implementation details. This is a known challenge in empirical deep RL, and prior work emphasizes controlled protocols, multiple seeds, consistent evaluation, and careful reporting [6,7,8,9].
> > > > >
> > > > > Following typical empirical RL standards, we have controlled for as much as reasonably possible to isolate the activation-function effect. For each comparison, the RL algorithm, replay buffer, target updates, optimizer, network width, training budget, evaluation protocol, and random seeds are fixed. The architectural change is limited to the **activation function family and its activation-specific hyperparameters**. We do not claim that this fully explains the internal mechanism by which the activation changes learning dynamics, but it provides a controlled empirical test of whether activation design affects performance.
> > > > >
> > > > > Regarding Figure 6(c,d), the best-configuration results should be read as a **tuned-performance view**. They are not intended to imply that every \\(\\alpha\\) value or every placement is better. In our setting, \\(\\alpha\\) and placement are activation-specific hyperparameters, similar to other tunable design choices in RL. Since RL performance is known to be sensitive to such choices [6,8,9], reporting the best setting within a predefined search space is appropriate for evaluating whether an activation family can be useful when configured properly.
> > > > >
> > > > > For transparency, the appendix reports the **full configuration-level view**, including results across \\(\\alpha\\) values, placements, algorithms, and tasks. These results show that the best \\(\\alpha\\) and placement are task dependent. This is why averaging over all \\(\\alpha\\) values and placements can be misleading for our main question: it mixes tuned and untuned settings and may hide useful configurations. Such an aggregate answers whether all alpha-placement combinations are good on average, whereas our question is whether fractional activation design can improve performance when its hyperparameters are selected appropriately.
> > > > >
> > > > > In the updated manuscript, we will make this distinction clearer. Our claim is that activation choice is an important and underexplored design factor in off-policy actor--critic RL, and that fractional variants provide a simple additional shape-control mechanism that can be beneficial when \\(\\alpha\\) and placement are tuned for the task and algorithm.
> > > > >
> > > > > **4. Clarification of the FReLU formula**
> > > > >
> > > > > The change in the FReLU definition is intentional and corrects an inconsistency between the earlier written equation and the implementation used in the reported experiments. The Gamma-normalized form
> > > > >
> > > > > $$ \\frac{1}{\\Gamma(2-\\alpha)}x^{1-\\alpha} $$
> > > > >
> > > > > corresponds to the fractional derivative of the identity function and has been used in prior work on fractional ReLU variants [4]. In our implementation, however, FReLU is used as a **minimal one-sided fractional-power rectifier**:
> > > > >
> > > > > $$ F_{\\alpha}(x)=\\max(x,0)^{1-\\alpha}. $$
> > > > >
> > > > > Equivalently, this gives \\(F_{\\alpha}(x)=x^{1-\\alpha}\\) for \\(x>0\\), and \\(F_{\\alpha}(x)=0\\) for \\(x\\le 0\\).
> > > > >
> > > > > This choice is intentional. The omitted Gamma term is independent of \\(x\\), and therefore acts as an order-dependent amplitude scale rather than changing the activation curvature. Removing this extra scale keeps FReLU **lightweight** and focuses the comparison on the curvature effect introduced by the exponent \\(1-\\alpha\\). This follows the same motivation as lightweight rectifier modifications such as PReLU, where a small activation-level change can improve model fitting with little additional computational cost [10]. Both the normalized and unnormalized forms reduce to ReLU when \\(\\alpha=0\\), since \\(\\Gamma(2)=1\\).
> > > > >
> > > > > We retain the Gamma normalization in FLReLU and FPReLU because these are **two-sided fractional rectifiers**. In these variants, both positive and negative branches are fractional-power branches, and the same \\(1/\\Gamma(2-\\alpha)\\) factor keeps their relative scaling controlled as \\(\\alpha\\) changes. This is especially important for FPReLU, where the negative-side coefficient is learned. Without shared scale control, the learned negative coefficient would need to compensate for both the negative-side response and the order-dependent amplitude change. We will revise the manuscript to make this distinction explicit.

---

> > > > > > ### Author Response · Authors · 2026-07-13
> > > > > > **Response to Remaining Concerns on Fractional Activation Behavior and Evaluation part3**
> > > > > >
> > > > > > References
> > > > > >
> > > > > > [1] V. Molek and Z. Alijani. Fractional concepts in neural networks: Enhancing activation functions. Pattern Recognition Letters, 190:126--132, 2025.
> > > > > >
> > > > > > [2] J. Zamora, A. D. Rhodes, and L. Nachman. Fractional Adaptive Linear Units. Proceedings of the AAAI Conference on Artificial Intelligence, 36(8):8988--8996, 2022.
> > > > > >
> > > > > > [3] J. Zamora Esquivel, A. Cruz Vargas, R. Camacho Perez, P. Lopez Meyer, H. Cordourier, and O. Tickoo. Adaptive Activation Functions Using Fractional Calculus. Proceedings of the IEEE/CVF International Conference on Computer Vision Workshops, 2019.
> > > > > >
> > > > > > [4] M. S. Job, P. H. Bhateja, M. Gupta, K. Bingi, and B. R. Prusty. Fractional rectified linear unit activation function and its variants. Mathematical Problems in Engineering, 2022.
> > > > > >
> > > > > > [5] X. Glorot and Y. Bengio. Understanding the difficulty of training deep feedforward neural networks. Proceedings of AISTATS, 2010.
> > > > > >
> > > > > > [6] P. Henderson, R. Islam, P. Bachman, J. Pineau, D. Precup, and D. Meger. Deep Reinforcement Learning that Matters. AAAI, 2018.
> > > > > >
> > > > > > [7] R. Islam, P. Henderson, M. Gomrokchi, and D. Precup. Reproducibility of Benchmarked Deep Reinforcement Learning Tasks for Continuous Control. ICML Reproducibility in Machine Learning Workshop, 2017.
> > > > > >
> > > > > > [8] L. Engstrom, A. Ilyas, S. Santurkar, D. Tsipras, F. Janoos, L. Rudolph, and A. Madry. Implementation matters in deep policy gradients: A case study on ppo and trpo. ICLR, 2020.
> > > > > >
> > > > > > [9] A. Patterson, S. Neumann, M. White, and A. White. Empirical Design in Reinforcement Learning. JMLR, 2024.
> > > > > >
> > > > > > [10] K. He, X. Zhang, S. Ren, and J. Sun. Delving Deep into Rectifiers: Surpassing Human-Level Performance on ImageNet Classification. ICCV, 2015.

---

### Comment · Action_Editor_ruUK · 2026-05-25

Dear Reviewer 9Hdm

Could you please provide your review so that the overall review process is not extracted too much?

Thank you very much.
Best,

AE

---

### Comment · Action_Editor_ruUK · 2026-06-14

Dear Authors

The third review is now available.
You can provide your responses.

Also author-reviewer discussion has started.

AE

---

### Comment · Action_Editor_ruUK · 2026-06-26
**Extension of the response time.**

Dear Hoda and authors

Since the last review was delayed, I will give you the requested extension of the response time to July 7, 2026.

Sincerely,

AE

---

> ### Author Response · Authors · 2026-06-26
> **Thank You**
>
> Dear Professor Sung,
>
> Thank you very much for granting the extension. We sincerely appreciate your consideration and the additional time to prepare a thorough response.

---

### Author Response · Authors · 2026-07-07
**Summary of Revisions Addressing Reviewer Feedback**

We thank the reviewers for their constructive comments and the Action Editor for allowing additional time to prepare a careful revision. We have uploaded an updated version of the paper that addresses the main concerns raised during the discussion.

The revised paper includes an expanded experimental scope. In response to the request for stronger modern activation baselines, we added **GELU** and **Swish** as non-fractional smooth baselines. We also implemented and evaluated two smooth fractional variants, **FGELU** and **FSwish**. As a result, the revised study now compares ReLU, LReLU, PReLU, GELU, Swish, and their corresponding fractional variants, FReLU, FLReLU, FPReLU, FGELU, and FSwish, under matched TD3 and SAC settings.

We also revised the paper's writing and structure to clarify the motivation, methodology, and results. In particular, we added a new Related Work subsection, **Motivation for Fractional Activations in Actor-Critic RL**, to better explain why activation-function design can matter in off-policy Actor-Critic learning. We updated the Methodology section to clearly distinguish baseline activations, adapted fractional rectifier variants, and the newly introduced smooth fractional variants. We also revised the Results section to focus more directly on the key findings and moved additional detailed results to the appendix. To strengthen the empirical analysis, we added **statistical validation**, including bootstrap confidence intervals, paired Wilcoxon signed-rank tests, Cliff’s delta effect sizes, and aggregate sign tests.

We thank the reviewers again for their helpful feedback, and we look forward to any further comments.

---

### Comment · Action_Editor_ruUK · 2026-07-07

Dear Reviewers

A revision of this paper has been submitted. Please check it and provide your further comments if any.

Thank you.

AE